

# An integrative monograph of *Carex* section *Schoenoxiphium* (Cyperaceae)

Modesto Luceño[1], Tamara Villaverde[1,2], José Ignacio Márquez-Corro[1],
Rogelio Sánchez-Villegas[1], Enrique Maguilla[3], Marcial Escudero[3],
Pedro Jiménez-Mejías[4], Manuel Sánchez-Villegas[1], Monica Miguez[1],
Carmen Benítez-Benítez[1], A. Muthama Muasya[5] and
Santiago Martín-Bravo[1]

[1] Department of Molecular Biology and Biochemical Engineering, Botany area, Universidad Pablo
  de Olavide, Seville, Seville, Spain
[2] Department of Biology and Geology, Universidad de Almería, Almería, Almería, Spain
[3] Department of Plant Biology and Ecology, Universidad de Sevilla, Seville, Seville, Spain
[4] Department of Biology, Botany area, Universidad Autónoma de Madrid, Madrid, Madrid, Spain
[5] Department of Biological Sciences, Bolus Herbarium, University of Cape Town, Cape Town,
  South Africa, South Africa

Corresponding authors
Modesto Luceño, mlucgar@upo.es
Tamara Villaverde,
tvilhid@gmail.com

## ABSTRACT

*Carex* section *Schoenoxiphium* (Cariceae, Cyperaceae) is endemic to the Afrotropical
biogeographic region and is mainly distributed in southern and eastern Africa, with
its center of diversity in eastern South Africa. The taxon was formerly recognized as a
distinct genus and has a long history of taxonomic controversy. It has also an
important morphological and molecular background in particular dealing with the
complexity of its inflorescence and the phylogenetic relationships of its species.
We here present a fully updated and integrative monograph of *Carex* section
*Schoenoxiphium* based on morphological, molecular and cytogenetic data. A total of
1,017 herbarium specimens were examined and the majority of the species were
studied in the field. Previous molecular phylogenies based on Sanger-sequencing of
four nuclear and plastid DNA regions and RAD-seq were expanded. For the
first time, chromosome numbers were obtained, with cytogenetic counts on 44
populations from 15 species and one hybrid. Our taxonomic treatment recognizes
21 species, one of them herein newly described (*C. gordon-grayae*). Our results
agree with previous molecular works that have found five main lineages in
*Schoenoxiphium*. We provide detailed morphological descriptions, distribution maps
and analytical drawings of all accepted species in section *Schoenoxiphium*, an
identification key, and a thorough nomenclatural survey including 19 new
typifications and one *nomen novum*.

South Africa, Systematics, Taxonomy

# INTRODUCTION

*Carex* L. is a member of the grass-like plant family Cyperaceae (sedges). This hyperdiverse
genus is one of the three largest plant genera, with around 2,000 accepted species, which
also makes it larger than 90% of plant families (*Escudero et al., 2012*; *Global Carex Group,*

**How to cite this article** Luceño M, Villaverde T, Márquez-Corro JI, Sánchez-Villegas R, Maguilla E, Escudero M, Jiménez-Mejías P,
Sánchez-Villegas M, Miguez M, Benítez-Benítez C, Muasya AM, Martín-Bravo S. 2021. An integrative monograph of *Carex* section
*Schoenoxiphium* (Cyperaceae). PeerJ 9:e11336 DOI 10.7717/peerj.11336

2015; *Jiménez-Mejías et al., 2016*; *Govaerts et al., 2020*). *Carex* has a nearly cosmopolitan distribution, with higher species richness in cold-temperate regions of the Northern Hemisphere and lower species numbers in the Southern Hemisphere, especially at tropical latitudes. In particular, the Afrotropical biogeographic realm is the poorest one in terms of *Carex* species, with 111 species (*Martín-Bravo et al., 2019*). Nonetheless, important hotspots of *Carex* diversity and endemicity are found in southern Africa and Madagascar, which harbour the highest number of species in Sub-Saharan Africa (*Govaerts et al., 2020*; *Larridon et al., 2021*). An important proportion of South African's *Carex* diversity corresponds to section *Schoenoxiphium* (Nees) Baillon, which is considered one of the few examples of an in-situ radiation of the genus in the Afrotropical region (*Martín-Bravo et al., 2019*; *Márquez-Corro et al., 2020*).

## Taxonomic and systematic background

*Schoenoxiphium* was traditionally considered a distinct Cyperaceae genus (*Nees, 1832*), and the most primitive within tribe Cariceae on the basis of inflorescence structure (*Kükenthal, 1909*, *1910*). Its relationship with other Cariceae lineages (e.g. former genus *Kobresia* Willd. and subgenus *Vigneastra* (Tuckerman) Kük.) based on morphology have long remained controversial (see *Gehrke et al., 2010*). However, the former genus *Schoenoxiphium* has been subsumed within *Carex* (*Global Carex Group, 2015*) together with three other satellite genera (*Cymophyllus* Mack. ex Britton & A.Br., *Kobresia* and *Uncinia* Pers.), after molecular phylogenies demonstrated that they were all nested within a paraphyletic *Carex* (e.g. *Waterway & Starr, 2007*; *Starr & Ford, 2009*; *Waterway, Hoshino & Masaki, 2009*). Subsequently, the support of former genus *Schoenoxiphium*'s monophyly (*Gehrke et al., 2010*; *Villaverde et al., 2017*), together with its morphological synapomorphies led to the proposal of considering it as a *Carex* section (*Villaverde et al., 2017*). Recent phylogenetic evidence (*Jiménez-Mejías et al., 2016*; *Martín-Bravo et al., 2019*, *Villaverde et al., 2020*) indicates that section *Schoenoxiphium* is placed in subgenus *Psyllophorae* (Degland) Peterm., a main *Carex* lineage that also contains a small number of other species (e.g. *C. andina* Phil. clade, *C. distachya* Desf. clade; *Gehrke et al., 2010*; *Villaverde et al., 2017*; *Roalson et al., 2020*). The subgeneric classification of *Carex* has recently been rearranged following a genomic Hyb-Seq *Carex* phylogeny (*Villaverde et al., 2020*).

Previous molecular phylogenies focused on section *Schoenoxiphium* were based on a relatively small number of DNA regions (nuclear ITS and ETS; plastid *trn*L-F, *matK* and *rps16*). They found a well-supported internal phylogenetic backbone composed of five strongly supported main clades (Clades A-E; *Gehrke et al., 2010*; *Villaverde et al., 2017*; *Márquez-Corro et al., 2020*), although species relationships remained partially unresolved, sometimes suggesting the existence of cryptic species, which have been recently described (*Márquez-Corro et al., 2017*; this study). Furthermore, the available phylogenetic evidence suggests a complex evolutionary history, as illustrated by a documented case of intersectional hybridization followed by recombination involving section *Schoenoxiphium* (Clade E) and another member of subgenus *Psyllophorae* (*C. camptoglochin* V.I.Krecz., section *Junciformes*) (*Gehrke et al., 2010*). Molecular phylogenies have also revealed that

*C. acocksii* C. Archer, a poorly known unispicate species with remarkable morphological, molecular, biogeographical and ecological differentiation, unexpectedly belongs to the section *Schoenoxiphium* (*Márquez-Corro et al., 2020*). More recently, a phylogenomic approach based on RAD-seq has further helped to clarify the systematics of section *Schoenoxiphium* (*Villaverde et al., 2021*).

Taxonomy in section *Schoenoxiphium* is complex and characterized by long-standing problems regarding species circumscription and nomenclature (*Villaverde et al., 2017*). Thus, previous taxonomic treatments have considerably varied in the number of accepted species (6 in *Kükenthal, 1909*; 15 in *Kukkonen, 1978*, *1983*, *1986*, and in *Gordon-Gray, 1995*; c. 18–20 in recent phylogenetic studies: *Gehrke et al., 2010*; *Villaverde et al., 2017*; *Márquez-Corro et al., 2020*; *Villaverde et al., 2021*; see Table 1).

In addition, while the five main lineages detected by previous phylogenies are well-supported, there are no clear combinations of morphological synapomorphies characterizing them. Moreover, weak morphological boundaries and species non-monophyly are common patterns found within some lineages (e.g. Clades C, D and E; *Villaverde et al., 2017*). High phenotypic plasticity has also been suggested to be related with these delimitation problems (*Márquez-Corro et al., 2017*). Thus, conspicuous morphological differences (e.g. in organ size or inflorescence complexity) have been observed between populations of the same species growing at different altitudes (e.g. *C. killickii* Nelmes).

## Biogeographic and evolutionary patterns

Section *Schoenoxiphium* is endemic to the Afrotropical biogeographic region, with a clear center of diversity in eastern South Africa. It is distributed in southern and eastern Africa, including Madagascar, and marginally reaches the mountains of SE Arabian Peninsula (*Villaverde et al., 2017*). Interestingly, the high species number of the section in South Africa makes this one of the few regions worldwide where the richest *Carex* group is not the large subgenus *Carex*, but another one, in this case subgenus *Psyllophorae*.

The diversification of section *Schoenoxiphium* has been dated back to the Middle to Late Miocene (c. 8–16 mya; *Martín-Bravo et al., 2019*; *Márquez-Corro et al., 2020*; *Villaverde et al., 2021*), with its ancestral area probably located in the Drakensberg range in E South Africa, and several subsequent colonizations out of this area have been inferred, including the Cape region, tropical E Africa and Madagascar (*Márquez-Corro et al., 2020*). Active speciation processes in the Drakensberg could have taken place in concert with the uplift of this range during the Mio-Pliocene boundary (5.5 mya; *Márquez-Corro et al., 2020*). The weak morphological boundaries, species lack of monophyly, together with the often overlapping distribution of species within lineages (*Villaverde et al., 2017*; see maps in Taxonomic treatment) and their frequent turnover along various ecological gradients (elevation, wetness, forest to grassland; see habitat description under each species) suggest that ecological specialization may have played an important role in the diversification of some main lineages within section *Schoenoxiphium*, perhaps linked to geomorphological evolution in the region (*Bentley, Verboom & Bergh, 2014*).
**Table 1 Main taxonomic treatments of *Carex* section *Schoenoxiphium*.**

| Kükenthal (1909) | Kukkonen (1978, 1983, 1986) | Haines & Lye (1983)[1] | Gordon-Gray (1995)[2] | Global Carex Group (2015), Villaverde et al. (2017) | This study |
|---|---|---|---|---|---|
| | | | | Clade A | Clade A |
| | *Schoenoxiphium gracile* Cherm. | | | *Carex chermezonii* Luceño & Martín-Bravo[3] | *Carex chermezonii* Luceño & Martín-Bravo[3] |
| S. lanceum (Thunb.) Kük. | S. lanceum (Thunb.) Kük. | | | C. lancea (Thunb.) Baill. | C. lancea (Thunb.) Baill. |
| | S. madagascariense Cherm. | | S. madagascariense Cherm. | C. multispiculata Luceño & Martín-Bravo | C. multispiculata Luceño & Martín-Bravo |
| | S. schweickerdtii Merxm. & Podlech | | S. schweickerdtii Merxm. & Podlech | C. schweickerdtii (Merxm. & Podlech) Luceño & Martín-Bravo | C. schweickerdtii (Merxm. & Podlech) Luceño & Martín-Bravo |
| | | | | | Clade B |
| | S. basutorum Turrill | | S. basutorum Turrill | C. basutorum (Turrill) Luceño & Martín-Bravo | C. basutorum (Turrill) Luceño & Martín-Bravo |
| | | | S. burkei C.B. Clarke | C. burkei (C.B.Clarke) Luceño & Martín-Bravo | C. burkei (C.B.Clarke) Luceño & Martín-Bravo |
| | S. distinctum Kukkonen | | S. distinctum Kukkonen | C. distincta (Kukkonen) Luceño & Martín-Bravo | C. distincta (Kukkonen) Luceño & Martín-Bravo |
| | S. filiforme Kük. | | S. filiforme Kük. | C. killickii Nelmes | C. killickii Nelmes |
| | S. molle Kukkonen | | | | |
| | S. strictum Kukkonen | | | | |
| | | | | | Clade C |
| S. ecklonii Nees | S. ecklonii Nees | | | C. capensis Thunb. | C. capensis Thunb. |
| | S. altum Kukkonen | | | | C. sciocapensis Luceño, Márq.-Corro & Sánchez-Villegas |
| | | | | | Clade D |
| | S. perdensum Kukkonen | | S. perdensum Kukkonen | C. perdensa (Kukkonen) Luceño & Martín-Bravo | C. perdensa (Kukkonen) Luceño & Martín-Bravo |
| S. kunthianum Kük. | | S. caricoides C.B. Clarke | S. caricoides C.B. Clarke | C. spartea Wahlenb. | C. dregeana Kunth |
| S. sparteum (Wahlenb.) C.B. Clarke | S. sparteum (Wahlenb.) C.B. Clarke | S. sparteum (Wahlenb.) C.B. Clarke | S. sparteum (Wahlenb.) C.B. Clarke | | C. spartea Wahlenb. |
| S. sparteum var. schimperianum (Boeckeler) Kük. | | | S. schimperianum (Boeckeler) C.B. Clarke | C. schimperiana Boeckeler | |
| S. sparteum var. lehmannii (Nees) Kük. | S. lehmannii (Nees) Steud. | S. lehmannii (Nees) Steud. | S. lehmannii (Nees) Steud. | C. uhligii K.Schum. ex C.B. Clarke | C. esenbeckiana Boeckeler |
| | | | | | Clade E |
| | | | S. ludwigii sensu Gordon-Gray, non Hochst. | | C. badilloi Luceño & Márq.-Corro |
| S. buchananii C.B. Clarke ex Kük. | | | S. buchananii C.B. Clarke | C. kukkoneniana Luceño & Martín-Bravo | C. kukkoneniana Luceño & Martín-Bravo |

| Kükenthal (1909) | Kukkonen (1978, 1983, 1986) | Haines & Lye (1983)[1] | Gordon-Gray (1995)[2] | Global Carex Group (2015), Villaverde et al. (2017) | This study |
|---|---|---|---|---|---|
| S. rufum Nees | S. rufum Nees | S. rufum Nees | S. rufum Nees | C. ludwigii (Hochst.) Luceño & Martín-Bravo | C. ludwigii (Hochst.) Luceño & Martín-Bravo |
|  | S. bracteosum Kukkonen |  | S. bracteosum Kukkonen |  | C. bolusii C.B.Clarke[4] |
|  | S. burttii Kukkonen |  | S. burttii Kukkonen | C. pseudorufa Luceño & Martín-Bravo | C. pseudorufa Luceño & Martín-Bravo |
|  |  |  |  |  | *Incertae sedis* |
|  |  |  |  |  | C. acocksii C.Archer |
|  |  |  |  |  | C. gordon-grayae sp. nov. Luceño, Márq.-Corro & Sánchez-Villegas |

Notes:
Clades are based on *Villaverde et al. (2017)*, *Márquez-Corro et al. (2020)* and this study.
[1] Geographically limited to East Tropical Africa.
[2] Geographically limited to Kwazulu-Natal (South Africa).
[3] Unsampled in molecular phylogenies, tentative assignment to Clade A based on morphology of the type specimen.
[4] *Márquez-Corro et al. (2017)*, *C. bolusii* C.B. Clarke is the correct name of *C. parvirufa* Luceño & Márq.-Corro, see below.

## Cytogenetics

Sedges (Cyperaceae) present several uncommon cytological characteristics among angiosperms: (i) degeneration of three nuclei during pollen formation (pseudomonads), (ii) postreductional meiosis with separation of chromosomes in anaphase II instead of anaphase I (inverted meiosis), and (iii) extended kinetochoric activity during cell division (holocentric/holokinetic chromosomes). These peculiarities allow a more relaxed chromosome number inheritance, as fragments from fission events are very likely to carry functional centromeres, and fused chromosomes would not have division problems due to inverted meiosis (*Mola & Papeschi, 2006*; *Hipp, Escudero & Chung, 2013*; *Márquez-Corro et al., 2019a*). Chromosome number evolution in *Carex* is dominated by dysploid events–even within species–, with exception of the polyploid early-diverging subgenus *Siderosticta* Waterway and other minor sparse lineages (*Roalson, 2008*; *Hipp, Rothrock & Roalson, 2009*; *Escudero et al., 2012*). Some species present large dysploid series (*Luceño & Castroviejo, 1991*; *Hipp et al., 2010*). The fact of dealing with extremely variable, wide chromosome number ranges even within species has historically hindered the estimation of ancestral numbers for the genus (*Wahl, 1940*; *Roalson, 2008*), even with recent evolutionary analyses (*Escudero et al., 2014*; *Márquez-Corro et al., 2019b*, *2021*).

No cytological study has ever been carried out in *Carex* section *Schoenoxiphium*. This lack of karyological knowledge has motivated its study during the last few years. Although some of these chromosome counts have been previously used for evolutionary works at different levels (*Luceño et al., 2013*; *Márquez-Corro et al., 2019b*; *Márquez-Corro et al., 2021*), these chromosome counts are formally published here for the first time.

## Inflorescence structure

The structure of the inflorescence and the terminology of its different parts in Cyperaceae, and particularly in the *Cariceae* tribe–composed only by genus *Carex* according to the

current concept (*Global Carex Group, 2015*)–has been the subject of attention since the nineteenth century (*Kunth, 1838*; *Caurel, 1867*). However, most works dealing with this topic have been published throughout the twentieth and twenty-first centuries (*Snell, 1936*; *Blaser, 1944*; *Levyns, 1945*; *Holttum, 1948*; *Kukkonen, 1967*, *1983*, *1984*, *1990*; *Kern, 1974*; *Eiten, 1976*; *Smith & Faulkner, 1976*; *Goetghebeur, 1986*, *Reznicek, 1990*; *Bruhl, 1991*; *Timonen, 1998*; *Vegetti, 2002*, *2003*; *Richards, Bruhl & Wilson, 2006*; *Guarise & Vegetti, 2008*; *Vrijdaghs et al., 2009*, *2010*; *Molina, Acedo & Llamas, 2012*; *Reutemann et al., 2012*; *Gehrke et al., 2012*). A summary of the different interpretations may be found in *Global Carex Group (2015)* and *Jiménez-Mejías et al. (2016)*.

The inflorescence of the species in the section *Schoenoxiphium* has often been discussed in these studies due to its particularities (*Kukkonen, 1983*, *1994*; *Timonen, 1998*; *Global Carex Group, 2015*; *Jiménez-Mejías et al., 2016*). In short, the inflorescences of the species in the section vary from very simple, reduced to an androgynous spike at the end of the fertile stem, as is the case of *C. acocksii* and certain morphotypes of *C. killickii* (*Márquez-Corro et al., 2020*), to complex, constituting paniculiform inflorescences whose density and branching pattern is variable (Fig. 1).

Inflorescences of the species in the section *Schoenoxiphium* show 1–4 branching orders (Fig. 1). The shape of the last order branches (rachilla) is very typical of the species in the section: straight, flat, linear to lanceolate in outline, with one central vein and ciliated or scabrous at the margins. Complex inflorescences are composed of a variable number of spiciform or paniculiform partial inflorescences (paracladia). Frequently, the lower partial inflorescences are variably pedunculated, usually distant and sometimes nodding, while the upper ones are usually sessile or subsessile and usually appear congested in the upper part of the inflorescence, which makes their individualization quite difficult. Each partial inflorescence is subtended by a leaf-like bract (more rarely, glumaceous, setaceous or intermediate between the latter two types) and surrounded at its base by an usually tube-shaped prophyll called tubular cladoprophyll (*Jiménez-Mejías et al., 2016*). The type of prophyll in the section *Schoenoxiphium* depends on the order of branching, so that, except in unispicate inflorescences, the cladoprophyll of the first-order branches is tubular (Fig. 1) and usually hyaline, while those of the following branching orders are utriculiform cladoprophylls (Fig. 2C), bisexual or unisexual utricles (Figs. 2A, 2B) or, more rarely (*C. lancea*, *C. multispiculata* and *C. schweickerdtii*), open perigynia (Figs. 2D–2F). However, the morphology of each type of cladoprophyll is not always homogeneous, so that tubular cladoprophylls may vary from strictly tubular to hypocrateriform, symmetric or asymmetric in the apical opening (mouth); likewise, the utriculiform cladoprophylls also vary notably in shape, although those that resemble bisexual utricles predominate, and are distinguishable from unisexual utricles by their broad and obliquely truncated mouth. It is worth noting the extraordinary morphological variability of prophylls, encompassing all imaginable intermediate forms between unisexual utricles and open perigynia (see Materials and Methods for a detailed explanation of prophyll types and the terminology adopted here; Fig. 2).

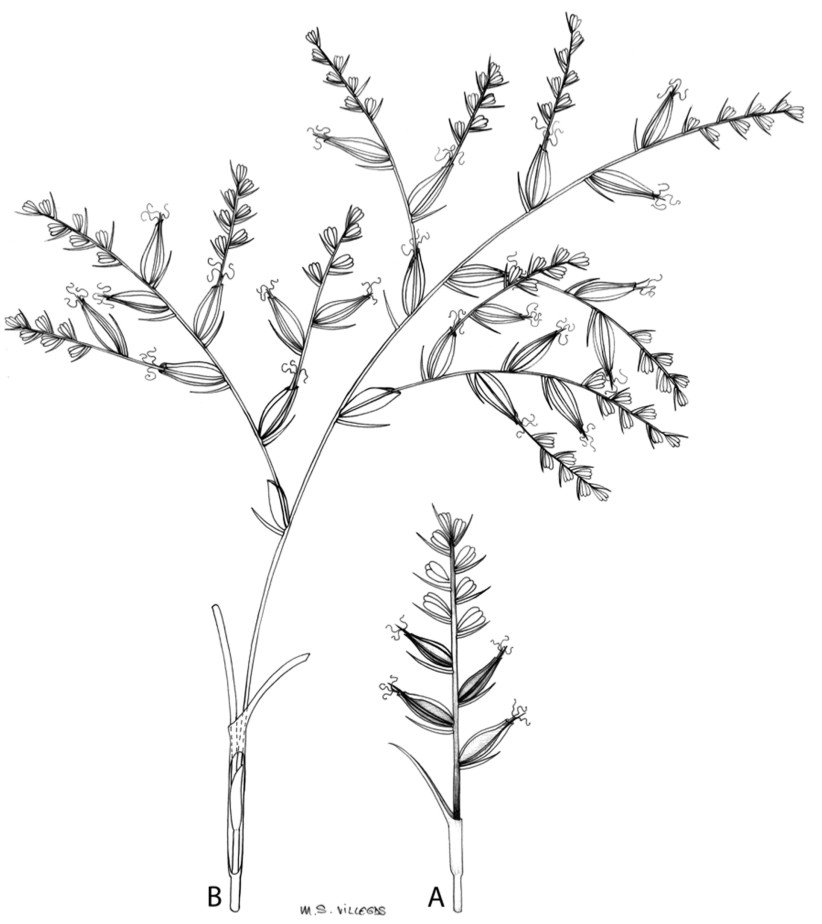

**Figure 1 Inflorescence structure in *Carex* section *Schoenoxiphium*.** (A) Simple inflorescence with a single branching order. (B) Partial inflorescence of a complex inflorescence showing the different branching orders and prophylls. Illustration by M. Sánchez-Villegas.

To summarize, there is a plethora of useful but fragmentary data that have contributed to improve our knowledge on the systematics, biogeography and evolution of section *Schoenoxiphium*. Thus, during the last 13 years, the authors of this study have been studying these aspects, which has resulted in several publications specifically focused on this group (*Gehrke et al., 2010*; *Villaverde et al., 2017*, *2021*; *Márquez-Corro et al., 2017*, *2020*; *Márquez-Corro et al., 2021*). It is particularly noteworthy how the increasing efforts in taxonomic and molecular sampling have enabled a much more robust and sound phylogenetic inference for section *Schoenoxiphium*. This has been possible thanks to the progressive development of sequencing methods (from a few Sanger-sequenced DNA regions to the massive parallel sequencing of hundreds or thousands of loci with genomic techniques like Hyb-Seq or RAD-seq). However, a critical taxonomic revision that accounts for this phylogenetic framework is still lacking. Therefore, we herein present a fully updated and integrative global monograph of the group that includes the study of more than 1,000 herbarium specimens and considers all available sources of evidence.

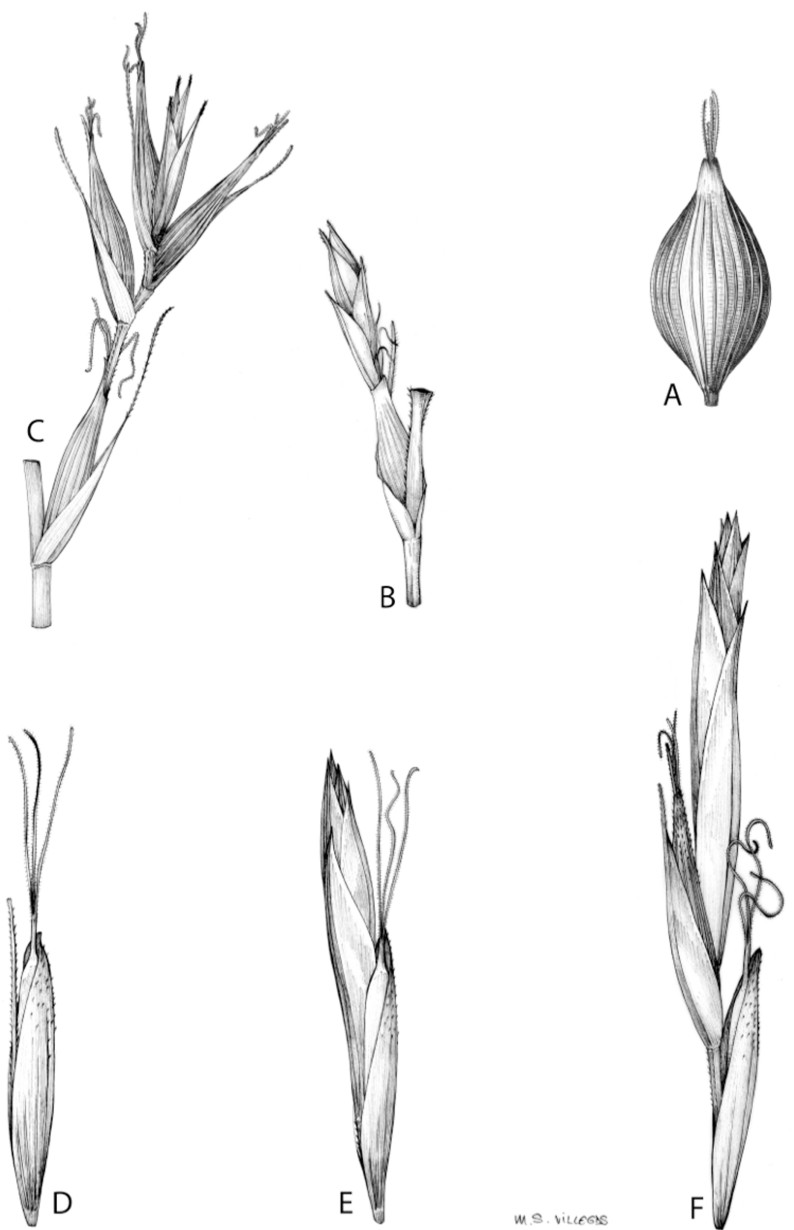

**Figure 2 Types of fertile prophylls in *Carex* section *Schoenoxiphium*.** (A) Unisexual utricle. (B) Bisexual utricle. (C) Utriculiform cladoprophyll. (D) Unisexual glumiform perigynium. (E) Bisexual glumiform perigynium. (F) Glumiform cladoprophyll. Illustration by M. Sánchez-Villegas.

## MATERIALS & METHODS

### Morphological study

We have studied 1,017 vouchers of type and representative material (Table S1 in Supplementary Material) from the following herbaria (codes following *Thiers, 2020*), through in situ visits, the request of material on loan or digitized images of herbarium specimens, and the study of specimens available in the online repository JSTOR Global

Plants (https://plants.jstor.org/): BM, BOL, E, EA, GRA, K, MA, MO, NBG, NU, P, PRE, S, SALA, SBT, TCD, TUB, UBT, UPOS, and Z. Material was also collected in the field during seven fieldwork campaigns to South Africa and Lesotho by some of the authors between 2008 and 2017, and deposited at BOL, PRE, NU and UPOS. Permits to collect were issued by CapeNature (AAA0005-00054-0028 to Abraham Muthama Muasya) and Ezemvelo KZN Wildlife (34/2008 to Abraham Muthama Muasya) and by The Lesotho Ministry of Tourism, Environment and Culture (collecting number permit MTEC/NES/ CONV/1 to Tamara Villaverde). Observations from iNaturalist (https://www.inaturalist. org) considered as reliable (i.e. identifiable with confidence) were identified to the species level; these records also added relevant chorological information. In addition, we performed a thorough nomenclatural revision to clarify the accepted names and their synonyms. We consulted all the pertinent protologues and traced for original material with the help of *Stafleu & Cowan (1976)* and herbarium staff, resulting in the finding of type material at E, H, HAL, S, SAM and UPS.

Herbarium material was identified and studied taking into account all relevant previous taxonomic literature, and with a special focus on the most important characters for the taxonomy of section *Schoenoxiphium* (*Kükenthal, 1909*; *Levyns, 1945*; *Kukkonen, 1983*; *Gordon-Gray, 1995*; *Márquez-Corro et al., 2017*; this study): width and length of rhizome internodes; lower sheath leaves bladeless or with lamina, decaying or not in fibres; width and cross section of the leaves; presence-absence of papilles and/or pricklets in leaf margins; ligule length; lowest inflorescence bract sheathing or not; inflorescence structure; length of partial inflorescences peduncles; presence-absence of open perigynia and utriculiform cladoprophylls; shape, size and indumentum of unisexual utricles; relative length of the rachilla with respect to the utricle; shape and size of the achenes; and shape of persistent style base. Macroscopic measurements were performed using a standard ruler. An Olympus SZX16 binocular magnifying glass was used to measure culm-width, leaf-width, ligules, glumes, achenes and utricles.

Regarding the terminology used in the key and in the descriptions, we basically follow the guidelines set out by *Global Carex Group (2015)* and *Jiménez-Mejías et al. (2016)*, with minor modifications. We consider rhizome "slender" when its diameter does not exceed 3 mm, "moderately stout" when the diameter ranges between 3 and 6 mm and "stout" when exceeding 6 mm in diameter. We consider that the lowest bract of the inflorescence is not sheathing when the sheath is open or closed up to 7 mm above the insertion of the bract on the culm. The basal sheaths characters are referred only to those of fertile culms. Features and dimensions of the utricle always refer to mature unisexual utricles; in the same way, the shape and dimensions of the female glumes refer exclusively to those axilating the unisexual utricles, not the bisexual ones or to the utriculiform cladoprophylls. On the contrary, achenes have been described considering those included in utriculiform cladoprophylls, in glumiform perigynia, in glumiform cladoprophylles (see below) and in unisexual and bisexual utricles, since no variation was observed. We have considered partial inflorescence (first order paracladium; *Guarise & Vegetti, 2008*) as the branch that arises directly from the main axis and branches at least twice

(Fig. 1). Regarding prophylls, *Jiménez-Mejías et al. (2016)* consider two types: (i) *perigynium* as any prophyll enclosing a female flower and surrounding the base of a terminal truncated short branch, and (ii) *cladoprophyll* as any modified prophyll surrounding lower order branches. When the margins of perigynia are fused, constituting a more or less closed structure, we distinguish two types of perigynia: (i) *bisexual utricle* (Fig. 2B), which encloses an achene and whose axis projects, outside the cladoprophyll, into a male spikelet; and (ii) *unisexual utricle* (Fig. 2A), when the axis is vestigial or protrudes from the apex of the cladoprophyll, but does not elongate into new branches that carry flowers nor a male spikelet, at most the branches carry some vestigial scales (glumes) at the apex. In addition, when the margins of cladoprophylls are fused, *Jiménez-Mejías et al. (2016)* also accept two types: (i) *tubular cladoprophyll* (Fig. 1), when it does not enclose a female flower; and (ii) *utriculiform cladoprophyll* (Fig. 2C), if they contain an achene and the axis of the branch protrudes from the apex of the cladoprophyll, generating new branches that produce female flowers and end in a male spike. Since the species of the section *Schoenoxiphium* show a great variability in the shape and position of the different types of prophylls, we accept here the classification proposed by *Jiménez-Mejías et al. (2016)*, but we will additionally refer to *glumiform perigynia*, as those last-order branch prophylls whose edges are not fused or only very shortly in the base (Figs. 2D, 2E), similar to those of the species of the former genus *Kobresia* and those observed in *C. lancea* (Thunb.) Baill., *C. multispiculata* Luceño & Martín-Bravo, and *C. schweickerdtii* (Merxm. & Podlech) Luceño & Martín-Bravo, and to *glumiform cladoprophylls* as the open or shortly fused in the base prophylls that contain an achene and whose branch axis protrudes from the apex of the cladoprophyll, generating new branches that produce female flowers and end in a male spike (Fig. 2F).

The distribution of taxa was specified using TDWG geographical codes at level 3 ("Botanical countries"; *Brummitt, 2001*), and represented in maps using the program QGis (https//qgis.org). Herbarium specimens without exact coordinates were manually georeferenced when the locality was clear and precise, in order to represent species distributions as complete as possible. Habitats description was based on field observations as well as on the classification of South African vegetation by *Mucina & Rutherford (2006)*. Analytical drawings were prepared for all accepted species by M. Sánchez-Villegas, except for *C. badilloi* Luceño & Márq.-Corro, *C. bolusii* C.B.Clarke (prepared by R. Tavera) and *C. chermezonii* Luceño & Martín-Bravo (only known from the type material), including details of the most important diagnostic characters of inflorescences, utricles and achenes. Representative iconography and selected references relevant for each accepted species were cited. The conservation status of species was reviewed and mainly obtained at the national level for South Africa (Red List of South African Plants; *SANBI, 2020*) and only for one species at the global level (*Carex ludwigii*; *IUCN, 2020*).

## Molecular study

We included new samples in the previous molecular phylogenies of section *Schoenoxiphium* based on Sanger-sequencing of DNA regions (*Villaverde et al., 2017*; ten new samples, representing six species, with all four regions each; see Table S2 in

Supplementary Material) and RAD-seq (*Villaverde et al., 2021*; 2 new samples representing two species). Thus, we expanded the taxon sampling, including one herein newly described species previously unsampled (*C. gordon-grayae* Luceño, Márq.-Corro & Sánchez-Villegas sp. nov.) and one species recently included in the section (*C. acocksii*; *Márquez-Corro et al., 2020*). Methods for DNA extraction, PCR amplification and phylogenetic analysis were similar to the ones used in the respective Sanger and RAD-seq studies (*Villaverde et al., 2017*, *2021*, respectively). For the RAD-seq assembly, we used iPyrad v.0.9.59 (*Eaton & Overcast, 2020*). Maximum likelihood (ML) trees, using the concatenated individual marker matrices and the concatenated RAD-seq matrix, independently, were inferred in RAxML 7.2.6 (*Stamatakis, 2014*) and bootstrap support for clades were calculated using 200 non-parametric replicates searches from random starting trees using an unpartitioned GTR+CAT nucleotide substitution model.

### Cytogenetic study

Cytogenetic preparations were performed through the fixation of developing pollen grains from immature anthers, following the standard protocol for *Carex* described in *Luceño (1988)* and *Escudero et al. (2008)*. Diploid numbers were inferred from obtained meiotic plates in Diakinesis (DK), Metaphase I (MI) or Metaphase II (MII) of the meiosis, as well as in Pollen Grain Mitosis (PGM), more rarely in premeiotic mitosis.

### Nomenclature

The electronic version of this article in Portable Document Format (PDF) will represent a published work according to the International Code of Nomenclature for algae, fungi, and plants (ICN), and hence the new names contained in the electronic version are effectively published under that Code from the electronic edition alone. In addition, new names contained in this work which have been issued with identifiers by IPNI will eventually be made available to the Global Names Index. The IPNI LSIDs can be resolved and the associated information viewed through any standard web browser by appending the LSID contained in this publication to the prefix "http://ipni.org/".
The online version of this work is archived and available from the following digital repositories: PeerJ, PubMed Central, and CLOCKSS.

## RESULTS

### Taxonomic revision

Our taxonomic treatment considers 21 accepted species, one of them newly described here (*C. gordon-grayae* sp. nov.). Detailed morphological descriptions, distribution maps and analytical drawings are provided for all of them, as well as a general description for section *Schoenoxiphium*. An identification key is provided to distinguish between all species. A total of 19 new formal typifications and one *nomen novum* are provided. Our exhaustive revision of materials also revealed that section *Schoenoxiphium* is distributed in 13 countries through southern and eastern Africa, with an interesting disjunction in West Africa (*C. dregeana* in W of Angola) and a population of *C. spartea* in the Republic of Yemen (Arabian Peninsula, Asia; *Al-Khulaidi, 2013*), whose voucher we have not been able

to confirm, although we consider its presence plausible. South Africa has the highest number of species (20 out of 21), followed by Lesotho (11). Within South Africa, the provinces with the greatest species richness are the Eastern Cape with 17 species and KwaZulu-Natal with 15 (Fig. 3).

## Molecular results

*Carex* section *Schoenoxiphium* consists of five evolutionary lineages (Fig. 4): (A) *C. schweickerdtii*, *C. lancea* and *C. multispiculata*; (B) *C. burkei*, *C. basutorum*, *C. distincta*, and *C. killickii*; (C) *C. capensis* and *C. sciocapensis*; (D) *C. acocksii*, *C. perdensa*, *C. dregeana*, *C. esenbeckiana*, and *C. spartea*; (E) *C. pseudorufa*, *C. ludwigii*, *C. kukkoneniana*, *C. badilloi* and *C. bolusii*. The herein described species *C. gordon-grayae* appears in an unresolved lineage.

The monophyly of *Carex* section *Schoenoxiphium* is strongly supported using both Sanger and RAD-seq datasets (97% and 100% BS, respectively; Fig. 4 and Figs. S1, S2). Most of the main lineages are strongly supported in both phylogenetic reconstructions, with the exception of Clade A. The relationships between all these clades are weakly supported in the phylogenetic reconstruction using four different DNA regions (Fig. 4A), but they are strongly supported using the genomic RAD-seq dataset (Fig. 4B). *Carex ackocsii* appears in a weakly supported lineage sister to Clade D, while *C. gordon-grayae* in an unresolved lineage (Fig. 4A). *Carex sciocapensis* is retrieved as paraphyletic (Figs. 4A, 4B). Summary statistics for the individual marker matrices obtained with AMAS (*Borowiec, 2016*) are found in Table S3. Summary statistics for the RAD assembly are found in Table S4 in Supplementary Material.

## Chromosome numbers and meiotic configurations

We report new chromosome numbers for 15 species (two thirds of the section) and one putative (morphologically intermediate) hybrid in Table 2, Figs. 4 and 5. The counts show a distribution around three chromosome number clusters (see also *Márquez-Corro et al., 2021*): $2n = 24–36$, $2n = 60–72$ and a single count of $2n = 88$ (Fig. 5). We also indicate the meiotic configuration, which may be different for the same chromosome number, due to the presence of univalents, bivalents or trivalents.

Specifically, in Clade A, *C. schweickerdtii* displayed $30^{II}$ in MI (inferred diploid $2n = 60$; Fig. 5A); *C. lancea* showed $44^{II}$ in MI (inferred diploid $2n = 88$; Fig. 5B); and *C. multispiculata* showed $31^{II}$ in MI (inferred diploid $2n = 62$; Fig. 5C), $1^{III}+29^{II}+1^{I}$ (inferred diploid $2n = 62$; Fig. 5C) and $2n = $ ca. 62 in premeiotic mitosis. In Clade B, *C. burkei* showed $33^{II}$ in MI (inferred diploid $2n = 66$; Fig. 5D), $35^{II}$ in MI (inferred diploid $2n = 70$; Fig. 5E) and $36^{II}$ in MI (inferred diploid $2n = 72$; Fig. 5F); and *C. killickii* displayed $1^{III}+34^{II}$ in MI (inferred diploid $2n = 71$; Fig. 5G), $36^{II}$ in MI (inferred diploid $2n = 72$; Fig. 5H) and $n = 36$ in pollen grain mitosis (inferred diploid $2n = 72$). In Clade C, *C. capensis* showed $n = 36$ in pollen grain mitosis (inferred diploid $2n = 72$; Fig. 5I); and *C. sciocapensis* displayed $n = 34$ in pollen grain mitosis (inferred diploid $2n = 68$; Fig. 5J). In Clade D, *C. perdensa* showed $15^{II}$ in MI ($2n = 30$; Fig. 5O) and $n = 15$ in pollen grain mitosis (inferred diploid $2n = 30$); *C. dregeana* showed $16^{II}$ in MI (inferred

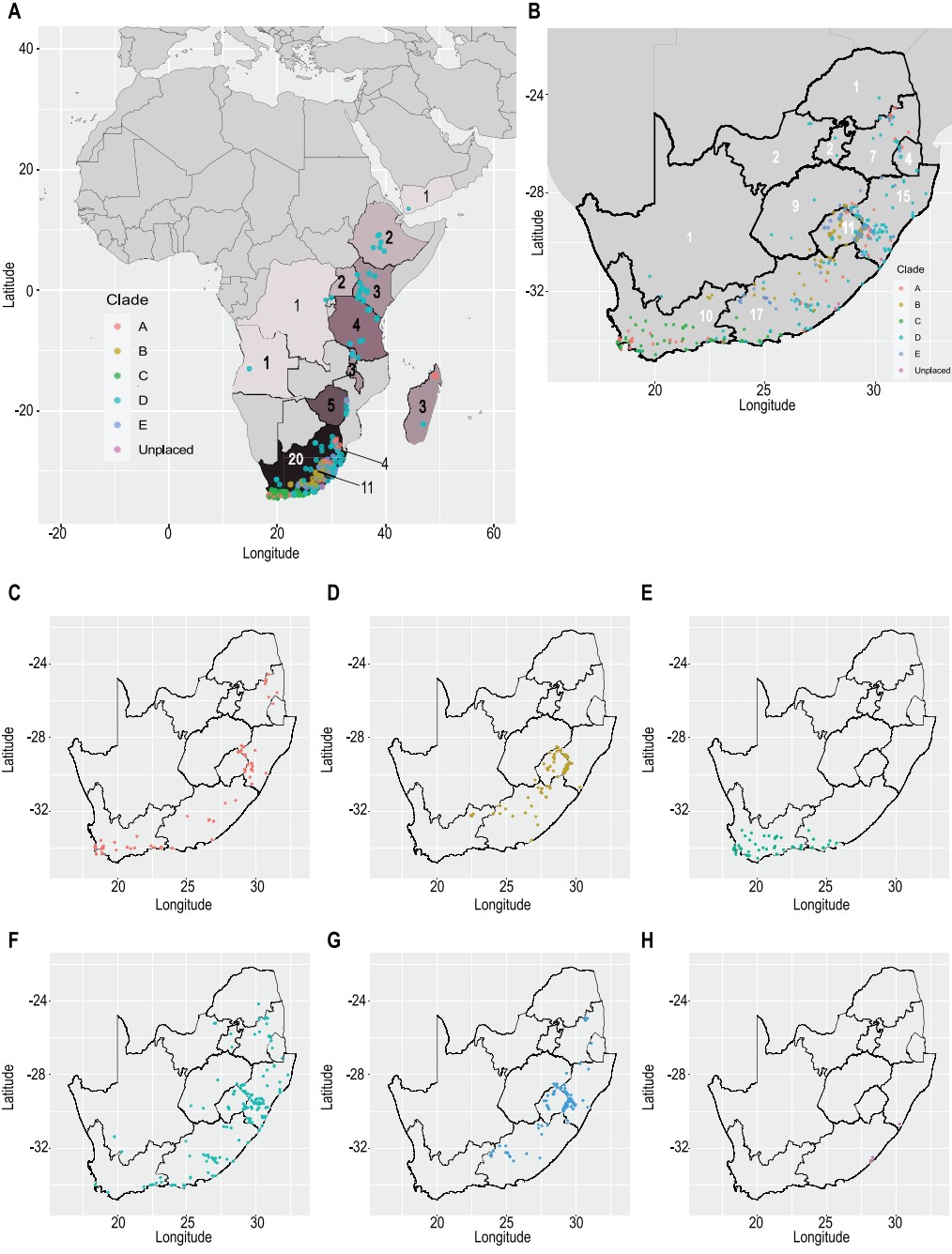

**Figure 3 Species richness of *Carex* section *Schoenoxiphium*.** (A) Complete distribution of *Schoenoxiphium* by clade. Countries are colored to reflect their *Schoenoxiphium* species richness in a brown color scale, between 1 (lightest brown) and 20 (darkest brown). (B) Distribution of *Schoenoxiphium* by clade in South Africa, Lesotho and Eswatini. Dot colors indicate a particular clade, whose species are indicated in Table 1. Total number of *Schoenoxiphium* species per country and South African provinces is indicated. (C) Distribution of clade A in southern Africa. (D) Distribution of clade B in southern Africa. (E) Distribution of clade C in southern Africa. (F) Distribution of clade D in southern Africa. (G) Distribution of clade E in southern Africa. (H) Distribution of unplaced species in southern Africa. The distribution of the species was mapped using R.

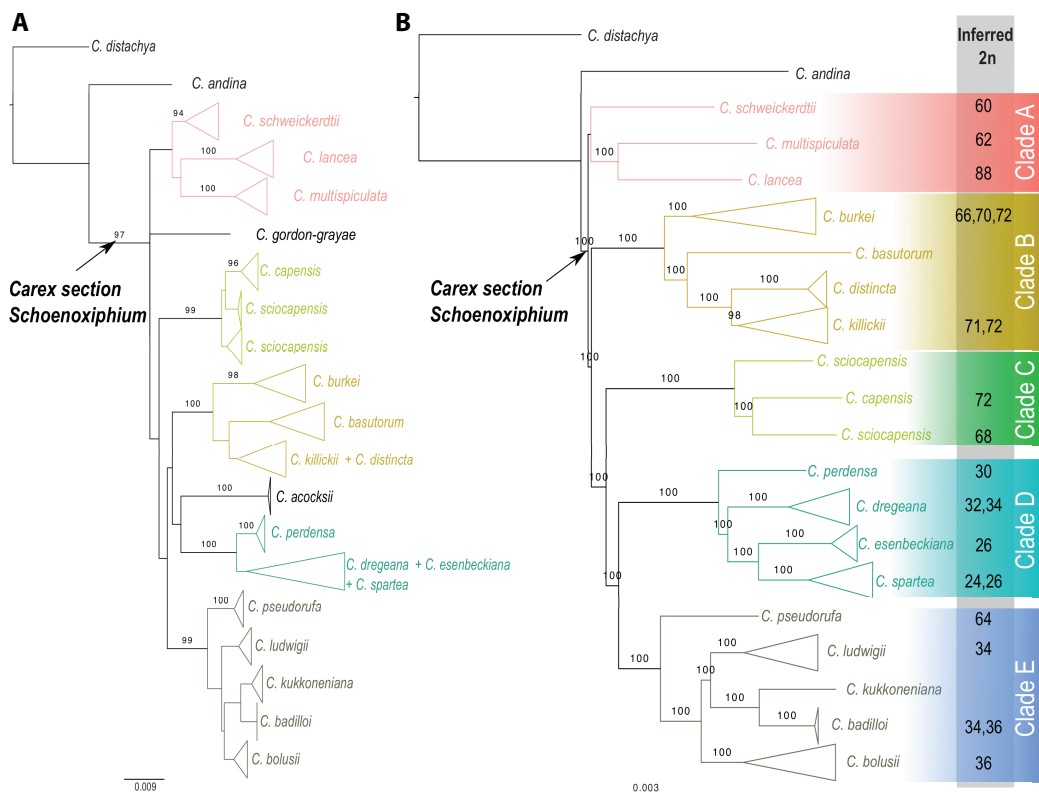

**Figure 4 Phylogenetic relationships within *Carex* section *Schoenoxiphium*.** (A) Maximum likelihood trees obtained from the RAxML analysis using a combined matrix of ETS, ITS, *matK*, and *rps16* DNA regions (120 samples, 2,944 bp). (B) Maximum likelihood trees obtained from the RAxML analysis using a RAD-seq matrix of (54 samples, 4,478,844 bp). Numbers above branches corresponding to bootstrap values are shown (only if >0.9). Lateral colored squares indicate names of the clades. Inferred chromosome numbers (2*n*) are indicated for all counted species. See Figs. S1–S2 for expanded trees.

diploid 2*n* = 32; Fig. 5P), *n* = 17 in pollen grain mitosis (inferred diploid 2*n* = 34; Fig 5Q); *C. esenbeckiana* displayed $13^{\text{II}}$ in MI (inferred diploid 2*n* = 26; Fig. 5R); and *C. spartea* displayed *n* = 12 in pollen grain mitosis (inferred diploid 2*n* = 24; Fig. 5S) and *n* = 13 in pollen grain mitosis (inferred diploid 2*n* = 26; Fig. 5T). The hybrid *C. dregeana* × *C. spartea* showed *n* = 13, 14 in pollen grain mitosis (inferred diploid 2*n* = 27), *n* = 12, 13, 14 in pollen grain mitosis (inferred diploid 2*n* = 26; Fig. 5U) and *n* = 14, 15, 16 in pollen grain mitosis (inferred diploid 2*n* = 30; Fig. 5V). Finally, in Clade E, *C. pseudorufa* showed $32^{\text{II}}$ in MI (inferred diploid 2*n* = 64; Fig. 5K); *C. bolusii* showed $18^{\text{II}}$ in DK (2*n* = 36; Fig. 5L) and *n* = 18 in pollen grain mitosis (inferred diploid 2*n* = 36); *C. badilloi* showed *n* = 17 (inferred diploid 2*n* = 34; Fig. 5M) and n = 18 (inferred diploid 2*n* = 36; Fig. 5M) in pollen grain mitosis; and *C. ludwigii* displayed $17^{\text{II}}$ in MI (inferred diploid 2*n* = 34) and *n* = 17 in pollen grain mitosis (inferred diploid 2*n* = 34; Fig. 5N).

## DISCUSSION

We have produced a fully updated and integrative study of *Carex* section *Schoenoxiphium* based on 1,017 herbarium specimens, field works conducted between 2008–2017, 164

**Table 2 Cytogenetic results.**

| Species | Chromosome number | Inferred 2n | Populations |
|---|---|---|---|
| **Clade A** | | | |
| *C. lancea* | $2n = 44^{II}$ in MI | 88 | South Africa, Western Cape, Table Mountain. IX-2009. **4544MM** (BOL). |
| *C. multispiculata* | $2n = 31^{II}$ in MI | 62 | Two populations: South Africa, KwaZulu-Natal, Cathedral Peak Nature Reserve, 1,711 m, 11-XI-2012, **53EMS12(1)** (UPOS). Ibidem, 2,287 m, **58EMS12** (UPOS). |
| | $2n = 1^{III}+29^{II}+1^{I}$ in MI | 62 | South Africa, KwaZulu-Natal, Cathedral Peak Natural Reserve, 1,711 m, 11-XI-2012, **53EMS12(2)** (UPOS). |
| | $2n = $ ca. 62 in premeiotic mitosis | ca. 62 | South Africa, KwaZulu-Natal, Mlambonja Wilderness area, 1,700 m, 17-XI-2010 **129SMB10** (UPOS). |
| *C. schweickerdtii* | $2n = 30^{II}$ in MI | 60 | South Africa, KwaZulu-Natal, Cathedral Peak Nature Reserve, 2,287 m, 11-XI-2012, **57EMS12** (UPOS). |
| **Clade B** | | | |
| *C. burkei* | $2n = 33^{II}$ in MI | 66 | South Africa, KwaZulu-Natal, Garden Castle Nature Reserve, 1,803 m, 16-XI-2012, **83EMS12** (UPOS). |
| | $2n = 35^{II}$ in MI | 70 | South Africa, KwaZulu-Natal, road to Sani Pass, 2,258 m, 21-XI-2010, **158SMB10** (UPOS). |
| | $2n = 36^{II}$ in MI | 72 | South Africa, Western Cape, W of Teepunt, Nuweveldberge, 1,844 m, 4-III-2008, **Clark 51** (UPOS). |
| *C. killickii* | $2n = 1^{III}+34^{II}$ in MI | 71 | South Africa, KwaZulu-Natal, Garden Castel Forest Reserve, 2,286 m, 19-XI-2010, **134SMB10 (**UPOS). |
| | $2n = 36^{II}$ in MI | 72 | South Africa, KwaZulu-Natal, Monk's Cowl Nature Reserve, 1,650 m, 14-XII-2008, **77ML08** (UPOS-3620). |
| | $n = 36$ in Pollen Grain Mitosis | 72 | South Africa, KwaZulu-Natal, Bushman's Nek Area, 1,846 m, 15-XI-2012, **72EMS12** (UPOS). |
| **Clade C** | | | |
| *C. capensis* | $n = 36$ in Pollen Grain Mitosis | 72 | South Africa, Western Cape, Overberg area (BOL). |
| *C. sciocapensis* | $n = 34$ in Pollen Grain Mitosis | 68 | South Africa, Western Cape, Kannaland, Towerkop Nature Reserve, 906–968 m, 12-X-2017, **142JMC17bis** (UPOS). |
| **Clade D** | | | |
| *C. perdensa* | $2n = 15^{II}$ in MI | 30 | South Africa, KwaZulu-Natal, Bushman's Nek, 1,804 m, 15-IX-2012, **74EMS12** (two individuals; UPOS). Ibidem, 1,850 m, 20-XI-2010, **152SMB10** (UPOS). |
| | $n = 15$ in Pollen Grain Mitosis | 30 | South Africa, KwaZulu-Natal, Bushman's Nek, 1,804 m, 15-IX-2012, **74EMS12** (UPOS). |
| *C. spartea* | $n = 12$ in Pollen Grain Mitosis | 24 | South Africa, KwaZulu-Natal, Bushman's Nek Natural Reserve, 1,800 m, 20-XI-2010, **154SMB10** (UPOS). |
| | $n = 13$ in Pollen Grain Mitosis | 26 | South Africa, Western Cape, Mossel Bay, 325 m, 14-X-2017, **183JMC17** (UPOS). |
| *C. dregeana* | $2n = 16^{II}$ in MI | 32 | South Africa, Free State, Scheepershoek, XI-2009, M. Muasya **4947MM** (BOL). |
| | $n = 17$ in Pollen Grain Mitosis | 34 | South Africa, KwaZulu-Natal, Bushman's Nek, 1,779 m, 15-XI-2012, **70EMS12a** (UPOS). |
| | $2n = 17^{II}$ in MI | 34 | South Africa, KwaZulu-Natal, near Boston, 1,450 m, 8-X-2009, **4900bMM** (UPOS). |
| *C. dregeana* x *C. spartea* | $n = 13, 14$ in Pollen Grain Mitosis | 27 | Two populations: South Africa, KwaZulu-Natal, Bushman's Nek, 1,779 m, 15-XI-2012, **70EMS12d** (UPOS). South Africa, KwaZulu-Natal, Garden Castle Nature Reserve, 1,803 m, 16-XI-2012, **84EMS12** (UPOS). |
| | $n = 12, 13, 14$ in Pollen Grain Mitosis | 26 | South Africa, KwaZulu-Natal, Bushman's Nek, 1,800 m, 20-XI-2010, **155SMB10** (UPOS). |

(Continued)

| Species | Chromosome number | Inferred 2n | Populations |
|---|---|---|---|
| | *n* = **14, 15, 16** in Pollen Grain Mitosis | 30 | South Africa, KwaZulu-Natal, Bushman's Nek, 1,850 m, 20-XI-2010, **153SMB10** (UPOS). |
| *C. esenbeckiana* | **2n = 13**$^{\text{II}}$ in MI | 26 | Three populations: Kenya, Laikipia county, Ndaragwa forest, 2,300 m, 25-VII-2007, **70UPO-K** (UPOS). South Africa, Western Cape, Table Mountain, 17-II-2012, **6459MM** (BOL). South Africa, Brown Hooded Kingfisher path, 14-X-2017. |
| **Clade E** | | | |
| *C. badilloi* | *n* = **17** in Pollen Grain Mitosis | 34 | South Africa, KwaZulu-Natal, Garden Castle, 1,860 m, 16-IX-2012, **76EMS12a** (UPOS). |
| | *n* = **18** in Pollen Grain Mitosis | 36 | Ibidem, **76EMS12b** (UPOS). |
| *C. ludwigii* | **2n = 17**$^{\text{II}}$ in MI | 34 | Three populations: South Africa, KwaZulu-Natal, Garden Castle Nature Reserve, 1,810 m, 16-XII-2008, **106ML08** (UPOS). Ibidem, road to Sani Pass, 2,225 m, 21-XI-2010, **162SMB10** (UPOS). Ibidem, **159SMB10** (UPOS). |
| | *n* = **17** in Pollen Grain Mitosis | 34 | Three populations: South Africa, KwaZulu-Natal, Nganang river, Drakensberg Gardens hotel, 1,794 m, 9-XI-2009, **4922MM** (UPOS). Ibidem, road to Sani Pass, 2,225 m, 21-XI-2010, **162SMB10** (UPOS). Ibidem, **159SMB10** (UPOS). |
| *C. bolusii* | **2n = 18**$^{\text{II}}$ in MI | 36 | Two populations: South Africa, Free State, Golden Gate Highlands National Park, 1,967 m, 15-I-2010, **42ML10** (UPOS). Ibidem, Bushman's Nek Nature Reserve, 1,779 m, 15-XI-2012, **70EMS12b** (UPOS). |
| | *n* = **18** in Pollen Grain Mitosis | 36 | Two populations: South Africa, Free State, Golden Gate Highlands National Park, 1,800-2,000 m, 13-XII-2008, **143SMB08** (UPOS). Ibidem, KwaZulu-Natal, Garden Castle Nature Reserve, 1,803 m, 16-XI-2012, **77EMS12** (UPOS). Ibidem, 2,267 m, 19-XI-2010, **141SMB10** (UPOS). |
| *C. pseudorufa* | **2n = 32**$^{\text{II}}$ in MI | 64 | South Africa, KwaZulu-Natal, Cathedral Peak Nature Reserve, 2,307 m, 11-XI-2012, **59EMS12** (UPOS). |

**Note:**
Species names, chromosome number (in Metaphase I or Pollen Grain Mitosis), inferred diploid chromosome number and information of population of origin are indicated. Populations data include country, province, locality, altitude, collection code (in bold) and the herbarium where the witness voucher is preserved. The symbols I, II and III represent univalent, bivalent and trivalent chromosome association, respectively.

sequenced samples represented in a Sanger and a RAD-seq phylogeny, cytogenetic counts on 44 populations, a nomenclatural survey and an exhaustive review of previous taxonomic and phylogenetic works.

Our taxonomic treatment considers 21 accepted species (Table 3), one of them newly described here (*C. gordon-grayae* sp. nov.). We provide detailed morphological descriptions, distribution maps, analytical drawings and an identification key including all species, as well as a general description for section *Schoenoxiphium*. A total of 19 new formal typifications and one *nomen novum* are provided.

## Systematics of *Schoenoxiphium*

*Carex* section *Schoenoxiphium* consists of five well-supported evolutionary lineages: (A) *C. schweickerdtii*, *C. lancea* and *C. multispiculata*; (B) *C. burkei*, *C. basutorum*, *C. distincta* and *C. killickii*; (C) *C. capensis* and *C. sciocapensis*; (D) *C. acocksii*, *C. perdensa*, *C. dregeana*, *C. esenbeckiana* and *C. spartea*; (E) *C. pseudorufa*, *C. ludwigii*, *C. kukkoneniana*, *C. badilloi* and *C. bolusii*. Although Clade A is retrieved in a weakly supported lineage (Fig. 4), it has been shown to be strongly supported in *Márquez-Corro*
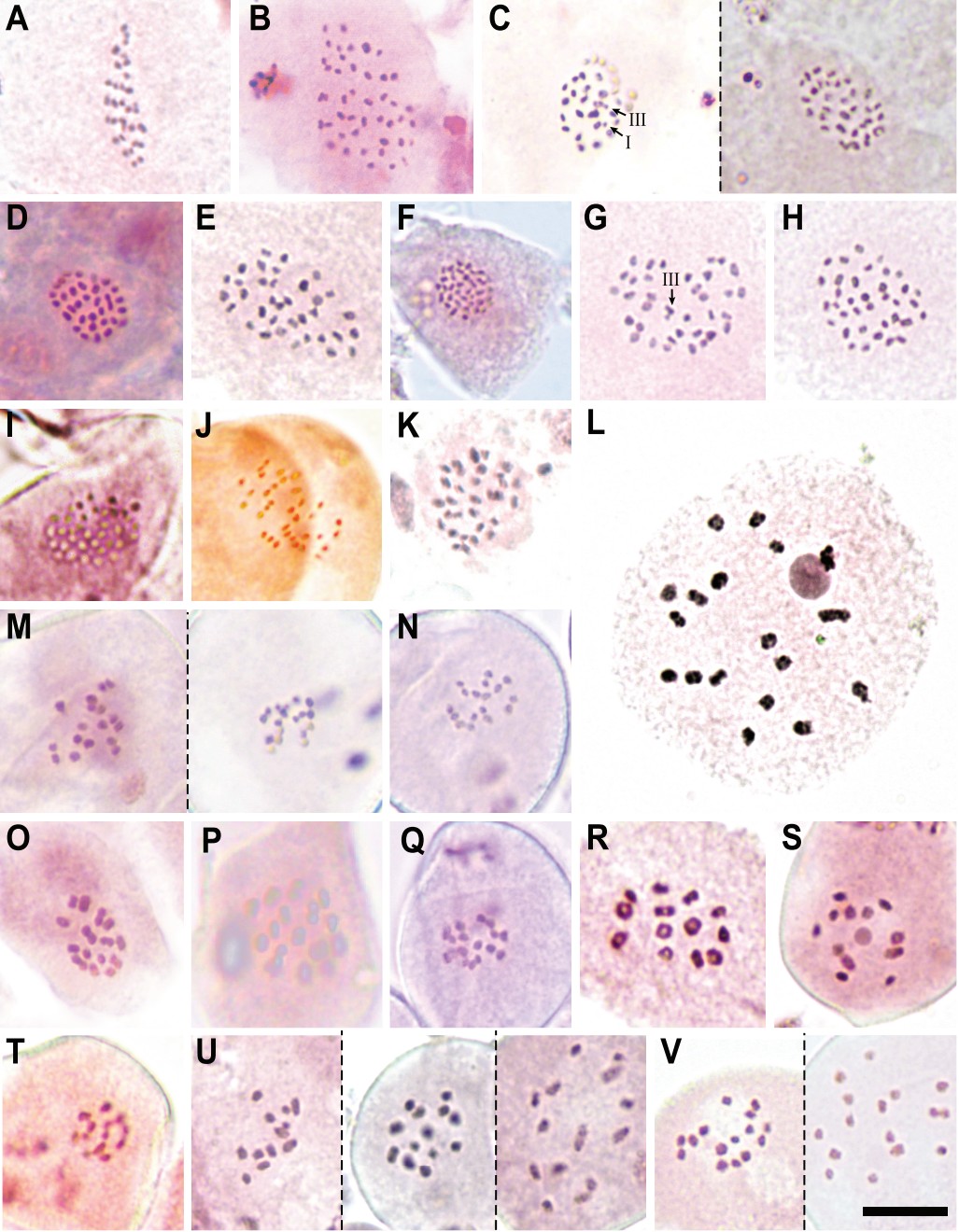

**Figure 5 Chromosome counts for all the configurations found in *Carex* section *Schoenoxiphium*.**
(A) *C. schweickerdtii* (57EMS12, 30$^{II}$). (B) *C. lancea* (MM4544, 44$^{II}$). (C) *C. multispiculata* (53EMS12, 1$^{III}$+29$^{II}$+1$^{I}$ and 31$^{II}$, respectively). (D–F) *C. burkei* (83EMS12, 33$^{II}$; 158SMB10, 35$^{II}$; Clark51, 36$^{II}$). (G–H) *C. killickii* (134SMB10, 1$^{III}$+34$^{II}$; 77ML08, 36$^{II}$). (I) *C. capensis* (Western Cape, *n* = 36). (J) *C. sciocapensis* (142JMC17bis, *n* = 34). (K) *C. pseudorufa* (59EMS12, 32$^{II}$). (L) *C. bolusii* (42ML10, 18$^{II}$). (M) *C. badilloi* (76EMS12a,b; *n* = 17 and 18, respectively). (N) *C. ludwigii* (78EMS12, n = 17). (O) *C. perdensa* (74EMS12, 15$^{II}$). (P–Q) *C. dregeana* (MM4947, 16$^{II}$; 70EMS12a, *n* =17). (R) *C. esenbeckiana* (70UPO-K, 13$^{II}$). (S–T) *C. spartea* (154SMB10, *n* = 12; 183JMC17, n = 13). (U) *C. dregeana* × *C. spartea* (155SMB10, *n* = 12, 13 and 14, respectively). (V) *C. dregeana* × *C. spartea* (153SMB10, *n* = 15 and 16, respectively). The symbols I, II and III represent univalent, bivalent and trivalent chromosome association, respectively. Scale bar at bottom right = 10 μm.

**Table 3 Summary of the distribution, habitat and elevation range of the species of *Carex* section *Schoenoxiphium* accepted in this monograph grouped by the main lineages retrieved in the molecular phylogenies.**

| Species | Distribution | Habitat | Elevation |
|---|---|---|---|
| **Clade A** | | | |
| *C. chermezonii* | Madagascar | Mountain forests | c. 2,500 m |
| *C. lancea* | South Africa (SA) | Edges of streams and other damp and shady places in forest | 20–1,500 m |
| *C. multispiculata* | Eswatini, Madagascar and SA | Edges of streams, meadows and other damp places in mountains | 1,100–2,800 m |
| *C. schweickerdtii* | Lesotho, SA and Zimbabwe | Stony meadows and, more rarely, edges of streams and pools in mountains | 1,400–3,100 m |
| **Clade B** | | | |
| *C. basutorum* | Lesotho, SA | Dry slopes and grassland in scrubs | 1,500–2,100 m |
| *C. burkei* | Lesotho, SA | Grassland, clearing of shrub, usually in dry places, but sometimes in temporary flooded meadows and edges of streams | 1,400–2,900 m |
| *C. distincta* | Lesotho, SA | Open, stony and usually dry meadows and shrubs at high altitudes | (1,990)2,300–3,000 m |
| *C. killickii* | Lesotho, SA | Open, stony and dry to mesophilous meadows, and shrubs, usually at high altitudes | (550)1,650–3,150 m |
| **Clade C** | | | |
| *C. capensis* | SA | Open and sunny places in renosterveld vegetation, growing on shale soils, commonly observed in after-fire vegetation fynbos | 5–750 m |
| *C. sciocapensis* | SA | Shady places, mainly in margins and clearing of southern afrotemperate forest, but also in fynbos areas, especially on shale and sandstone soils | 10–1,500 m |
| **Clade D** | | | |
| *C. acocksii* | SA | Open grounds on dolerite summits | 1,450–1,600 m |
| *C. perdensa* | SA | Grassland | 1,200–1,950 m |
| *C. spartea* | E & S Africa and SE Arabian Peninsula | Dry to damp grassland, edges of streams, grassy clearing of shrubs | 45–2,800 m (up to 2,310 m in South Africa) |
| *C. dregeana* | E & S of Africa, W of Angola and Madagascar | Grassland | 200–2,850 m |
| *C. esenbeckiana* | E & S of Africa | Shady places, mainly in forest understories | 4–2,300 m |
| **Clade E** | | | |
| *C. badilloi* | Lesotho, SA | Mesophilous, open grassland on clay soils | 1,750–2,500 m |
| *C. bolusii* | Lesotho, SA | Edges of streams, grasslands, damp meadows and open bushy places | (1,400)1,650–3,150 m |
| *C. kukkoneniana* | E Africa, from Tanzania to SA | Edges of streams, marshy grounds and other damp soils in mountains | 950–2,500 m |
| *C. ludwigii* | Lesotho, SA | Edges of streams and other damp places in mountains | 1,350–2,850 m |
| *C. pseudorufa* | SA | Edges of streams and other wet places in mountains | 1,850–3,200 m |
| **Incertae sedis** | | | |
| *C. gordon-grayae* | SA | Shady places, mainly in margins and clearing of forests in the KwaZulu-Natal-Eastern Cape Coastal Belt | 170–400 m |

*et al. (2020)*. Similar situation occurs with *C. acocksii*, which is found in a moderately supported lineage in *Márquez-Corro et al. (2020)* but it lacks of support here. *Carex gordon-grayae* sp. nov. and *C. chermezonii* (the later not included in any molecular analysis to date) are the only two species whose phylogenetic relationships have not been resolved

yet. However, their morphological characteristics led us to hypothesize that *C. chermezonii* would belong to Clade A, and *C. gordon-grayae* to an isolated lineage.

Section *Schoenoxiphium* probably originated in the Mid-Miocene in the Afrotropical region and diversified in-situ (mean c. 8–16 mya; *Martín-Bravo et al., 2019*; *Márquez-Corro et al., 2020*; *Villaverde et al., 2021*), probably in the Drakensberg and adjacent mountain ranges (*Márquez-Corro et al., 2020*). Most of the species in *Schoenoxiphium* are suspected to have originated during the Late Miocene (*Villaverde et al., 2021*), which corresponds to the uplift of the Drakensberg Mountains, the only South Africa's alpine zones. The provinces of Eastern Cape and KwaZulu-Natal treasure the highest number of section *Schoenoxiphium* species in South Africa (17 and 15, respectively; Fig. 3) followed by the country of Lesotho (11 species).

(A) *C. schweickerdtii, C. lancea, C. multispiculata*, (and *C. chermezonii?*).

This small clade of three species (or perhaps four; our untested hypothesis is that *C. chermezonii* belongs to this clade) is formed by large plants with wide leaves and straight apex leaves (but usually somewhat curved in *C. lancea*). Unlike other species in the section, these species may have open perigynia (occasional in *C. multispiculata*; not seen in *C. chermezonii*). Utricles are smooth, linear and gradually attenuated into a smooth beak; usually of medium size (4.4–6.5 mm), but C. *schweickerdtii* may display among the largest utricles in the section (6–10 mm). Species in this clade are mainly distributed in southern Africa; additionally, *C. schweickerdtii* is also found in Zimbabwe, *C. multispiculata* in Madagascar, and *C. chermezonii* has only been found in its type locality in the Tsaratanana mountains of Madagascar. They usually occur in edges of streams or damp places in medium-high mountains (*C. schweickerdtii* also in stony meadows), although *C. lancea* is found in forests at low elevations (20–1,500 m).

(B) *C. burkei, C. distincta, C. basutorum* and *C. killickii*.

Species in this clade are characterized by plants of small height, with very narrow or filiform leaves (0.2–3.7 mm, but *C. burkei* has wider ones, 1.5–5.5 mm) curved to curled at the tip. Utricles are linear or narrowly ellipsoid, more or less hispid in the upper half, and smaller than 5 mm, with the exception of *C. basutorum* (5–7.2 mm) and some forms of *C. killickii*. This clade of four species is endemic to Lesotho and South Africa and they occur at medium-high elevations (1,400–3,150 m; but *C. killickii* has been exceptionally found at 550 m). They are typically found in open and dry grasslands, although *C. killickii* also occur in mesophilous meadows and *C. burkei* in temporary flooded meadows and edges of streams.

(C) *C. capensis* and *C. sciocapensis*

*Carex capensis* and *C. sciocapensis* have basal sheaths that are bladeless and the lowest bract of the inflorescence is not sheathing. The remaining species of the section have basal sheaths with lamina (except sometimes only the lowest with lamina), and a sheathing lowest bract in the inflorescence (except *C. basutorum, C. chermezonii* and *C. multispiculata*). *Carex capensis* and *C. sciocapensis* are two species endemic to the southernmost areas of the Cape region (South Africa). Both species occur at low elevations (5–750 m) and *C. sciocapensis* can also be found at medium elevations (below 1,500 m).

*Carex capensis* inhabit open and sunny places whereas *C. sciocapensis* is common in shady places and margins and clearing of forest (and sometimes in fynbos areas).

**(D) *C. acocksii, C. perdensa, C. dregeana, C. spartea* and *C. esenbeckiana***

The clade grouping *C. acocksii, C. perdensa, C. dregeana, C. spartea* and *C. esenbeckiana* is characterized by fibrous basal sheaths (although in *C. perdensa* are entire or scarcely fibrous) and by smooth utricles up to 5.5 mm (rarely 6 mm in *C. esenbeckiana*). This clade of five species has the broadest distribution. *Carex dregeana, C. spartea* and *C. esenbeckiana* are distributed across S and E Africa (*C. dregeana* also disjunctly in Angola and Madagascar) from low to high elevations (4–2,850 m). *Carex spartea* is the species with the northernmost distribution in the section, reaching SE Arabian Peninsula. In contrast with these widely distributed species, *C. acocksii* and *C. perdensa* are restricted endemics in South Africa, where they are found at medium-high elevations (1,200–1,950 m). All of these species occur in open places, except *C. esenbeckiana*, which is found in shady places in afromontane forests.

**(E) *C. pseudorufa, C. ludwigii, C. kukkoneniana, C. badilloi* and *C. bolusii***

The clade grouping *C. pseudorufa, C. ludwigii, C. kukkoneniana, C. badilloi* and *C. bolusii* is characterized by basal sheaths entire to slightly fibrous and by scabrid utricles towards the apex. These species are mainly distributed in South Africa and Lesotho, while *C. kukkoneniana* reaches further north to Tanzania. They are found in edges of streams and other wet places at medium-high elevations, although *C. badilloi* and *C. bolusii* can also be found in grasslands and open places.

*Carex gordon-grayae*

*Carex gordon-grayae*, long confused with *C. esenbeckiana* (Clade D), shows a number of distinctive morphological characters that do not match any of the groups described so far: (i) basal sheaths not or scarcely fibrous, typical of all groups except the Clade D; (ii) broad leaves (up to 9 mm), as in Clade A species; (iii) subsessile or shortly pedunculated partial inflorescences, which is common in the Clades B, C and D; (iv) lowest bract sheathing, unlike Clade C; (v) hyaline glumes and pyramidal style base, as in *C. acocksii*; and (vi) long (5.4–7.9 mm) utricles narrowly linear and gradually attenuated in a long beak, unlike the D and E Clades. Moreover, this species is a rare endemic to SE of KwaZulu-Natal and E of Eastern Cape, where it inhabits forest near the coast, unlike the remaining species of the section, except *C. esenbeckiana* and *C. sciocapensis*.

## Cytogenetics

Chromosome number evolution in *Carex* is dominated by dysploid events (*Roalson, 2008*; *Hipp, Rothrock & Roalson, 2009*; *Escudero et al., 2012*) with some species presenting large dysploid series (*Luceño & Castroviejo, 1991*; *Hipp et al., 2010*). The section *Schoenoxiphium* does not seem an exception to this general pattern in genus *Carex*, and it displays a wide range of chromosome numbers, from $2n = 24$ to $2n = 88$ (*C. spartea* and *C. lancea*, respectively; Table 2, Figs. 4 and 5). However, the distribution of the counts is rather discontinuous either overall and within some of the clades. High numbers have been reported for the Clade A ($2n = 60$, 62 and 88), Clade B ($2n = 66$, 70–72), Clade C ($2n = 68$, 72) and part of the Clade E ($2n = 64$). The consistent presence of relatively high

chromosome numbers in all these clades suggests a likely high ancestral number for the section. The most diverse and karyologically complex would be Clade D, in which the chromosome counts are reduced to half ($2n = 24, 26, 27, 30, 32, 34$) and in Clade E ($2n = 34, 36$), excluding *C. pseudorufa* ($2n = 64$). This conspicuous reduction of chromosome numbers is probably due to a massive series of fusion events that have occurred through the diversification of the lineage, rather than several polyploid events in the Clades A, B, C and E. In fact, fusion events have been inferred to happen ca. 1.5 times as fission events in the non-*Siderostictae Carex* (Supplementary Data 7 in *Márquez-Corro et al., 2019b*). Moreover, a preliminary study that is currently being carried out considering genome size variation among and within species of the section points to the fusion hypothesis, because genome sizes do not vary proportionally to chromosome number (*Márquez-Corro et al., 2021*). However, the detection of an unusual high number within one of the two reduced lineages (i.e., *C. pseudorufa*) is very intriguing. Genome size of this species is yet unknown, so there are two possible hypotheses: either a rare polyploid event in *C. pseudorufa* or, at least, two convergent, independent fusion events in the reduced lineages (based on the topology retrieved in the RAD-seq phylogeny, see Fig. 4). Chromosome number variation of the section is very promising and posits possible evolutionary scenarios in which the establishment in different niches with sympatric species could be through karyotype-related adaptation, since most members of the section inhabit the Drakensberg area (including most of the Eastern Cape mountains).

## Taxonomic treatment

*CAREX* sect. *SCHOENOXIPHIUM* (Nees) Baillon, Hist. pl., monogr. Cypér.: 345, 1894 [1893]

≡ *Schoenoxiphium* Nees in Linnaea 7: 531, 1832 [basionym]

*Type*: *Schoenoxiphium capense* Nees (=*Schoenus lanceus* Thunb.)

*Etymology*: From the Greek σχοῖνος (schoĩnos), rush, and χίφος (xíphos), sword; probably because of the shape and sharp edges of the leaves of some of its species.

Perennial herbs, not caespitose to densely caespitose. Rhizome with short to long internodes, brown. Flowering culms (4.5)10–180(200) cm, erect or, more rarely, nodding, acutely to obtusely trigonous, smooth to scabrid, 0.4–4 mm wide at the middle. Leaves (0.2)0.4–15(18) mm wide, shorter to longer than the inflorescence, soft to coriaceous, filiform to linear, flat, involute, canaliculate, carinate or plicate, usually scabrid at the margin and distal part of abaxial midrib, with straight to curled apex; ligule (0.1)0.3–12 (17) mm long. Basal sheaths entire to very fibrous, usually with lamina, but sometimes lowermost bladeless. Lowest bract of the inflorescence leaf-like, more rarely glumaceous or setaceous, shorter to longer than the inflorescence, not sheathing or with a sheath up to 84 cm long. Inflorescence branching 1–4 times, reduced to a single, terminal spike or, more frequently, composed by several panicles and/or spikes, one terminal and the

remaining lateral (partial inflorescences) sessile to longly pedunculate, overlapping to distant, erect to nodding. Glumiform perigynia and glumiform cladoprophylls rarely present. Tubular cladoprophylls always present except in *C. acocksii* and unispicate morphotypes of *C. killickii*. Utriculiform cladoprophylls frequently present. Male glumes usually ovate to lanceolate, more rarely oblong, obovate or elliptic, brown to yellowish-brown, with a green central band, ending in an aculeate mucro or ariste, more rarely acuminate, acute or obtuse. Female glumes usually ovate, more rarely elliptic, lanceolate, obovate, oblong or suborbicular, brown, reddish-brown or yellowish-brown, with a green central band, ending in an aculeate mucro or ariste, more rarely acuminate, acute or obtuse. Unisexual utricles present or, sporadically, absent, linear, lanceolate, oblong, ovate or elliptic in outline, straight or, more rarely, curvate or arcuate, straw-coloured, yellowish-brown or brown when mature, smooth to densely aculeate, especially in the upper tiers, with numerous, very prominent veins across the entire surface, rarely faintly veined, suberect to patent, gradually attenuate or abruptly contracted into an smooth to aculeate, bidentate, slightly bifid, split, truncate or irregular beak; rachilla usually reaching the apex or protruding from it, more rarely rudimentary to reaching the half of the utricle length. Bisexual utricles wide and obliquely truncate at the apex, rarely absent. Achenes ovate-trigonous or, more frequently ellipsoid-trigonous to oblong-trigonous, straw-coloured, yellowish-brown to dark-brown when mature, tipped by an obtusely trigonous to subterete, neck-like or, more rarely, pyramidal, persistent style base.

**Notes**

In addition to the names (accepted and synonyms) contained in the present monograph, the following names have been included under *Schoenoxiphium*, although they are now considered as synonymous of accepted names included in other sections of the genus *Carex* (*Global Carex Group, 2015*; *Roalson et al., 2020*; *Villaverde et al., 2020*):

*Schoenoxiphium clarkeanum* Kük. (accepted name *Carex bonatiana* (Kük.) Ivanova, former *Kobresia bonatiana* Kük.).

*Schoenoxiphium fragile* (C.B. Clarke) C.B. Clarke (accepted name *C. bonatiana* (Kük.) Ivanova).

*Schoenoxiphium hissaricum* Pissjauk. (accepted name *Carex pseudolaxa* (C.B. Clarke) O. Yano, former *Kobresia pseudolaxa* Pissauk.).

*Schoenoxiphium kobresioideum* Kuk. (accepted name *Carex kobresioidea* (Kük.) S.R. Zhang, former *Kobresia kobresioidea* (Kük.) J. Kern).

*Schoenoxiphium kuekenthalianum* (Hand.-Mazz.) Ivanova N.A. (accepted name *Carex liangshenensis* S.R.Zhang, former *Kobresia kuekenthaliana*).

*Schoenoxiphium laxum* (Nees) Ivanova (accepted name *Carex pseudolaxa* (C.B. Clarke) O. Yano, former *Kobresia pseudolaxa* Pissauk.)

## KEY

1. Utricles papyraceous, not or very faintly veined, broadly ellipsoid; female glumes mostly scarious, much wider than the utricles and concealing them; inflorescence reduced to a dense, androgynous spike. . . . . . . . . . . . . . . . . . . . . . . . . . . . . . . . . . . . . . . ***C. acocksii***
- Utricles not papyraceous, prominently veined, linear, lanceolate, ovate, oblong or elliptical in outline; female glumes colored, more rarely hyaline, usually narrower than the utricles, never concealing them; inflorescence paniculiform, rarely reduced to a solitary, lax, androgynous spike, but then utricles narrowly linear or female glumes brown, narrower than utricles . . . . . . . . . . . . . . . . . . . . . . . . . . . . . . . . . . . . . . . . . . . . . . . 2

2. Rachilla rudimentary or reaching up to ½(⅔) of the length of the utricle
. . . . . . . . . . . . . . . . . . . . . . . . . . . . . . . . . . . . . . . . . . . . . . . . . . . . . . . ***C. dregeana***
- Rachilla well developed, reaching the apex of the utricle or protruding from it. . . . . . 3

3. Inflorescence very dense, rarely somewhat lax, multispiculate, broadly ovoid to suborbicular, occupying up to the upper ⅕ of the of the culm, rarely up to ⅓, but then composed by 2–3 suberect to erect, ovoid to suborbicular parts, one terminal and 1–2 lateral, long-pedunculate, distant parts. . . . . . . . . . . . . . . . . . . . . . . ***C. multispiculata***
- Inflorescence not as above. . . . . . . . . . . . . . . . . . . . . . . . . . . . . . . . . . . . . . . . 4

4. Leaves plicate when fresh, light green; young inflorescence compressed and partially concealed by two lower, distichous bracts; utricles narrowly linear in outline
. . . . . . . . . . . . . . . . . . . . . . . . . . . . . . . . . . . . . . . . . . . . . . . .***C. schweickerdtii***
- Leaves not plicate when fresh, if so (some individuals of *C. esenbeckiana*), then glaucous and utricles widely ovoid to ellipsoid; young inflorescence never compressed nor concealed by distichous bracts . . . . . . . . . . . . . . . . . . . . . . . . . . . . . . . . . . . . . . . . . . . . . 5

5. Partial inflorescences nodding (except the uppermost); unisexual utricles (5.3)5.5–8 (8.4) mm long; linear, narrowly lanceolate or narrowly elliptic . . . . . . . . . . . . . . . . . . . 6
- Partial inflorescences never nodding (except rarely the lower one), if so (*C. ludwigii*), then unisexual utricles (2.6)2.8–3.7(4.9) mm long, broadly ellipsoid . . . . . . . . . . . . . . . . . 7

6. Partial inflorescences very dense, more or less ovoid, with peduncles much longer than the fertile part . . . . . . . . . . . . . . . . . . . . . . . . . . . . . . . . . . . . . . . . . .***C. pseudorufa***
- Partial inflorescences more or less lax, linear to oblong, with peduncles usually shorter than the fertile part . . . . . . . . . . . . . . . . . . . . . . . . . . . . . . . . . . . . . . . . .***C. lancea***

7. Utricles linear, lanceolate, narrowly fusiform or oblong, gradually attenuate into the beak . . . . . . . . . . . . . . . . . . . . . . . . . . . . . . . . . . . . . . . . . . . . . . . . . . . . . . . . . 8
- Utricles broadly ovate or widely elliptic, abruptly contracted, very rarely attenuate (occasionally in *C. spartea*), into the beak . . . . . . . . . . . . . . . . . . . . . . . . . . . . . 16

8. Leaves canaliculate, rarely flat, up to 1.5(3.2) mm wide . . . . . . . . . . . . . . . . . . . . . . 9
- Leaves flat or slightly carinate, (2)2.5–9(10) mm wide. . . . . . . . . . . . . . . . . . . . . . 11

9. Utricles (5)6–7.2 mm long, densely hispid, at least in the upper half, more or less suberect at maturity: rachilla protruding (0.5)1–1.5 mm from the apex of the utricle. . . . . . . . . . . . . . . . . . . . . . . . . . . . . . . . . . . . . . . . . . . . . . . . . . . . . . . . . . . . .**C. basutorum**
- Utricles up to 4.1 mm long, if longer, then glabrous or sparsely aculeate at the apex and patent to erect-patent at maturity; rachilla protruding up to (0.3)0.5 mm from the apex of the utricle. . . . . . . . . . . . . . . . . . . . . . . . . . . . . . . . . . . . . . . . . . . . . . . . . . . . . . . . . . 10

10. Utricles and utriculiform cladoprophylls erect to suberect, narrowly ellipsoid; leaf apex strongly curved to curled; female glumes usually dark reddish-brown . . . . . .**C. distincta**
- Utricles and utriculiform prophylls patent, erect-patent or somewhat reflexed, linear to lanceolate in outline, leaf apex straight to little curved, exceptionally somewhat curled; female glumes pale to dark brown . . . . . . . . . . . . . . . . . . . . . . . . . . . . . . . . .**C. killickii**

11. Leaves coriaceous, very scabrid and rough on the margins, with the apex usually and clearly curled; lowest bract sheathing, utricles up to 4.2(5) mm long. . . . . . . . .**C. burkei**
- Leaves not or scarcely coriaceous, not or slightly scabrid, with the apex straight, curved or, more rarely, curled; lowest bract sheathing or not; utricles (3.8)4.3–8(10.1) mm long . . . . . . . . . . . . . . . . . . . . . . . . . . . . . . . . . . . . . . . . . . . . . . . . . . . . . . . . . . . . . 12

12. Lowest bract sheathing. . . . . . . . . . . . . . . . . . . . . . . . . . . . . . . . . . . . . . . . . . . . . . . 13
- Lowest bract not sheathing . . . . . . . . . . . . . . . . . . . . . . . . . . . . . . . . . . . . . . . . . . . . . 14

13. Lowermost partial inflorescences long pedunculate, dense; female glumes yellowish-brown to brown . . . . . . . . . . . . . . . . . . . . . . . . . . . . . . . . . . **C. kukkoneniana**
- Partial inflorescences subsessile to very shortly pedunculate, lax; female glumes hyaline . . . . . . . . . . . . . . . . . . . . . . . . . . . . . . . . . . . . . . . . . . . . . . . . . **C. gordon-grayae**

14. Basal sheaths usually with lamina; lowest partial inflorescences pedunculate; utricles usually arcuate . . . . . . . . . . . . . . . . . . . . . . . . . . . . . . . . . . . . . . . . . . . . . **C. chermezonii**
- Basal sheaths bladeless; partial inflorescences sessile; utricles usually straight . . . . . . 15

15. Densely caespitose (rhizome with short internodes); flowering culms (7.5)10–30(42) cm long; inflorescence ovoid or shortly oblong, dense; utricles (3.8)4.3–5(5.8) mm long, suberect in fruit . . . . . . . . . . . . . . . . . . . . . . . . . . . . . . . . . . . . . . . . . . . . . **C. capensis**
- Not or loosely caespitose (rhizome with more or less long internodes); flowering culms (25)40–70(96) cm long; inflorescence usually long oblong, lax at the maturity; utricles (6.1) 6.4–8.5(10.1) mm long, patent in fruit . . . . . . . . . . . . . . . . . . . . . . . . . . **C. sciocapensis**

16. Leaves narrowly canaliculate or involute, up to 0.7(1.5) mm wide; partial inflorescences very lax, with up to (2)3 female flowers. . . . . . . . . . . . . . . . . . . . . . . . . . .**C. perdensa**
- Leaves flat or scarcely carinate, more than (1)1.5 mm wide, if narrower, then leaves flat in cross section; at least some partial inflorescence with more than (5)8 female flowers . . . . . . . . . . . . . . . . . . . . . . . . . . . . . . . . . . . . . . . . . . . . . . . . . . . . . . . . . . . . 17

17. Utricles usually curved, rarely straight; lowermost partial inflorescence nodding; rhizome stout. . . . . . . . . . . . . . . . . . . . . . . . . . . . . . . . . . . . . . . . . . . . . . . . . . . **C. ludwigii**

- Utricles straight, exceptionally some of them slightly curved; lowermost partial inflorescence never nodding; rhizome slender to moderately stout . . . . . . . . . . . . . . . 18

18. Utricles (4.3)4.5–5.5(6) mm long, ending in a beak (0.7)1.1–2.2(2.3) mm long, if smaller, then plants distinctly glaucous, lowermost partial inflorescence arising from close to the culm base, and plants growing in shady places . . . . . . . . . . . . . . . . . . . . . . . . 19
- Utricles (2.6)2.8–4(4.9) mm long, ending in a beak (0.2)0.5–0.8(1) mm long, plants light green to scarcely glaucous, lowermost partial inflorescence arising above of the upper ½(⅓) of the culm, and plants growing in sunny places . . . . . . . . . . . . . . . . . . . . . . . 20

19. Densely caespitose; basal sheaths usually very fibrous; culms (1)1.1–1.5(1.9) mm wide at the middle part; lowermost partial inflorescence located usually close to the base of the culm; peduncles of the partial inflorescences included in the sheaths or slightly protruding from it; bisexual utricles rarely present; unisexual utricles with a smooth beak; style base distinctly neck-like . . . . . . . . . . . . . . . . . . . . . . . . . . . . . . . . *C. esenbeckiana*
- Loosely caespitose; basal sheaths entire to somewhat fibrous; culms 1.7–2 mm wide at the middle part; lowermost partial inflorescence located above the ½(⅔) of the length of the culms; peduncles of the partial inflorescences much protruding from the sheaths; bisexual utricles usually present; unisexual utricles with an aculeate beak; style base never neck-like . . . . . . . . . . . . . . . . . . . . . . . . . . . . . . . . . . . . . . . . . . . . . . . *C. badilloi*

20. Basal sheaths entire, somewhat broken or slightly fibrous; utriculiform cladoprophylls usually present; most unisexual utricles more or less densely hispid at least in the beak . . . . . . . . . . . . . . . . . . . . . . . . . . . . . . . . . . . . . . . . . . . . . . . . . . *C. bolusii*
- Basal sheaths usually very fibrous; utriculiform cladoprophylls rarely present; unisexual utricles smooth, very rarely with a few, very disperse, minute prickles. . . . . . . *C. spartea*

**Carex acocksii** C. Archer, S. African J. Bot. 63: 342, 1998 [1997].

Type. South Africa. Northern Cape, Calvinia district, in vicinity of FM tower on top of Hantamsberg, Van Rhynshoek farm, 1,580 m, 03-X-1987, *Reid 1337* (holotype: PRE-0762273-0 digital image!; iso-: BM-000611185 digital image!, GENT-0000090034770 digital image!, K-001044967 digital image!, MO-193695 digital image!, NBG-0200446-0 digital image!, P-00199375 digital image!, S-06-20520 digital image!, S-G-10688 digital image!, TCD-0000356 digital image!).

Rhizome loosely cespitose, slender, brown. Flowering culms 14–40(47) cm long, bulbiform at the base, erect, terete and prominently ribbed, smooth, leafy only at the base, not reaching the third of the length, 0.6–1.2(1.3) mm wide at the middle. Leaves (0.2)0.6–1.1 (1.4) mm wide, much shorter than the inflorescence, moderately rigid, light green or somewhat glaucous, canaliculate or, more frequently, involute in cross-section, scabrous along the margins in all its length; abaxial and adaxial surfaces smooth; ligule 0.8–1.5 (2) mm long. Basal sheaths somewhat fibrous, bladeless. Lowest bract of the inflorescence glumaceous or setaceous with glumaceous base, shorter or longer than the inflorescence length, axilating an utricle, not sheathing. Inflorescence reduced to a solitary, androgynous

spike (10)15–24(27) × (3)4–6 mm, with the male part usually shorter, rarely equaling the female one. Glumiform perigynia and glumiform cladoprophylls absent. Tubular cladoprophylls absent. Utriculiform cladoprophylls absent. Male glumes (2.3)2.7–4.5(6) × (1.1)1.8–2.7(3) mm, ovate, obovate or elliptical, brown in the upper half and straw-coloured to hyaline in the lower one, acute to acuminate, exceptionally obtuse. Female glumes (4)4.8–6.2(6.7) × (2.1)2.6–4.3(5.2) mm, longer and much wider than utricles, hiding them, widely obovate to suborbicular, single mid-veined, pale to golden-brown with hyaline margins at the upper parts and hyaline to yellowish at the lower ones, usually ending in a mucro up to 2(2.8) mm long, very rarely obtuse to roundate at the apex. Utricles 3.4–4(4.6) × (1.4)1.5–1.9(2.2) mm, unisexual, widely ellipsoid, stipitate, straight, papyraceous, hyaline to pale or golden-brown, translucid, glabrous, smooth, with only a few veins or inconspicuous, suberect and erostrate; rachilla reaching up to the half of the utricle. Achenes 3.2–3.9 × 1.4–2 mm, elliptic, yellowish to dark brown when mature, tipped by a widely pyramidal (mitrate), persistent style base.

## Distribution
Endemic to the Northern Cape province in South Africa [27 CPP]. Figure 6A.

## Habitat
Open grounds and under shrub on dolerite outcrops in Succulent Karoo Biome (Western Mountain Karoo: Hantam Karoo and Roggeveld Karoo); 1,450–1,600 m.

## Etymology
Named after John Phillip Harison Acocks (1911–1979), a South African botanist and collector.

## Iconography
Figures 7 and 8A; *Archer & Balkwill (1997*, holotype image); *Márquez-Corro et al. (2020*, detailed photographs of herbarium specimens).

## Conservation
Previous assessments of the species performed at the national level in South Africa (*Victor, 2002*; *Raimondo et al., 2009*; *SANBI, 2020*) had consistently resulted in the category Vulnerable (VU). However, a recent reevaluation of the conservation status at global level following criteria, categories and guidelines from *IUCN (2012*, *2017)* has resulted in the category Critically Endangered (CR) (*Márquez-Corro et al., 2020*; *Márquez-Corro & Martín-Bravo, 2020*). The species is only known from two locations and appears to be severely threatened by overgrazing pressure by livestock and the possible impacts of climate change.

## Notes
This species was formerly included in section *Petraea* Lang (*Archer & Balkwill, 1997*), since it shows unispicate inflorescences, very wide female glumes and hyaline-papyraceous utricles. However, recent molecular studies (*Márquez-Corro et al., 2020*) have concluded that it should be included in section *Schoenoxiphium*, despite its deviant morphological

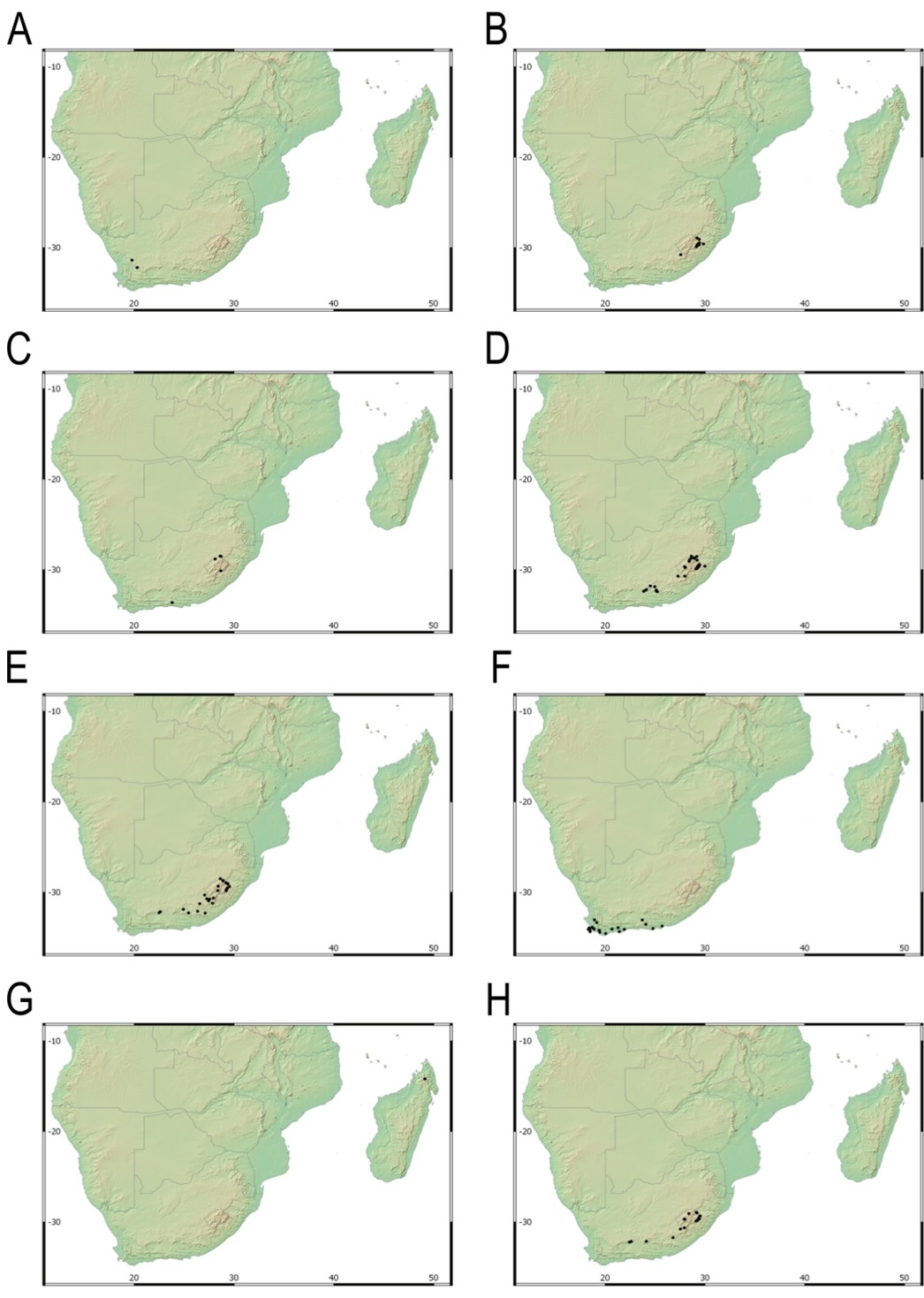

**Figure 6 Distribution map of different species of *Carex* section *Schoenoxiphium*.** (A) *C. acocksii*.
(B) *C. badilloi*. (C) *C. basutorum*. (D) *C. bolusii*. (E) *C. burkei*. (F) *C. capensis*. (G) *C. chermezonii*.
(H) *C. distincta*. Maps created with QGIS.      

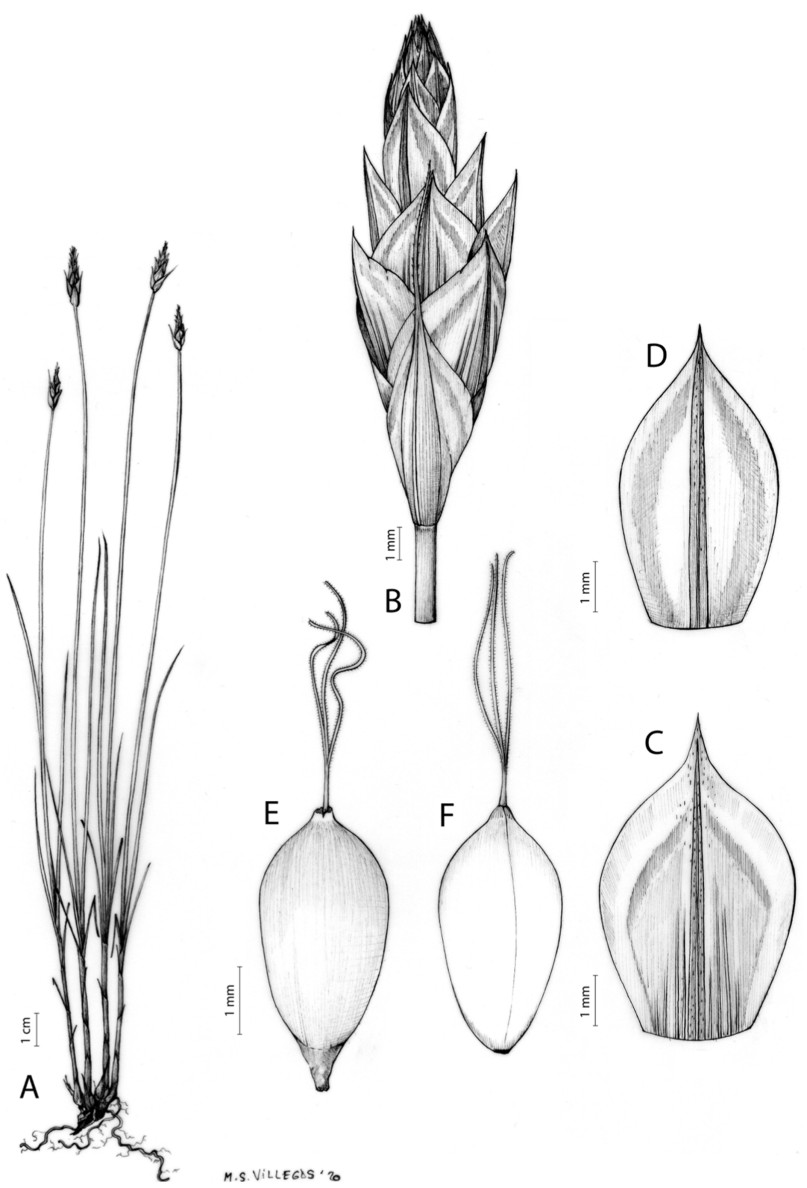

**Figure 7 Illustration of _Carex acocksii_.** (A) General aspect of the plant. (B) Inflorescence. (C) Female glume. (D) Male glume. (E) Utricle. (F) Achene. Illustration by M. Sánchez-Villegas.

features with respect to the remaining species in the section; nevertheless, its rachilla (flattened and ciliate at the margin) is quite typical of the species of this section (_Global Carex Group, 2015_; _Márquez-Corro et al., 2020_). Its position as sister group of the _C. spartea_ clade (Clade D) is not strongly supported (Fig. 4; _Márquez-Corro et al., 2020_), so further studies are needed to confirm its phylogenetic placement within this section.

**Selected references**

_Archer & Balkwill, 1997_. _Márquez-Corro et al., 2020_; _Márquez-Corro & Martín-Bravo, 2020_.

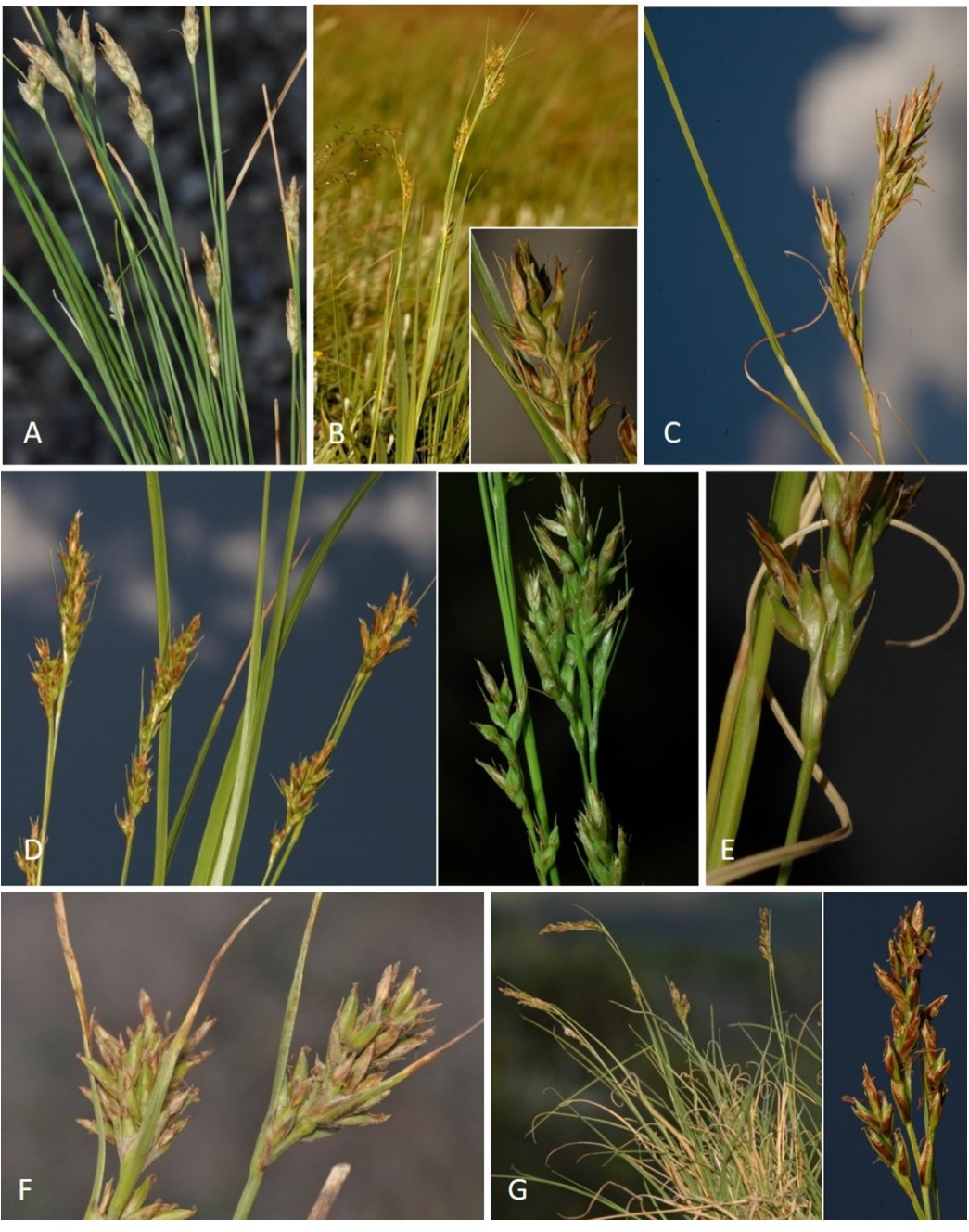

**Figure 8 Morphology of different species of *Carex* section *Schoenoxiphium*.** (A) *C. acocksii*. (B) *C. badilloi*. (C) *C. basutorum*. (D) *C. bolusii*. (E) *C. burkei*. (F) *C. capensis*. (G) *C. distincta*. Photos by M. Luceño.

**Carex badilloi** Luceño & Márquez-Corro, Phytotaxa 303(1): 36, 2017.

Type. South Africa. KwaZulu-Natal, Drakensberg mountains, Garden Castle Nature Reserve, pathway to Rhino Peak, 1,800 m, grassland, 29°44′38.70″S 29°12′21.20″E, 11-XI-2011, *S. Martín-Bravo & M. Luceño 96SMB11* (holotype: PRE!; iso-: NU-0049308!, UPOS-6576!).

*- Schoenoxiphium ludwigii* sensu Gordon-Gray (1995:168), non. Hochst. (1845:764).

Rhizome not or loosely caespitose, stout, dark-brown. Flowering culms (36)39–74(93) cm long, erect, obtusely trigonous, smooth, leafy up to the lower two-thirds of its length, 2–3.1 (3.2) mm wide at the middle. Leaves (3.7)4.5–6.5(9) mm wide, shorter than the inflorescence, moderately rigid, somewhat glaucous, slightly V-shaped in cross-section, scarcely to moderately scabrous along the edges and usually along the uppermost parts of the abaxial midrib; adaxial surface densely papillose; straight at the apex; ligule (0.8)1–3 (3.5) mm long. Basal sheaths more or less entire, lowermost bladeless and uppermost with lamina. Lowest bract of the inflorescence leaf-like, much shorter than the inflorescence, with a sheath 35–49(63) cm long. Inflorescence branching up to 3 times. Partial inflorescences 4–7(8), uppermost subsessile and overlapping, lowermost more or less distant, pedunculate, more or less erect. Glumiform perigynia and glumiform cladoprophylls absent. Tubular cladoprophylls always present, those of the base of the first order branches terete, hyaline, those of the second order branches asymmetrically hypocrateriformis, somewhat hispid in the upper parts, more or less prominently veined and open. Utriculiform cladoprophylls sometimes present. Male glumes (2.9)3–4.4(4.9) × (1)1.2–2.4(2.6) mm, ovate, yellowish-brown, with a green central band, ending in an aculeate mucro up to 0.2–1.7(3) mm long. Female glumes (3.3)3.5–4.2 × (2.1)2.3–2.8(3.5) mm, widely ovate to suborbicular, yellowish-brown, with a green central band, ending in a light green, prominent, aculeate mucro up to 3.1 mm long. Unisexual utricles (4.3)4.7–5.3(5.4) × 1.7–2 mm, ovoid to ellipsoid, straight, green to yellowish-brown when mature, hispid in the upper third or, very rarely, almost smooth, with prominent veins across the entire surface, suberect to erecto-patent; rachilla reaching the apex to protruding from it up to 0.5 mm, abruptly contracted into an aculeate, bidentate to irregular beak 1.1–1.6(1.8) mm long. Bisexual utricles wide and obliquely truncate at the apex. Achenes 3–3.3(3.5) × 1.4–1.9(2) mm, ovoid to ellipsoid, trigonous, straw-coloured to yellowish-brown when mature, tipped by a short, obtusely trigonous, persistent style base.

**Distribution**
Endemic to Drakensberg mountains in Lesotho (Qacha's Nek district) and South Africa (Eastern Cape and KwaZulu-Natal provinces) [27 CPP LES NAT]. Figure 6B.

**Habitat**
Mesophilous, open grassland on clay soils in Grassland Biome (Drakensberg Grassland, Mesic Highveld Grassland); 1,750–2,500 m.

**Etymology**
Named after Dr. Juan José González Badillo (1945-), Spanish researcher specialized in sport sciences at Universidad Pablo de Olavide (Seville, Spain).

**Iconography**
Figures 8B and 9, *Márquez-Corro et al. (2017*, analytical drawing and photographs of living specimens in the field).

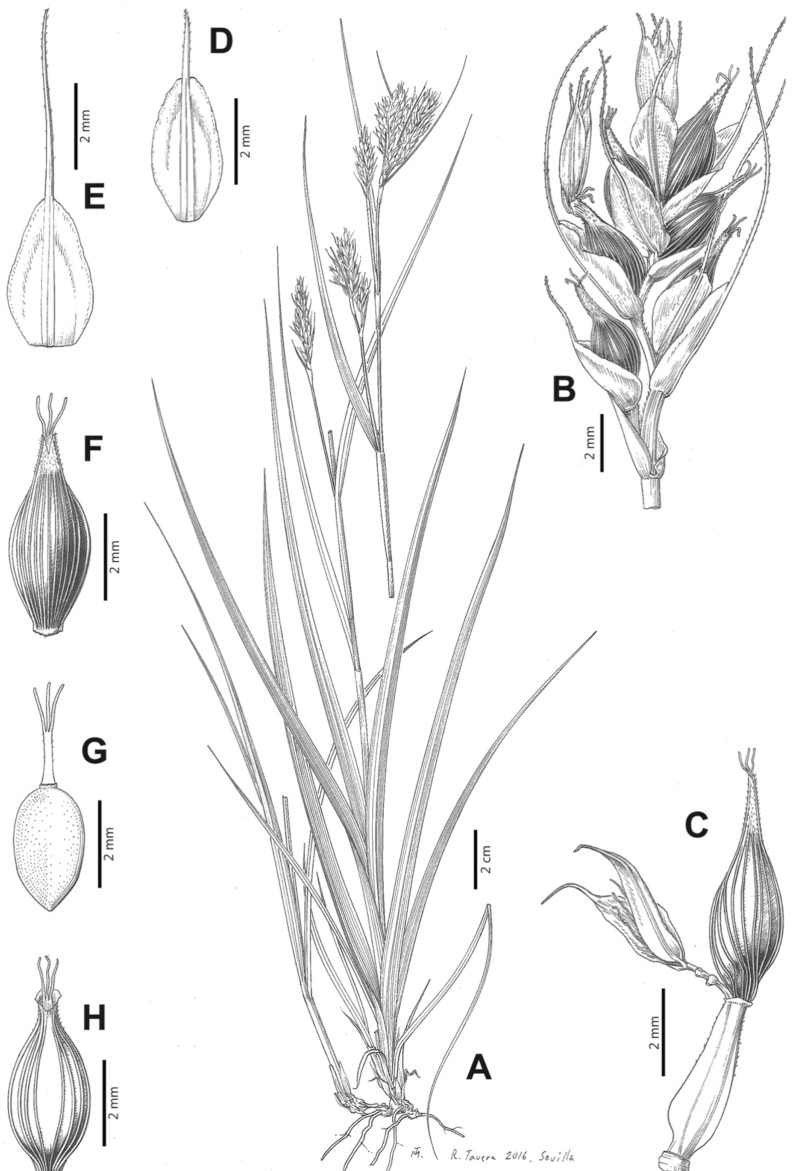

**Figure 9 Illustration of *Carex badilloi*.** (A) General aspect of the plant. (B) Second order branch of the inflorescence. (C) Utriculiform cladoprohyll. (D) Male glume. (E) Female glume. (F) Unisexual utricle. (G) Achene. (H) Bisexual utricle after removing the male spikelet. Illustration by R. Tavera.

## Conservation

Not evaluated (NE).

## Notes

This species traditionally used to be called *Schoenoxiphium ludwigii* Hochst.; however, since the type of the latter name belongs to a different species (*Schoenoxiphium rufum* Nees, currently known as *Carex ludwigii* (Hochst.) Luceño & Martín-Bravo; see *Global Carex Group, 2015*), the new species *C. badilloi* had to be described for this taxon

(*Márquez-Corro et al., 2017*). The identity of *C. badilloi* has been well-supported by morphological (*Márquez-Corro et al., 2017*; this study) and molecular (*Villaverde et al., 2017*; *Márquez-Corro et al., 2020*; this study) data.

**Selected references**
*Márquez-Corro et al. (2017)*.

**Carex basutorum** (Turrill) Luceño & Martín-Bravo, Bot. J. Linn. Soc. 179: 26, 2015.

Type. [Lesotho] Basutoland. Plateau, Leribe Mount. Flowering season: summer, XII-1912, *A. Dieterlen 948* (lectotype: K-000363525 digital image!, designated by *Kukkonen (1983)*; iso-: PRE-0107825 digital image!, P-00540800 digital image! [sic. 1913]).
≡ *Schoenoxiphium basutorum* Turrill, Bull. Misc. Inform. Kew 1914: 19, 1914 [basionym]

Rhizome caespitose, with short internodes, moderately stout, brown. Flowering culms (25)30–60(65) cm long, acutely trigonous and smooth or scarcely scabrid at the apex, leafy usually up to the upper third of its length, (0.6)0.9–1.4(1.5) mm wide at the middle. Leaves (0.3)1–1.5(2.3) mm wide, shorter, equaling or longer than the inflorescence, soft to scarcely rigid, light green, carinate or canaliculate, rarely trigonous or somewhat flat in cross-section, scabrid along the edges in all its length; abaxial surface smooth, except frequently in the midrib; adaxial surface smooth; trigonous and curled at the tip; ligule 1.5–3(3.7) mm long. Basal sheaths more or less fibrous, with lamina, but the lowest bladeless. Lowest bract of the inflorescence leaf-like, longer or, more rarely, shorter than the inflorescence, not sheathing. Inflorescence branching up to 2(3) times; partial inflorescences (3)5–8(10) overlapping or, rarely, the lowest somewhat distant; sessile to shortly pedunculate, erect to suberect. Glumiform perigynia and glumiform cladoprophylls absent. Tubular cladoprophylls present in the lowest partial inflorescence, rarely absent, hyaline. Utruliform cladoprophylls always present. Male glumes 4.2–7 × 1.5–1.9 mm, ovate to ovate-lanceolate, pale reddish-brown, with a narrow, green central band, acute, acuminate to shortly aristate. Female glumes (4.1)4.8–6.3(6.8) × 1.7–2.2(2.4) mm, ovate, pale reddish-brown, with a green central band, acute, acuminate or more frequently ending in a smooth ariste up to 1 mm. Unisexual utricles (5)6–7.2 × 0.7–1 mm, narrowly linear, shortly stipitate, straight or, very rarely, slightly arcuate, greenish, straw coloured to pale brown, densely papillose-scabriuscule at least in the upper half, with numerous prominent veins across the entire surface, erect to suberect, very gradually attenuated into an irregularly lacerate, asymmetrically truncate, bidentate or bifid beak up to 2 mm long; rachilla protruding from the apex of the utricle up to 1.5 mm. Bisexual utricles wide and obliquely truncate. Achenes (3.5)3.8–4.2 × 0.6–1 mm, oblong-trigonous, yellowish-brown, tipped by a trigonous, acute, persistent style base.

**Distribution**
Endemic to Drakensberg and Winterhoek mountains in Lesotho (Leribe and Qacha's Nek districts) and South Africa (Eastern Cape and Free State provinces) [27 CPP LES OFS].
Figure 6C.

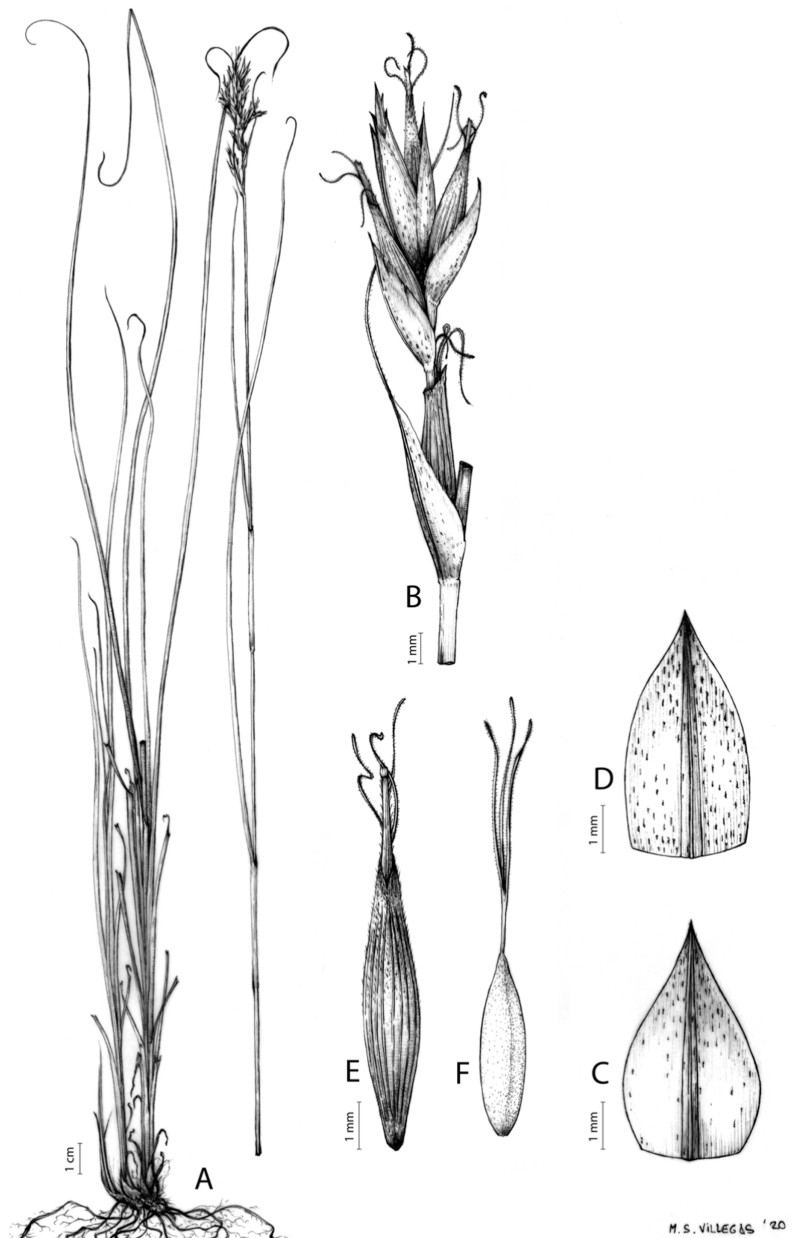

**Figure 10 Illustration of *Carex basutorum*.** (A) General aspect of the base of the plant and one fertile culm showing the inflorescence. (B) Second order branch showing an utriculiform cladoprophyll. (C) Male glume. (D) Female glume. (E) Unisexual utricle. (F) Achene. Illustration by M. Sánchez-Villegas.

## Habitat

Grassland and scrub clearing in Grassland Biome (Drakensberg Grassland and Mesic Highveld Grassland); 1,500–2,100 m.

## Etymology

The epithet honours Basotho people, inhabitants of the country currently named Lesotho.

## Iconography

Figures 8C and 10.

## Conservation

Not considered of conservation concern at the national level in South Africa, and thus categorized as Least Concern (LC) (*SANBI, 2020*; sub *Schoenoxiphium basutorum*).

**Carex bolusii** C.B.Clarke, Fl. Cap. 7: 304 (1898)

Type. South Africa. Eastern Cape, Graaff Reinet area, Compassberg, "In saxoso graminosis in monte Cave", 4,500 ft., XII-1871, *H. Bolus 1974* (lectotype: K-000693810 digital image!, here designated; isolecto-: BOL-70348 digital image!). Syntype: Natal, *Buchanan 328* (K-000693809 digital image!).

=*Carex parvirufa* Luceño & Márq.-Corro, Phytotaxa 303(1): 41, 2017.
Type. Lesotho, Leribe district, Pitseng, left side of the road in the ascent to the Mafika Lisiu Pass summit, 3,101 m, grasslands in the summit, 29°03′58.30″S 28°24′27.10″E, 14 January 2014, *T. Villaverde et al. 28TVH14* (holotype: PRE!; iso-: NU-0049310!, UPOS-8508!).

= *Schoenoxiphium bracteosum* Kukkonen, Notes Royal Botanic Garden Edinburgh 43: 365 (1986); non *Carex bracteosa* (Rchb.) Kunze ex Kunth (1837).
Type: [South Africa] Cape, Barkly East distr., Ben McDhi, 9,000 ft, in grass tussocks on hillside, 5-II-1983, *Hilliard & Burtt 16471* (holotype: H-1486367 digital image!; iso-: E-00200231 digital image!, GENT-0000090032912 digital image!, NU-0015688 digital image!).

Rhizome caespitose, with short internodes moderately stout, light-brown to dark-brown. Flowering culms (4)12–67(95) cm long, erect, obtusely trigonous, smooth, occasionally with any minute prickle to the apex, leafy from one third to nearly half of its length, (0.3)0.8–1.2(1.5) mm wide at the middle. Leaves (0.5)1.7–3.8(5.5) mm wide, shorter, rarely longer (in dwarf plants) than the inflorescence, slightly to moderately rigid, straight at the apex, light-green to somewhat glaucous, flat to ± V-shaped in cross-section, slightly to moderately scabrous along the edges and usually along the abaxial midrib; adaxial surface usually finely papillose; ligule (0.5)0.8–3(4) mm long. Basal sheaths with lamina or, more rarely, the lowermost bladeless, entire, somewhat broken or slightly fibrous. Lowest bract of the inflorescence leaf-like, from half of the inflorescence length to equaling it, with a sheath (9)17–51(73) mm long. Inflorescence up to the upper ½ of the length of the culm, branching up to 3(4) times; partial inflorescences (5)6–8(9), erect, the uppermost sessile, overlapping, the lowermost usually distant, long-pedunculate. Glumiform perigynia and glumiform cladoprophylls absent. Tubular cladoprophylls always present. Utriculiform cladoprophylls usually present, similar to bisexual utricles. Male glumes 2.2–3(3.3) × (1)1.1–1.8(2) mm, ovate to obovate, yellowish-brown to brown, with a green central band, ending in an aculeate mucro up to 2 mm long. Female glumes 2–2.7(3.1) × (1.1)1.3–2(2.1) mm, ovate, yellowish-brown, with a green central band, ending in an aculeate mucro up to 2.8(3.1) mm long, arising from the back, a little below

the apex of the glume. Unisexual utricles (2)2.5–3.6(4) × (0.6)1–1.4(1.6) mm, ovoid to ellipsoid, stipitate, straight or, more rarely, slightly arcuate, straw-coloured to yellowish-brown when mature, dispersely aculeate to hispid in the upper third, with conspicuous prominent veins over the entire surface, suberect to erecto-patent, abruptly contracted into an aculeate, bidentate to irregular beak (0.3)0.5–1 mm long; rachilla reaching the apex to protruding from it up to 0.3(0.4) mm. Bisexual utricles usually present, widely and obliquely truncate at the apex. Achenes (1.2)1.4–2.2(2.5) × (0.7)0.8–1.2 (1.5) mm, ellipsoid-trigonous, straw-coloured to yellowish-brown when mature, tipped by a short, obtusely trigonous, usually asymmetric, persistent style base.

## Distribution
Endemic to the Drakensberg and Central West of Eastern Cape mountains. Lesotho (Butha Buthe, Maseru, Leribe, Qacha's Nek and Thaba Tseka districts) and South Africa (Eastern Cape, Free State and KwaZulu-Natal). [27 CPP LES NAT OFS]. Figure 6D.

## Habitat
Edges of streams, grasslands, damp meadows and open bushy places (Grassland Biome: Drakensberg Grassland); (1,200)1,650–3,150 m.

## Etymology
Named after Harry Bolus (1834–1911), South African botanist and philanthropist who founded the Bolus Herbarium (BOL) in 1865.

## Iconography
Figures 8D and 11, *Márquez-Corro et al. (2017*, analytical drawing and photographs of living specimens in the field).

## Conservation
Not evaluated (NE).

## Notes
The name *C. bolusii* C.B.Clarke has been considered synonymous with *C. schimperiana* Boeckeler (a synonym of *C. spartea* Wahlenb.; *Global Carex Group, 2015*). However, the type materials indicated by Clarke (*Bolus 1974*, K-000693810 and *Buchanan 328*, K-000693809) contain specimens of *C. parvirufa* Luceño & Márq.-Corro, a species recently described (*Márquez-Corro et al., 2017*); so, by the principle of priority (*ICN et al., 2018*), the correct name of these plants should be the earlier *C. bolusii* C.B.Clarke. Kukkonen (in label) designated as lectotype the voucher *Bolus 1974* and left as residual syntype the specimen *Buchanan 328*, but such typifications were never effectively published. For this reason, we have selected here as lectotype the voucher *Bolus 1974*, and as syntype the voucher *Buchanan 328*. Moreover, after a thorough examination of the type materials from *Schoenoxiphium bracteosum* Kukkonen (E, GENT, NU) and a detailed reading of the protologue, we believe that the binomen must be considered as synonym of *C. bolusii* C.B.Clarke, contrary to a previous publication (*Global Carex Group, 2015*), in which it was synonymized under *C. schimperiana* Boeckeler (a synonym of *C. spartea*)

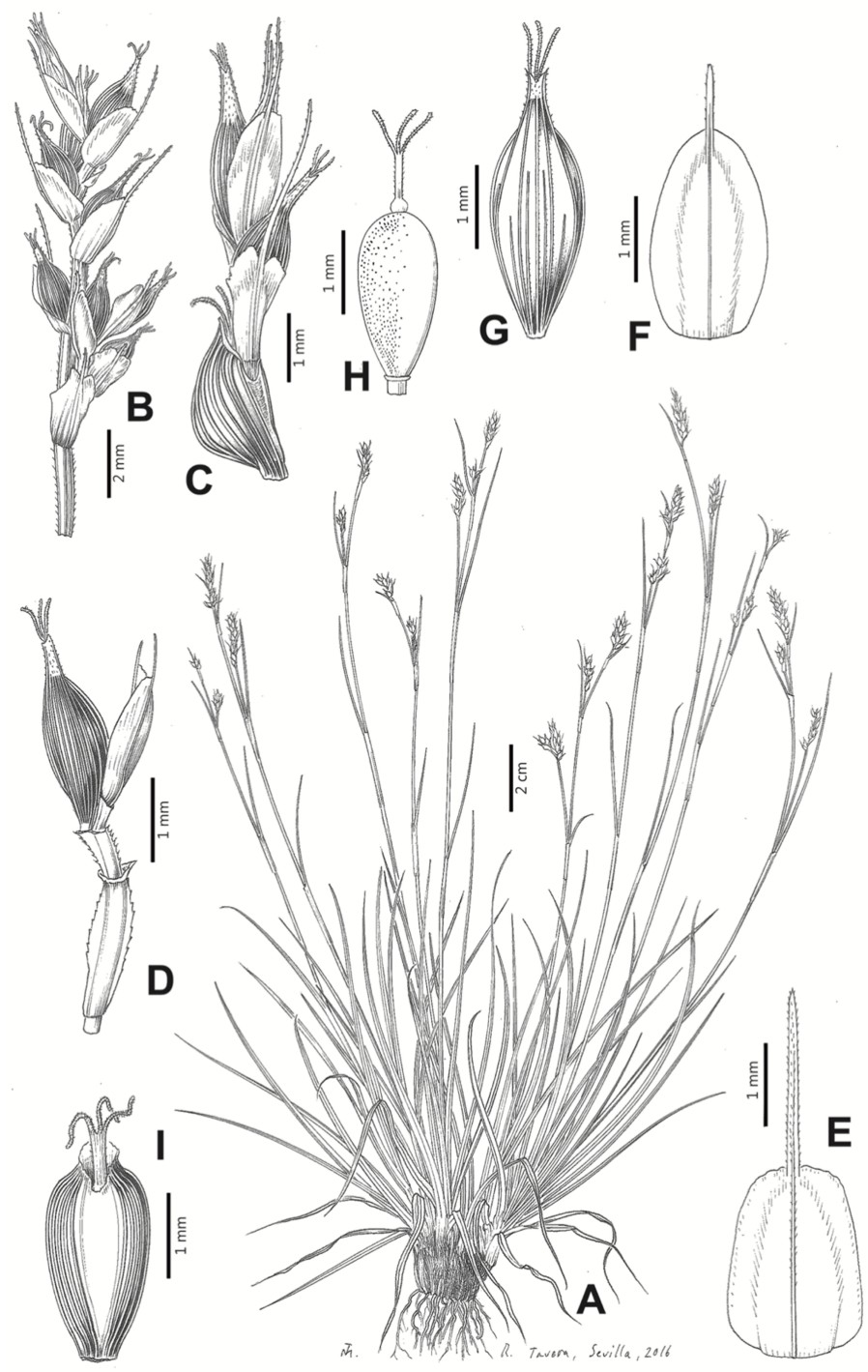

**Figure 11 Illustration of *Carex bolusii*.** (A) General aspect of the plant showing several fertile culms. (B) Second order branch of the inflorescence; (C–D) Utriculiform cladoprohylls. (E) Female glume. (F) Male glume. (G) Unisexual utricle. (H) Achene. (I) Bisexual utricle after removing the male spikelet. Illustration by R. Tavera.

circumscription. This is based on the slightly fibrous basal sheaths of the type materials, as typically occurs in *C. bolusii*, and not completely fibrous, as shared by the *C. spartea* specimens. Moreover, the leaf and culm width are similar to the measurements observed in *C. bolusii*.

**Selected references**
*Márquez-Corro et al. (2017)*.

**Carex burkei** (C.B. Clarke) Luceño & Martín-Bravo, Bot. J. Linn. Soc. 179: 26, 2015.

≡ *Schoenoxiphium burkei* C.B.Clarke, J. Linn. Soc. Bot. 20: 386, 1883 [basionym] Type: South Africa. Eastern Cape, Cradock Div. [Inxuba Yethemba Local Municipality], *Burke 211* (lectotype: K-000693801 digital image!, designated by *Gordon-Gray (1995)*).

Rhizome loosely caespitose to caespitose, with short internodes, scarcely to moderately stout, dark-brown. Flowering culms (29)40–80(93) cm long, acutely trigonous, more or less scabrid to the apex, leafy from the lower one-third to two-thirds of its length, (1)1.3–2.3(3) mm wide at the middle. Leaves (1.5)2.5–4.5(5.5) mm wide, shorter than the inflorescence, coriaceous, light green to somewhat glaucous, flat to, more rarely, carinate in cross-section, strongly scabrous along the edges and usually also along the abaxial midrib; adaxial surface papillose on the veins, especially in the upper half; tip usually very curled, ligule 0.5–4(7) mm long. Basal sheaths entire, with lamina. Lowest bract leaf-like, shorter than, rarely equaling, the inflorescence, with a sheath (19)30–60(80) mm long. Inflorescence branching up to 4 times; partial inflorescences (3)7–10(13), uppermost overlapping, sessile, lowermost distant, with scabrid peduncles up to 11 cm long, suberect. Glumiform perigynia and glumiform cladoprophylls absent. Tubular cladoprophylls present at the base of some peduncles, hyaline-brownish. Utriculiform cladoprophylls always present, usually numerous. Male glumes 3–4.5(5.1) × 1.3–2.2(2.5) mm, ovate to obovate, yellowish-brown with a green central band and, frequently, with a wide hyaline margins, acute or ending in an aculeate mucro or arista up to 0.8 mm long. Female glumes (3)3.2–4(4.1) × 1.6–2.3(2.6) mm, ovate, obovate or elliptic, with scarious margin, yellowish-brown to light-brown with a green central band, acute or with an mucro or ariste up to 1.5(2) mm long. Unisexual utricles (3.3)4.1–4.5(5) × (0.9)1.1–1.3 mm, narrowly ellipsoid or, more rarely, narrowly suboblong, stipitate, straight, green to yellowish-brown when mature, more or less hispid in the upper half, rarely subglabrous, with numerous prominent veins across the entire surface, suberect, gradually attenuate into a bidentate, obliquely truncate or irregularly lacerate beak up to 1 mm long; rachilla reaching the apex of the utricle or, more rarely, protruding from the mouth up to 0.5 mm, sometimes bearing some small, sterile glumes at the tip. Bisexual utricles ending in a beak wide and obliquely truncate. Achenes (2.3)2.5–3.1 × 0.8–1 mm, narrowly ellipsoid-trigonous to oblong-trigonous, yellowish-brown when mature, tipped by a very short and wide, obtusely trigonous, persistent, style base.

## Distribution
Endemic to Lesotho (Leribe, Mokhotlong, Qacha's Nek and Thaba Tseka districts) and South Africa (Eastern Cape, KwaZulu-Natal, Free State and Western Cape provinces). [27 CPP LES NAT OFS]. Figure 6E.

## Habitat
Grassland, clearing of shrubland, usually in dry places, but sometimes in temporary flooded meadows and edges of streams (Grassland Biome: Drakensberg Grassland); 1,400–2,900 m.

## Etymology
Named after Joseph Burke (1834–1911), British explorer who collected animals and plants for Edward Smith-Standley (Lord Derby) in South Africa between 1840 and 1842.

## Iconography
Figures 8E and 12.

## Conservation
Not evaluated (NE).

**Carex capensis** Thunb., Prodr. Pl. Cap.: 14, 1794.

Type. Caput Bonae Spei (lectotype UPS-21777, digital image!, specimen on the left, here designated)
≡ *Schoenoxiphium thunbergii* Nees, Linnaea 9: 305, 1834.
≡ *Archaeocarex thunbergii* (Nees) Pissjauk., Bot. Mater. Gerb. Bot. Inst. Komarova Akad. Nauk S.S.S.R. 12: 83, 1950.

= *Schoenoxiphium ecklonii* Nees, Linnaea 10: 200, 1835 [1836].
Type: [South Africa] 'Adowhúgel, 3' et., 'Distr. Uitenhagen,' Oct. 1829, Ecklon & Zeyher 909 (lectotype: SAM, designated by *Kukkonen (1983)*; isolecto-: S; see notes below)
≡ *Archaeocarex ecklonii* (Nees) Pissjauk., Bot. Mater. Gerb. Bot. Inst. Komarova Akad. Nauk S.S.S.R. 12: 83, 1950.
≡ *Kobresia ecklonii* (Nees) T.Koyama, J. Fac. Sci. Univ. Tokyo, Sect. 3, Bot. 8: 80, 1961.

= *Carex bisexualis* C.B.Clarke in Harvey & auct. suc. (eds.), Fl. Cap. 7: 302, 1898.
Type: [South Africa] In Caput Bonae Spei, *Ecklon 853* (lectotype: C-10000624 digital image!, here designated) Syntype: Lions rump, near Cape Town, Pappe (K-000693803 digital image!)

= *Schoenoxiphium ecklonii* var. *unisexuale* Kük. in Engler, Pflanzenr. 38(IV, 20): 33, 1909.
Type: [South Africa] In Caput Bonae Spei, *Ecklon 853* (lectotype: C-10000624 digital image!, here designated)

= *Carex zeyheri* C.B.Clarke in Harvey & auct. suc. (eds.), Fl. Cap. 7: 303, 1898.
Type: Swellendam Div., on mountains along the lower part of Zonder Einde river, 500–2,000 ft., *Zehyer 4441* (lectotype: K-000693804 digital image!, here designated,

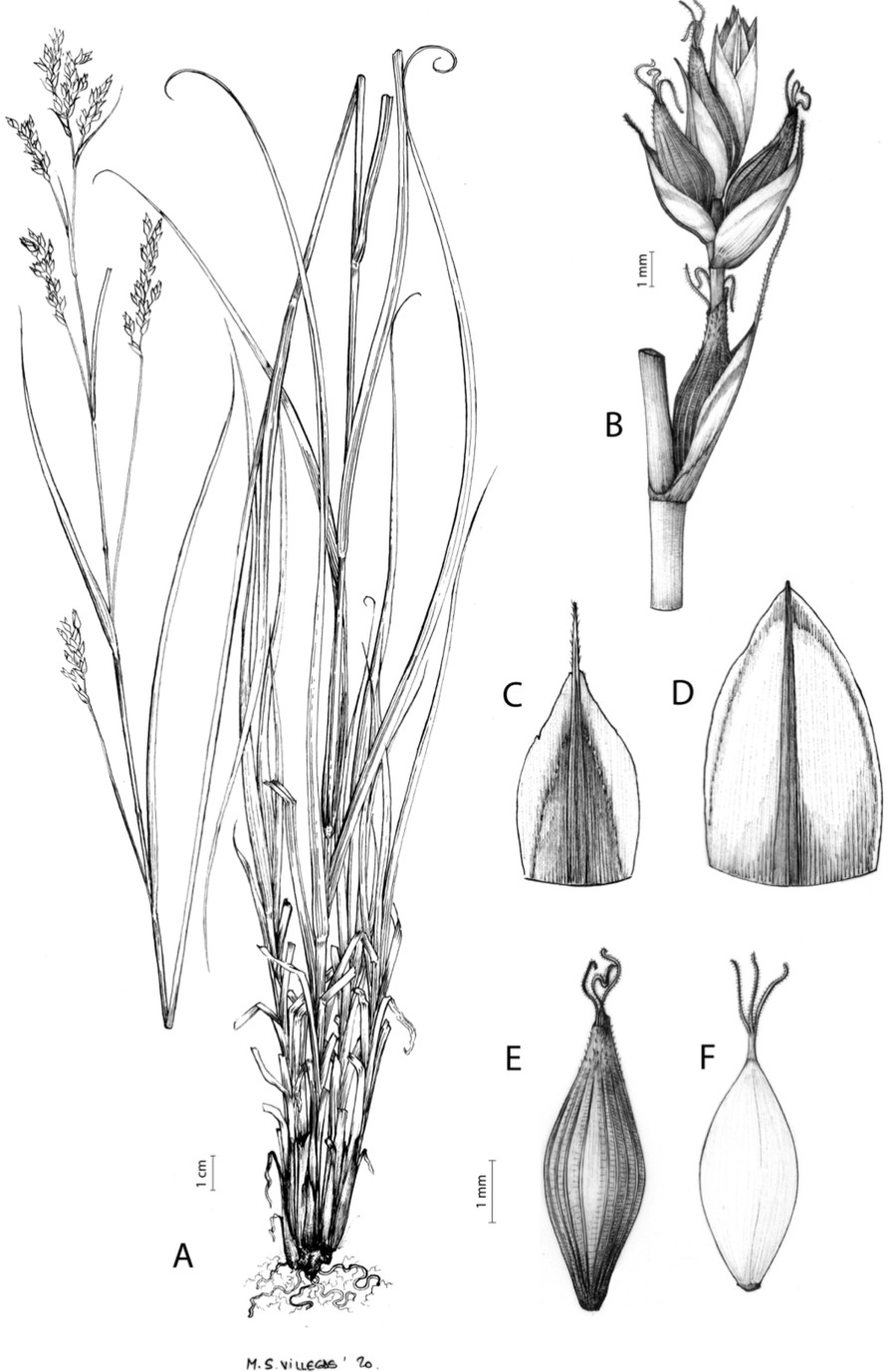

**Figure 12 Illustration of *Carex burkei*.** (A) General aspect of the plant. (B) Utriculiform cladoprophyll. (C) Female glume. (D) Male glume. (E) Unisexual utricle. (F) Achene. Illustration by M. Sánchez-Villegas.

isolecto-: FI-001040 digital image!, LE-000434 digital image!, LE-00010433 digital image!, P-00540798 digital image!, P-540799 digital image!, E-00286073 digital image!, S-10-1167).

Rhizome lax to densely caespitose, with short internodes, moderately stout, dark-brown. Flowering culms (5.5)12–30(42) cm long, erect, obtusely trigonous, smooth, leafy up to the lower third of its length, (0.7)0.8–1.5 mm wide at the middle. Leaves (1.5)2–3(3.8) mm wide, shorter than the inflorescence, somewhat rigid, light green to glaucous, flat to carinate in cross-section, scabrous along the edges and sometimes also along the abaxial midrib; adaxial surface smooth; straight to curled at the apex; ligule (0.5)0.7–1.6(2.5) mm long. Basal sheaths more or less fibrous, bladeless. Lowest bract of the inflorescence (and frequently the next one) leaf-like, longer than the inflorescence, not sheathing. Inflorescence ovoid to shortly oblong, branching up to 3 times; partial inflorescences 4–6 (8), overlapping, rarely the lowest distant, sessile. Glumiform perigynia and glumiform cladoprophylls absent. Tubular cladoprophylls always present at the base of the partial inflorescences, hyaline. Utriculiform cladoprophylls usually present in lower parts of developed inflorescences. Male glumes (3)3.5–4(4.6) × 1.3–2.2 mm, ovate-lanceolate to ovate, occasionally elliptic, light to yellowish-brown with a green central band, acute, rarely obtuse or very shortly acuminate. Female glumes (3)3.5–4.5(4.9) × 1.6–2.3(2.5) mm, ovate to ovate-lanceolate, with scarious margin, yellowish to dark brown with a green central band, acute to shortly acuminate. Unisexual utricles (3.5)4–5.1(5.8) × 1–1.4(1.5) mm, oblong to narrowly ellipsoid to ovoid, long-stipitate, straight, straw-coloured to yellowish-brown when mature, more or less hispid in the upper third, with numerous prominent veins across the entire surface, suberect, gradually attenuate into a bidentate or slightly bifid beak up to 1 mm long; rachilla reaching the apex of the utricle or, more rarely, protruding from the mouth and bearing some small, sterile glumes at the tip, but then wide and obliquely mouthed. Bisexual utricles widely and obliquely truncate at the apex. Achenes (3)3.2–3.6(4.1) × (0.7)1–1.1(1.4) mm, narrowly ellipsoid-trigonous to almost oblongoid, yellowish-brown when mature, tipped by a very short and wide, obtusely trigonous, persistent, style base.

**Distribution**
Endemic to Cape region (Eastern and Western Cape provinces), South Africa. [27 CPP]. Figure 6F.

**Habitat**
Open and sunny places on shale soils, commonly observed in after-fire vegetation (Fynbos Biome: mostly in Renosterveld); 5–750 m.

**Etymology**
From the latin *capens-is*, from the Cape, in reference to the South African region where it was discovered and to where it is endemic.

**Iconography**
Figures 8F and 13. *Kükenthal (1909*, sub *Schoenoxiphium ecklonii*).

**Conservation**
Considered as Least Concern (LC) at the national level in South Africa (*SANBI, 2020*; sub *Schoenoxipium ecklonii*).

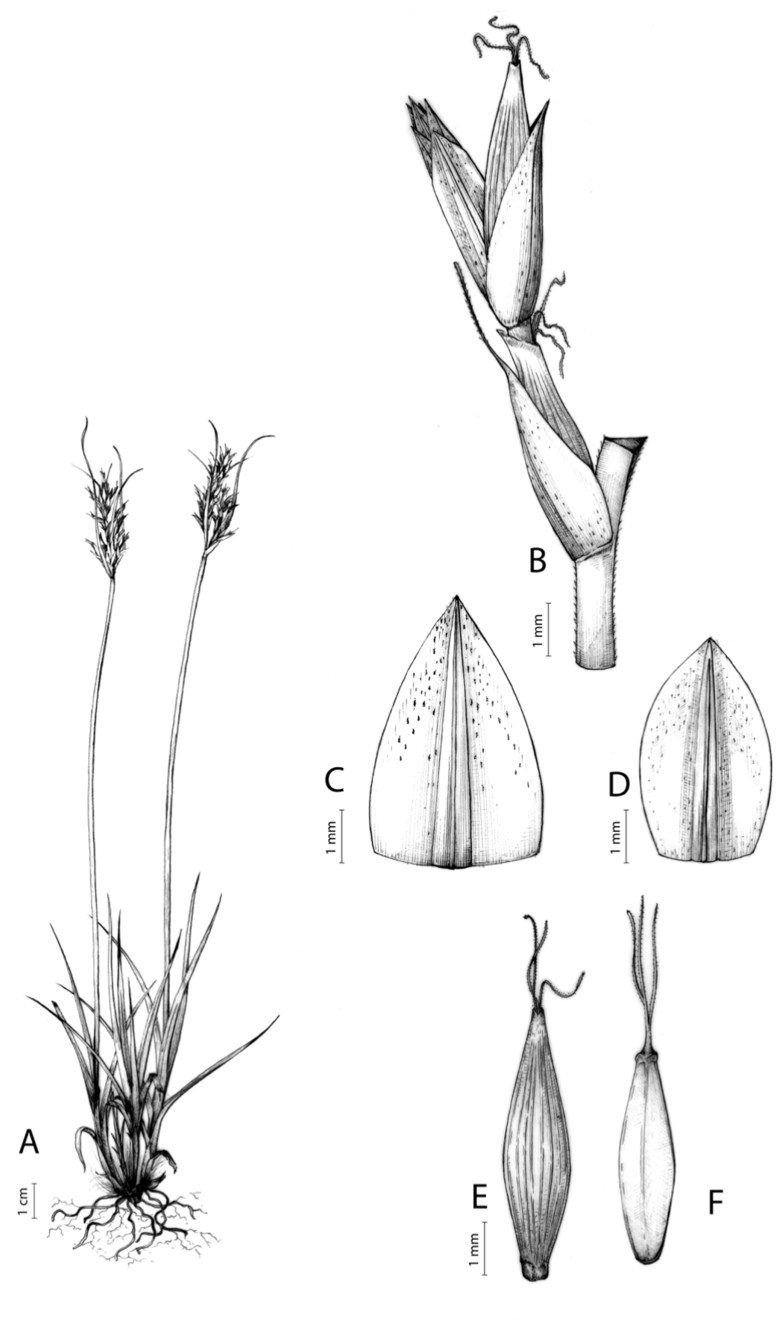

M.S. VILLEGAS '20.

**Figure 13 Illustration of *Carex capensis*.** (A) General aspect of the plant showing two fertile culms. (B) Utriculiform cladoprophyll. (C) Female glume. (D) Male glume. (E) Unisexual utricle. (F) Achene. Illustration by M. Sánchez-Villegas.

## Notes

We have been unable to locate the lectotype of *Schoenoxiphium ecklonii* Nees, designated by *Kukkonen (1983)*. Kukkonen considered this species as distinct (*Kukkonen, 1983*), as also confirms the revision labels he left in numerous herbarium specimens (M. Luceño, 2021, personal observation). After the examination of the type of *C. capensis* Thunb.,

we consider *S. ecklonii* to be a synonym of the former, in view of the multiple specimens revised by Kukkonen whose taxonomic identity matches that of the *C. capensis* type. In addition we have located a number of potential syntypes of *S. ecklonii* that belong to *C. capensis* ([South Africa], Zwartkopsrivier, August, *Ecklon & Zeyher 4440*, SAM-0023405 digital image!, second specimen from the left in the upper middle half; Uitenhage, Grassrug, Kalksteinhaltige, grasreiche flächen & Hügelsnischen, Koega & Zondagsrivier, 500–1,000 ft., 15-VIII, *Ecklon & Zeyher*, S-101142).

**Carex chermezonii** Luceño & Martín-Bravo, Bot. J. Linn. Soc. 179: 27, 2015.

≡ *Schoenoxiphium gracile* Cherm., Bull. Soc. Bot. France 70: 300, 1923; non *Carex gracilis* Curtis (1782).
Type. Madagascar. Mont Tsaratanana, 2,500 m, X-1912, *Perrier de la Bâthie, 2502* (holotype: P-00536708 digital image!).

Rhizome caespitose, moderately stout, brown. Flowering culms 29–51 cm long, erect, obtusely trigonous, scabrid to the apex, leafy up to 1/3 of its length, 1–1.2 mm wide at the middle. Leaves 1.6–4.2 mm wide, shorter than the inflorescence, moderately rigid, more or less glaucous, flat in cross-section, straight at the apex; ligule not measured. Basal sheaths not or scarcely fibrous, usually with lamina, but those of the sterile shoots bladeless. Lowest bract leaf-like, longer than the inflorescence, not sheathing. Inflorescence c. ⅓ of the length of the culm, branching up to 3 times; partial inflorescences 2–5, the lowest short-pedunculate and more or less distant, the upper sessile to shortly pedunculate and overlapping. Glumiform perigynia and glumiform cladoprophylls not seen. Tubular cladoprophylls not seen. Utriculiform cladoprophylls present. Male glumes up to 4.5 mm long, lanceolate to ovate, yellowish-brown to brown, with a green central band, acute to acuminate. Female glumes 3.4–4.5 mm long, ovate, yellowish-brown to brown, shortly aristate, with a green central band. Unisexual utricles 4.9–7 × 0.6–0.8 mm, linear to narrowly ellipsoid, stipitate, strongly arcuate to straight, straw-coloured to yellowish-brown when mature, with numerous prominent veins across the entire surface, suberect, gradually attenuate into an obliquely truncate beak up to 2 mm long; rachilla reaching to slightly protruding from the apex of the utricle; bisexual utricles with the beak widely and obliquely truncate. Achenes up to 6 mm long, oblong.

**Distribution**
Endemic to N Madagascar (Tsaratanana mountains), only known from the type collection. [29 MDG]. Figure 6G.

**Habitat**
Montane forest; c. 2,500 m.

**Etymology**
Named after Henri Chermezon (1885–1939), French botanist specialized in tropical sedges, mainly from Madagascar. He was who originally described this species.

## Conservation
Not Evaluated (NE).

**Notes**:
To our knowledge, the only known collection of this species are those of the type material, which consists of two specimens in whose photograph (P-00536708) we have relied on the preparation of this description. Our measurements match those contained in the protologue (Chermezon, 1923), but differ significantly from those published by *Kukkonen (1983)*. This is especially pronounced regarding the length of the fertile culms and of the achenes. We have taken this last character from the author of the species, who was astonished by the unusual length (6 mm) of the fruit and used this character to differentiate *C. chermezonii* from *C. multispiculata* (Chermezon, 1923).

**Carex distincta** (Kukkonen) Luceño & Martín-Bravo, Bot. J. Linn. Soc. 179: 27, 2015.

≡ *Schoenoxiphium distinctum* Kukkonen, Bot. Not. 131: 263, 1978 [basionym].
Type: Lesotho. Between Indumeni Dome and Castle Buttress, locally common in alpine grassveld on summit of Drakensberg, 9,800 ft., 10-XII-1957, *Killick 2274* (holotype: BM-000922721 digital image!; iso-: K-000363524 digital image!, PRE-0107831 digital image!).

Rhizome caespitose, with short internodes, slender, brown. Flowering culms (10)15–40 (64) cm long, obtusely trigonous, smooth or very scarcely aculeate at the apex, leafy usually up to the upper third of its length, (0.5)1–1.3(1.4) mm wide at the middle. Leaves (0.2)1.3–2.8(3.2) mm wide, shorter than the inflorescence, soft to scarcely rigid, light green, canaliculate, carinate or, more rarely, flat in cross-section, scabrous along the edges, at least in the distal half; abaxial surface smooth, except in the distal parts of the midrib; adaxial surface smooth; trigonous and usually curled at the apex; ligule (1)1.5–2 mm long. Basal sheaths not or scarcely fibrous, with lamina, more rarely the lowest bladeless. Lowest bract of the inflorescence leaf-like, shorter or, exceptionally, little longer than the inflorescence, with a sheath 18–37 mm long. Inflorescence branching up to 3 times; partial inflorescences 1–4(5) distant; lateral subsessile to long-pedunculate, erect to suberect; terminal more or less oblong, with the first-order branches more or less distichous. Glumiform perigynia and glumiform cladoprophylls absent. Tubular cladoprophylls always present except in the terminal partial inflorescence, hyaline. Utriculiform cladoprophylls always present. Male glumes 3.1–4.2(5) × 1.2–1.5(1.9) mm, ovate, ovate-lanceolate to elliptical, reddish-brown with a green central band, acute to shortly acuminate, rarely obtuse. Female glumes (2.8)3.2–4.3(4.5) × (1.3)1.4–1.8(2) mm, ovate, dark reddish-brown with a green central band, acute to acuminate, occasionally with a scabrid acumen up to 1 mm. Unisexual utricles (3.2)3.7–4(4.1) × 0.7–1 mm, linear-oblongoid to narrowly elipsoid, distinctly stipitate, straight, straw-coloured to yellowish-brown when mature, smooth or disperse and shortly aculeate at the apex, with numerous prominent veins across the entire surface, erect to suberect, gradually attenuated into a more or less bidentate beak 0.6–1 mm; rachilla reaching or protruding a little from the

apex of the utricle or, more rarely, much protruding from the mouth and bearing some small, sterile glumes at the top, but then, utricles wide and obliquely truncate. Bisexual utricles wide and asymmetrically truncate. Achenes (2.1)2.3–2.8(2.9) × 0.6–1 mm, oblong-trigonous, yellowish-brown, tipped by a short, obtusely trigonous, persistent style base.

## Distribution
Endemic to Drakensberg and the northeast mountains of Western Cape. Lesotho (Leribe, Maseru, Mohale's Hoek, Mokhotlong and Qacha's Nek districts) and South Africa (Eastern Cape, KwaZulu-Natal and Western Cape provinces). [27 CPP LES NAT]. Figure 6H.

## Habitat
Open, stony and usually dry meadows and shrubland at high altitudes (Grassland Biome: Drakensberg Grassland); 1,750–3,000 m.

## Etymology
From the latin *distinctus-a-um*, distinct, distinguishable to the eye, probably because the species is easily recognizable because of its aspect.

## Iconography
Figures 8G and 14. *Kukkonen (1978)*.

## Conservation
Considered as Least Concern (LC) at national level in South Africa (*SANBI, 2020*; sub *Schoenoxipium distinctum*).

**Carex dregeana** Kunth, Enum. Pl. 2: 511 (1837).

Type. [South Africa] Zuurbergen, Bontjesrivier-Strubels, 2,500–3,000′, 4-XI-1829, *Drège 2033a* (lectotype: P-00540802 digital image!, here designated). Syntypes: [South Africa] Afr. austr., Zuurberg range, Alexandria Div., July, 2,000–3,000 ft., Drège 1040 (K-000693805 specimen in the upper left corner, digital image!); [South Africa] S Africa, Zuurberg range, Alexandria Div., July, 2,000–3,000 ft., Drège s.n. (K-00363530 specimen in the left, digital image!)

≡ *Schoenoxiphium caricoides* C.B.Clarke, Bull. Misc. Inform. Kew, Addit. Ser. 8: 67 (1908).
≡ *Schoenoxiphium kunthianum* Kük. in Engler (ed.), Pflanzenr. 38(IV, 20): 31 (1909).
≡ *Archaeocarex kunthiana* (Kük.) Pissjauk., Bot. Mater. Gerb. Bot. Inst. Komarova Akad. Nauk S.S.S.R. 12: 83 (1950).
≡ *Kobresia kunthiana* (Kük.) T. Koyama, J. Fac. Sci. Univ. Tokyo, Sect. 3, Bot. 8: 80 (1961).

Rhizome caespitose, with short internodes, moderately stout, dark-brown. Flowering culms (15)20–50(60) cm long, erect, more or less acutely trigonous, smooth, leafy up to the lower two-thirds of its length, (0.5)0.8–1.2(1.6) mm wide at the middle. Leaves (0.9)1.5–3.2 (3.5) mm wide, shorter than the inflorescence, moderately rigid, light green, flat to

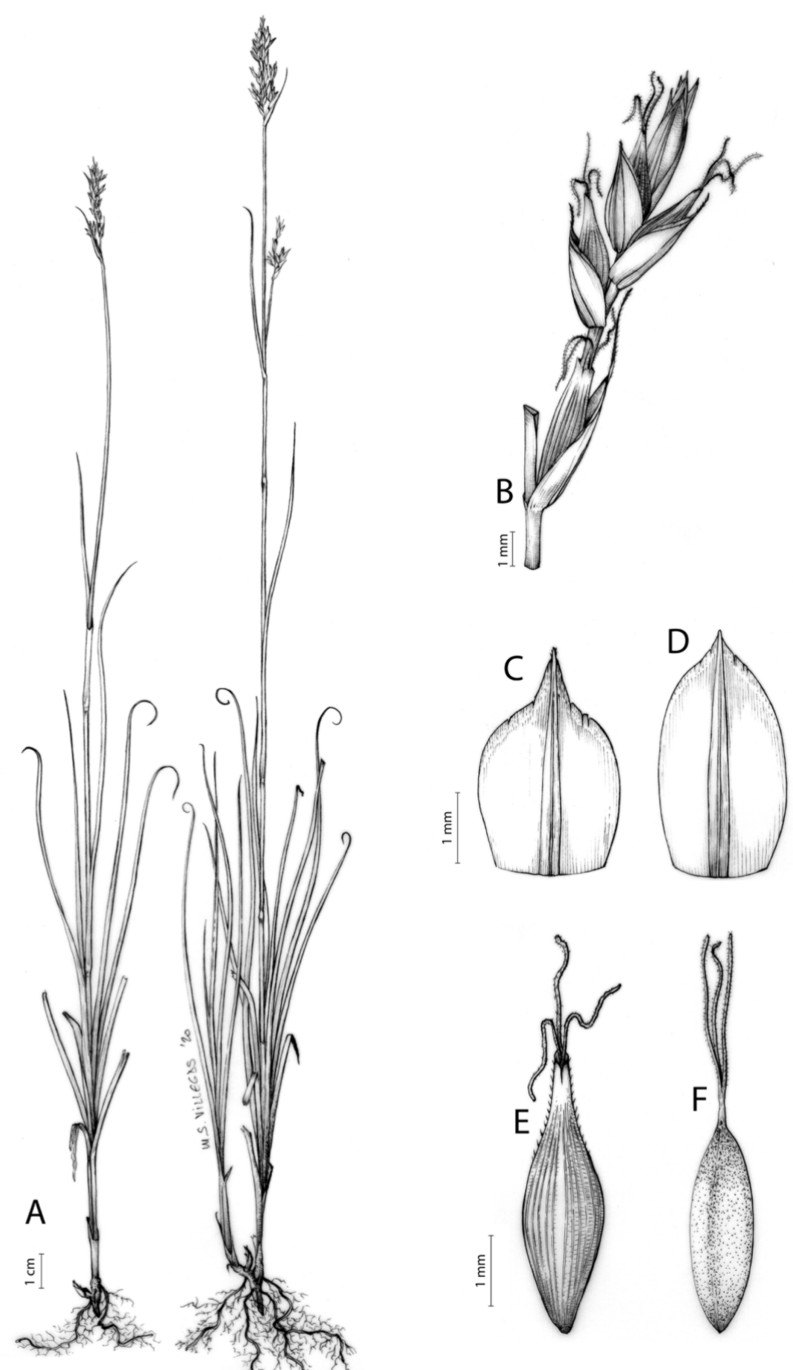

**Figure 14 Illustration of *Carex distincta*.** (A) General aspect of two specimens. (B) Utriculiform cladoprophyll. (C) Female glume. (D) Male glume. (E) Unisexual utricle. (F) Achene. Illustration by M. Sánchez-Villegas.                               

carinate or canaliculate in cross-section, scabrous along the edges and usually also along the abaxial midrib; adaxial surface smooth, sometimes very sparsely aculeate towards de apex, straight or, very rarely, curved to curled; ligule (0.6)0.8–5(8) mm long. Basal sheaths with lamina, densely fibrous. Lowest bract of the inflorescence leaf-like, shorter

than the inflorescence length, usually much widened at the base and with scarious margin, as the remaining bracts, with a sheath (8)15–30(32) mm long. Inflorescence up to the upper ½ of the length of the culm, branching up to 3 times; partial inflorescences 3–5(7), erect, the 1–3 lowermost distant, pedunculate, the (2)3–4 uppermost overlapping, subsessile, rarely all subsessile and overlapping. Glumiform perigynia and glumiform cladoprophylls absent. Tubular cladoprophylls always present at the base of the peduncles, hyaline-brownish. Utriculiform cladoprophylls very rarely present. Male glumes (2)2.5–3.8(4.1) × (1)1.5–2.1(2.3) mm, ovate to elliptical, yellowish-brown, ending in an aculeate mucro or arista up to 1.3 mm long. Female glumes (2)2.5–4(5.2) × (1.2) 1.7–2.6(2.8) mm, ovate to suborbicular, yellowish to yellowish-brown, ending in an aculeate mucro up to 3(4) mm long. Unisexual utricles (2.1)2.5–3.5(4) × 1.1–1.9 mm, ovate to ellipsoid, stipitate, straight, straw-coloured to yellowish-brown when mature, glabrous, smooth or, exceptionally, with some scattered pricklets, with numerous prominent veins across the entire surface, more or less suberect, abruptly contracted in a bidentate to irregularly truncate, smooth, minute beak up to 0.3 mm long; rachilla rudimentary to reaching up to ½(⅔) of the utricle. Bisexual utricles very rare, widely and obliquely truncate at the apex. Achenes 2–3(3.2) × 1–1.5(1.7) mm, ellipsoid-trigonous, dark brown when mature, tipped by a very short, obtusely trigonous, persistent style base.

## Distribution

Eastern and southern parts of Africa and Madagascar with a disjunct population in Western Angola. Angola (Benguela province), Eswatini (HhoHho district), Kenya (Kericho and Trans Nzoia counties), Lesotho (Leribe and Qacha's Nek districts), Madagascar (Fianarantsoa province), South Africa (Eastern Cape, Free State, KwaZulu-Natal, Mpumalanga, Northwest and Western Cape provinces), Tanzania (Rukwa and Ruvuma regions), Uganda (Northern region) and Zimbabwe (Manicaland province). [25 KEN TAN UGA 26 ANG ZIM 27 CPP LES NAT OFS SWZ TVL 29 MDG]. Figure 15A.

## Habitat

Dry to damp grassland and grassy clearing of shrubs (mainly in Grassland Biome, more rarely in Albany Thicket, Indian Ocean Coastal Belt and Savanna biomes) and extending into afromontane areas of tropical Africa; 200–2,850 m.

## Etymology

Named after Johann Franz Drège (1794–1881), German (with Huguenot ancestry) botanist and explorer, who collected c. 8,000 species and c. 200,000 specimens throughout the present Eastern Cape, KwaZulu-Natal, Northern Cape and Western Cape provinces, between 1826 and 1832.

## Iconography

Figures 16 and 17A. Haines & Lye (1983, sub Schoenoxiphium caricoides).

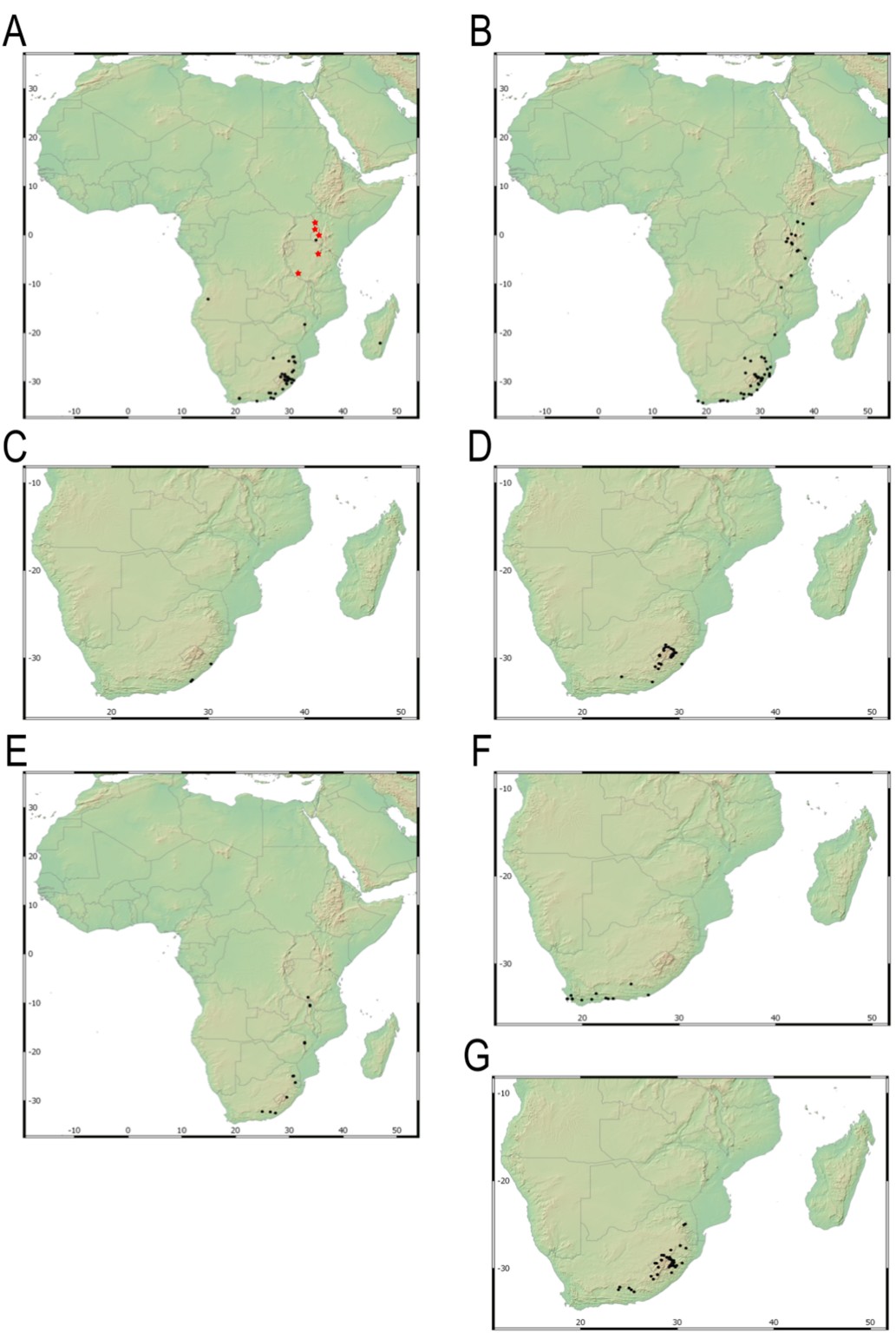

**Figure 15 Distribution map of different species of *Carex* section *Schoenoxiphium*.** (A) *C. dregeana* (red stars: references from *Haines & Lye (1983)*, black circles: studied materials). (B) *C. esenbeckiana* (C) *C. gordon-grayae*. (D) *C. killickii*. (E) *C. kukkoneniana*. (F) *C. lancea*. (G) *C. ludwigii*. Maps created with QGIS.

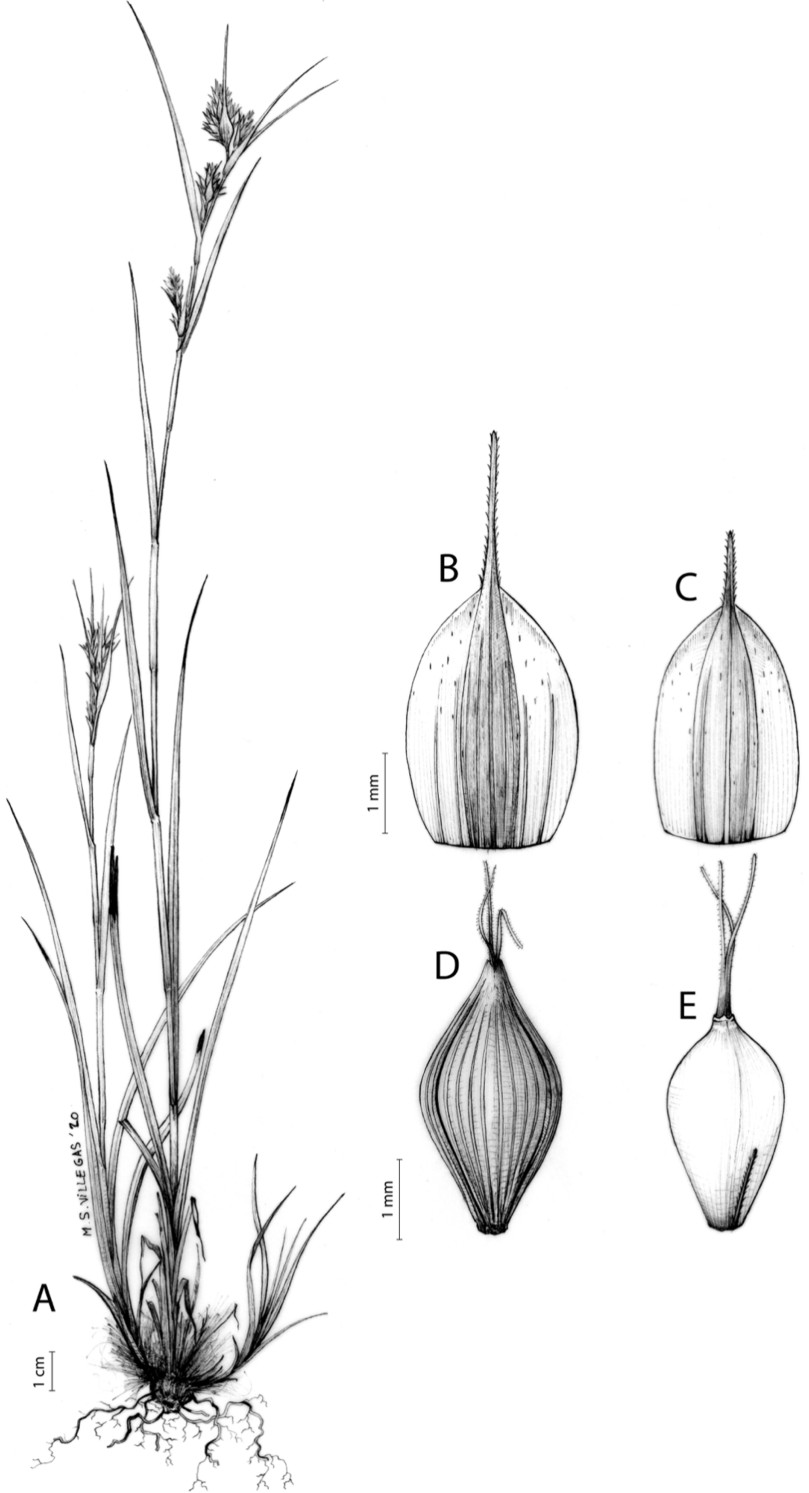

**Figure 16 Illustration of *Carex dregeana*.** (A) General aspect of the plant showing two fertile and one sterile culms. (B) Female glume. (C) Male glume. (D) Unisexual utricle. (E) Achene. Illustration by M. Sánchez-Villegas.

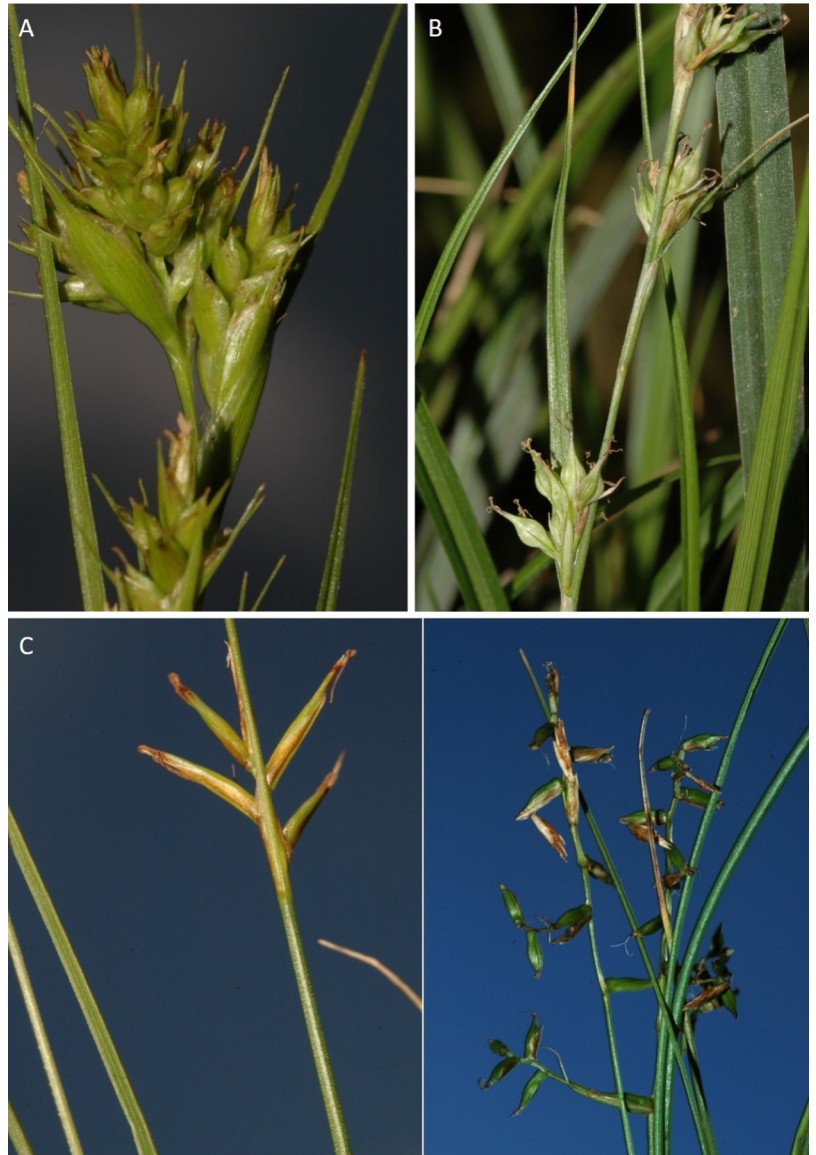

**Figure 17 Morphology of different species of *Carex* section *Schoenoxiphium*.** (A) *C. dregeana*. (B) *C. esenbeckiana*. (C) *C. killickii*. Photos by M. Luceño.

**Conservation**

Due to the long-standing problems concerning delimitation of this species, it has not previously been evaluated under the taxonomic concept here proposed, and should therefore be considered Not Evaluated (NE).

**Notes**

*Carex dregeana* is the only species in the section whose rachilla does not reach the apex of the unisexual utricles and, moreover, they are frequently reduced to a barely visible, vestigial elongation of the branch at the base of the utricle. This species is easily recognizable both for that peculiarity and for the very rare presence of bisexual utricles and utriculiform cladoprophylls. Another character that allows the correct identification of this

species from the similar ones, as *C. spartea* and *C. esenbeckiana*, is the utricle beak length, that does not lengthen beyond 0.3 mm. In addition, the upper partial inflorescence bracts usually present a wide base and an evident scarious margin, that elongates in a subulate apex. This species hybridizes with certain frequency with *C. spartea*, resulting in numerous intermediate morphotypes, characterised by the rachillas that do not reach the apex of the utricle, bisexual utricles sometimes present, bracts not or slightly widened at the base, and utricle beak up to 0.6 mm. These hybrids present some irregularities in the meiotic pairing (see Results, cytogenetic section).

We have designated the lectotype of *C. dregeana* Kunth on a specimen from P, where Kunth was established (*Stafleu & Cowan, 1976*). Our designation intends to conserve the current usage of the name. There are a number of specimens also collected by Drège in South Africa, and bearing a label probably handwritten by Kunth but their labels are not a perfect match to the one of the selected lectotype, so we have considered such material as syntypes.

**Carex esenbeckiana** Boeckeler, Linnaea 40: 372 (1876)

≡ *Uncinia lehmannii* Nees, Linnaea 10: 206 (1836), non *Carex lehmannii* Drejer (1844). Type: [South Africa] In montibus prope urban Cape, *Ecklon & Zehyer site 85* [Plantae capenses. "Cap, östlicher Abhang des Tafelberges, bei Konstantia" (see *Drège, 1847a*: 258)] (lectotype: S-10-1129 digital image!, designated by *Kukkonen (1983)*).
≡ *Schoenoxiphium lehmannii* (Nees) Kunth ex Steud., Syn. Pl. Glumac. 2: 245 (1855).
≡ *Schoenoxiphium sparteum* var. *lehmannii* (Nees) Kük. in Engler, Pflanzenr. 38(IV, 20): 32 (1909).
≡ *Kobresia lehmannii* (Nees) T.Koyama, J. Fac. Sci. Univ. Tokyo, Sect. 3, Bot. 8: 80 (1961).

=*Carex uhligii* K.Schum. ex C.B.Clarke, Bot. Jahrb. Syst. 38(2): 136, 1907.
Type: Tanzania, Usambara, zwischen Mbalu und Mlalo, XII-1901, *Uhlig 856* (lectotype: EA s.n. digital image!, here designated).

Rhizome densely caespitose, with very short internodes, moderately stout, dark-brown. Flowering culms (10)40–70(90) cm long, more or less acutely trigonous, smooth to slightly scabrous, leafy up to 1/10 (¼) of its length, (0.6)0.8–1.1(1.3) mm wide at the middle. Leaves (0.9)2–5(7) mm wide, shorter, more rarely equaling, than the inflorescence, soft to very slightly rigid, straight to scarcely curved at the apex, glaucous, flat, more rarely, scarcely canaliculate, carinate or plicate in cross-section, scabrous along the margins and the abaxial midrib to the apex, mainly in the upper half; adaxial surface smooth or somewhat aculeate; ligule (0.9)1.5–2.5(3.4) mm long. Basal sheaths with lamina, more rarely lowermost bladeless, moderately to densely fibrous. Lowest bract of the inflorescence leaf-like, shorter, rarely longer, than the inflorescence, with a sheath (7)15–30(44) mm long. Inflorescence up to the upper 85% of the length of the culm, branching up to 2(3) times; partial inflorescences (6)7–10(12), all distant and pedunculate, except the 2(3) uppermost, that are subsessile and overlapping; peduncles included in the sheaths or slightly protruding from it. Glumiform perigynia and glumiform cladoprophylls absent.

Tubular cladoprophylls always present. Utriculiform cladoprophylls rarely present. Male glumes (1.7)2–3.5(3.9) × (0.6)1.5–2 mm, widely ovate to oblong, straw-coloured to yellowish-brown, with a green central band, acute to ending in an aculeate mucro up to 0.3(2) mm long. Female glumes (1.8)2–3.5(3.7) × (1)1.5–1.9(2.1) mm, ovate, straw-coloured to, more frequently, pale reddish brown, with a green central band, acute to ending in an aculeate mucro up to 3 mm long. Unisexual utricles (3.1)3.8–5.5(6) × (1)1.1–1.5(1.9) mm, widely ovoid to ellipsoid, long-stipitate, straight, more rarely slightly curved, straw-coloured to brownish when mature, smooth, with conspicuous prominent veins over the entire surface, suberect to erecto-patent, very abruptly contracted, into a smooth, bidentate to irregularly truncate beak (0.7)1.3–2(2.3) mm long; rachilla reaching the apex of the utricle or, more frequently protruding from it up to 0.5(1) mm. Bisexual utricles very rare, althoug ocasionally predominant in some individuals, widely and obliquely truncate at the apex. Achenes (2.1)2.3–2.7(3) × (1.2)1.3–1.5(1.6) mm, ellipsoid-trigonous, more or less straw-coloured to brown when mature, tipped by a short, obtusely trigonous, distinctly neck-like, rarely asymmetric, persistent style base.

### Distribution
Eastern and southern parts of Africa. Ethiopia (Oromia region), Kenya (Laikipia and Marsabit counties), Lesotho (Leribe district), Malawi (Rumphi district), South Africa (Eastern Cape, Free State, Gauteng, KwaZulu-Natal, Mpumalanga, Northwest and Western Cape provinces), Tanzania (Arusha and Kilimanjaro regions) and Zimbabwe (Manicaland province). [24 ETH 25 KEN TAN 26 MLW ZIM 27 CPP LES NAT OFS TVL]. Figure 15B.

### Habitat
Shady places, mainly in forest understories (Afrotemperate, Subtropical and Azonal Forests Biome: Southern Afrotemperate Forest, Northern Afrotemperate Forest, Southern Mistbelt Forest, Northern Mistbelt Forest, Scarp Forest, Southern Coastal Forest and Northern Coastal Forest) and extending into afromontane areas of tropical Africa; 4–2,300 m.

### Etymology
Named after Christian Gottfried Daniel Nees von Esenbeck (1776–1858), German botanist, zoologist and natural philosopher that described nearly 7,000 plant species, almost the same number as Linnaeus. Remarkably, he described the genus *Schoenoxiphium* (*Nees, 1832*). He was president of the German Academy of Natural Scientist Leopoldina when Charles Darwin was admitted to that institution.

### Iconography
Figures 17B and 18. *Kükenthal (1909*; sub *Schoenoxiphium sparteum* var. *lehmannii*), *Haines & Lye (1983*; sub *Schoenoxiphium lehmannii)*.

### Conservation
Considered as Least Concern (LC) at the national level in South Africa (*SANBI, 2020*; sub *Schoenoxipium lehmannii*).

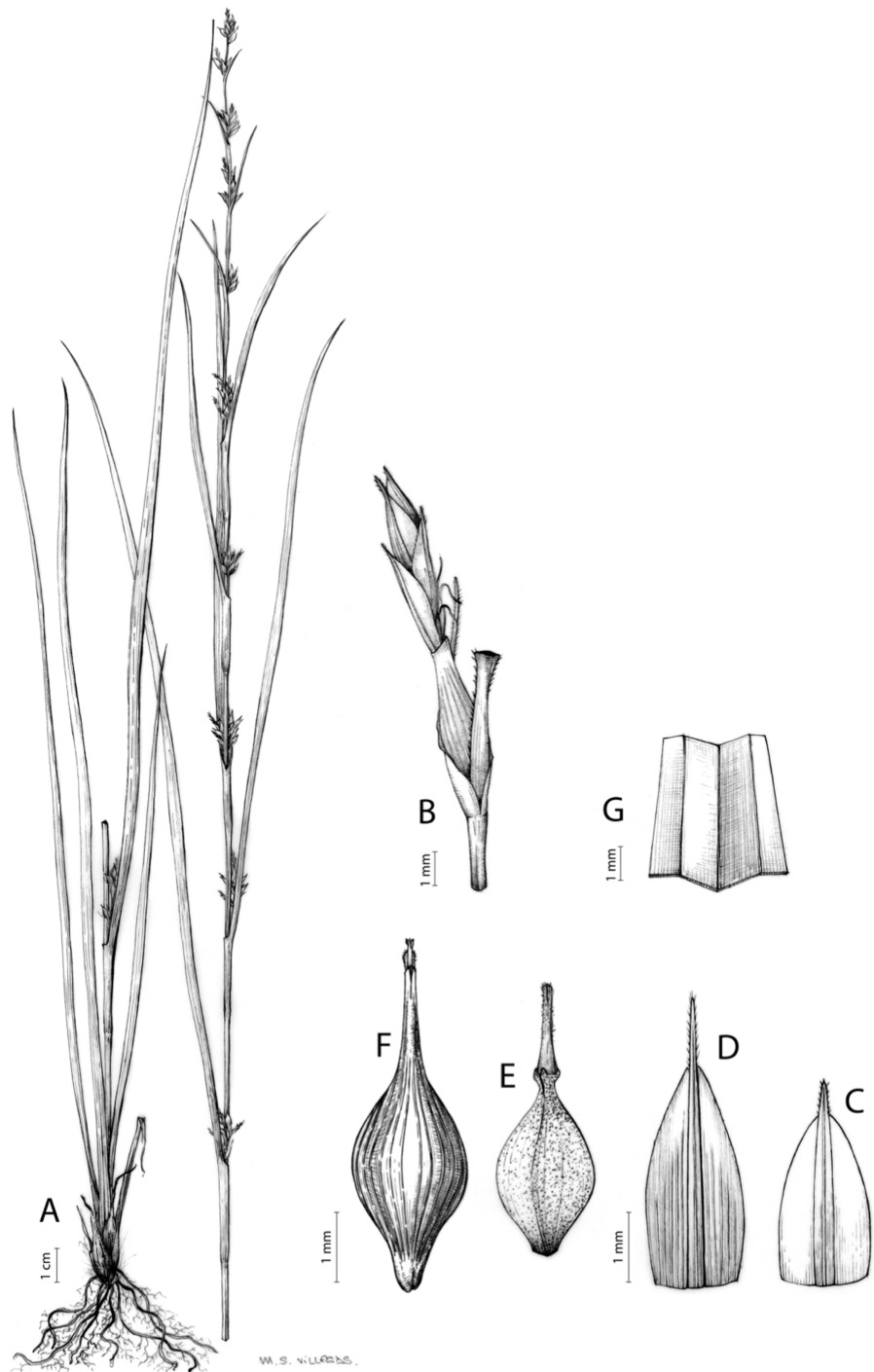

**Figure 18 Illustration of *Carex esenbeckiana*.** (A) General aspect of the base of the plant and inflorescence. (B) Utriculiform cladoprophyll. (C) Male glume. (D) Female glume. (E) Achene. (F) Unisexual utricle. (G) Fragment of a basal leaf. Illustration by M. Sánchez-Villegas.

**Carex gordon-grayae** Luceño, Márquez-Corro & Sánchez-Villegas, sp. nov.

**Diagnosis**: Similar in appearance to *C. esenbeckiana* Boeckeler, from which it differs by its wider leaves ((4.5)5.5–9 mm in *C. gordon-grayae* vs. (0.9)2–5(7) mm in *C. esenbeckiana*), the utricles (5.4–7.9 mm long, linear-oblong to narrowly ellipsoid, gradually attenuated in the beak in *C. gordon-grayae* vs. (3.1)3.8–5.5(6) mm long, ovoid to broadly ellipsoid, abruptly contracted in the beak utricles in *C. esenbeckiana*), and by its broadly pyramidal style base in *C. gordon-grayae* vs. shortly trigonous, neck-like style base in *C. esenbeckiana*.

Holotype: South Africa. KwaZulu-Natal, Port Shepstone district, Baboon Spruit, Oribi Gorge, Oribi Gorge, II-1973, *L.E. Davidson 2649* (NU-0002643-0!). Paratypes: South Africa, Eastern Cape, Hagga-Hagga district, marsh, 7-II-1995, *B. Sonnenberg 446* (NU-s. n.!). Ibidem, Kentani, 32°30′7.628″S 28°20′9.499″E, edge of stream in forest, 4-IV-1904, *A. M. Pegler 1097* (BOL-4655 digital image!).

Rhizome laxly caespitose, with short internodes, moderately stout, medium to dark-brown. Flowering culms 29.5–65 cm long, obtusely trigonous, smooth, leafy up to half its length, 0.9–1.7 mm wide at the middle. Leaves (4.5)5.5–9 mm wide, shorter than the inflorescence, somewhat rigid, glaucous, flat in cross-section, scabrous along the edges, at least in the upper two-thirds; abaxial surface smooth, but scabrous in the upper third of the mibrid; adaxial surface papillose-aculeate near the margins of the distal half, straight at the apex; ligule 4.5–5.3 mm long. Basal sheaths broken to slightly fibrous, lowermost bladeless. Lowest bract of the inflorescence leaf-like, shorter than the inflorescence length, with a sheath 20–40 mm long. Inflorescence branching up to three times; partial inflorescences 3–5, very distant, except terminal ones; lateral from subsessile to shortly pedunculate, erect. Glumiform perigynia and glumiform cladoprophylls absent. Tubular cladoprophylls always present at the base of partial inflorescences and in some first order branches of lateral partial inflorescences, hyaline. Utriculiform cladoprophylls usually present at lower parts of developed partial inflorescences. Male glumes 2.9–4.2 × 0.9–1.4 mm, lanceolate to ovate, hyaline, with a green central band, acute to acuminate, usually with a scabrid acumen up to 1.1 mm long. Female glumes 4–6.5 × 1.3–1.9 mm, lanceolate to ovate, hyaline, with a narrow, green central band, with a scabrid acumen up to 2.5 mm long. Unisexual utricles 5.4–7.9 × (0.9)1.1–1.5 mm, linear-oblong to narrowly-ellipsoid, shortly stipitate, straight to very slightly curved, green to yellowish brown when mature, sparsely aculeate at the apex of the beak, with numerous prominent veins across the entire surface, erect, gradually attenuated into an obliquely truncate, hyaline at the apex beak 1.4–2.5 mm long; rachilla protruding from the apex up to 2.5 mm. Bisexual utricles similar to unisexal ones, but more widely mouthed. Achenes 3.5–4.3 × 0.9–1.3 mm, ellipsoid-trigonous, yellowish-brown, tipped by a broadly pyramidal and shortly neck-like at the lower part, persistent style base.

### Distribution

Endemic to Eastern Cape and KwaZulu-Natal provinces (South Africa). [27 CPP NAT]. Figure 15C.

### Habitat

Shady places, mainly in margins and clearing of forests, close to the coast (Indian Ocean Coastal Belt Biome: Pondoland-Ugu Sandston Coastal Sourveld); 200–400 m.

### Etymology

The epithet honours Professor Kathleen D. Gordon-Gray (1918–2012), a South African Botanist, as homage to her excellent works in KwaZulu-Natal Cyperaceae, especially in the former genus *Schoenoxiphium*. In the holotype she noted: "This may possibly be an unusual specimen of *Sch. lehmannii* (Nees) Steudel, but the much wider leaves and longer utricles (7 mm) make this doubtful … Field investigation needed".

### Iconography

Figures 19 and 20.

### Conservation

Not Evaluated (NE).

***Carex killickii*** Nelmes, Kew Bull. 10: 89, 1955.

Type. [Lesotho] Basutoland, between Indumeni Dome and Castle Buttress, abundant in moist parts of alpine grasveld, 9,700 ft, 05-XII-1952, *Killick 1848* (lectotype: K-000363522 digital image!, here designated; isolecto-: K-000363523 digital image!, NU-0015692-0 digital image!, PRE).
= *Schoenoxiphium filiforme* Kük., Bull. Misc. Inform. Kew 1910: 129; non *Carex filiformis* L. (1753).

Type: South Africa. Cape Colony, summit of Great Winterberg, under rocks, 2,570 m, 8-III-1900, *Galpin 5605* (lectotype: K-000693800 digital image!, designated by *Kukkonen (1983)*; iso-: PRE-0107826 digital image!, GRA-0000244 digital image!).

= *Schoenoxiphium molle* Kukkonen, Notes Roy. Bot. Gard. Edinburgh 43: 366, 1986. Type: South Africa. KwaZulu-Natal, Underberg district, 5 - 7 miles NNW of Castle View Farm, headwaters of Mlahlangubo River. c. 7,000 ft, 23-XI-1980, *Hilliard & Burtt 13574* (holotype: H-1455011 digital image!; iso-: E-00286097 digital image!, GENT-0000090032813 digital image!, NU-0015652 digital image!).

= *Schoenoxiphium strictum* Kukkonen, Notes Roy. Bot. Gard. Edinburgh 43: 366, 1986. Type: South Africa. KwaZulu-Natal, Underberg district, Garden Castle Forest Reserve, Mlambonja Valley. 6,800 ft, 05.01.1982, *Hilliard & Burtt 14909* (holotype: H-1461500 digital image!; iso-: E-0286107 digital image!, GENT-0000090032929 digital image!, NU-0015651 digital image!).

Rhizome loosely to densely caespitose, with short internodes, slender, brown. Flowering culms (8.5)15–50(95) cm long, acutely trigonous, smooth or, more rarely, somewhat aculeate at the apex, leafy usually up to half (upper third) of its length, 0.3–0.9(1.5) mm wide at the middle. Leaves 0.2–1(3.6) mm wide, shorter to longer than the inflorescence,

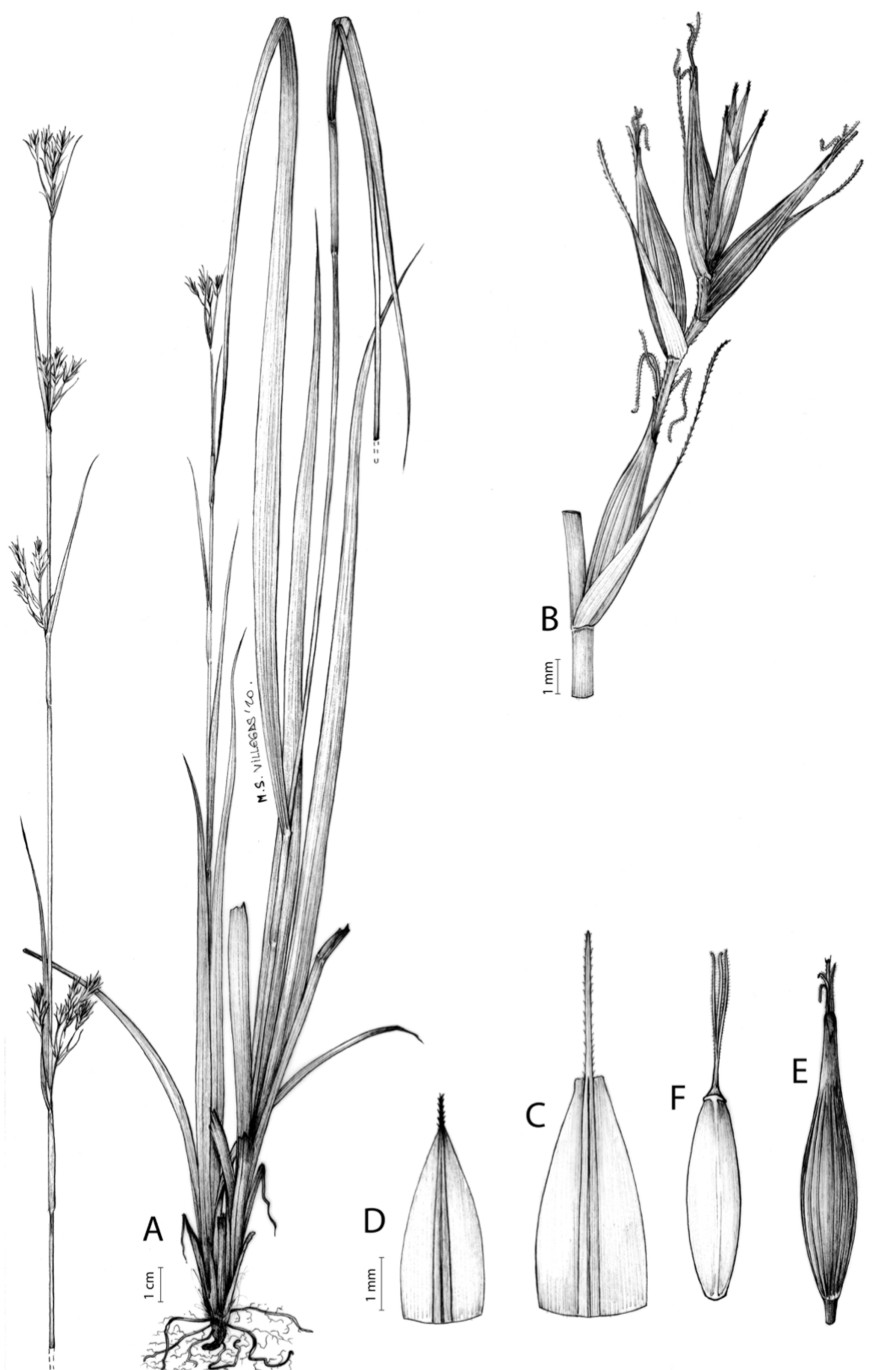

**Figure 19 Illustration of _Carex gordon-grayae._** (A) General aspect of the base of the plant and inflorescence. (B) Utriculiform cladoprophyll. (C) Female glume. (D) Male glume. (E) Unisexual utricle. (F) Achene. Illustration by M. Sánchez-Villegas.

soft to somewhat rigid, light green, canaliculate to involute more rarely flat in cross-section, more or less scabrous along the edges; abaxial surface smooth or scabrid in the midrib; adaxial surface smooth; somewhat curved or scarcely curled at the apex; ligule 0.3–1.2 mm long. Basal sheaths usually not or scarcely fibrous, more rarely very fibrous,

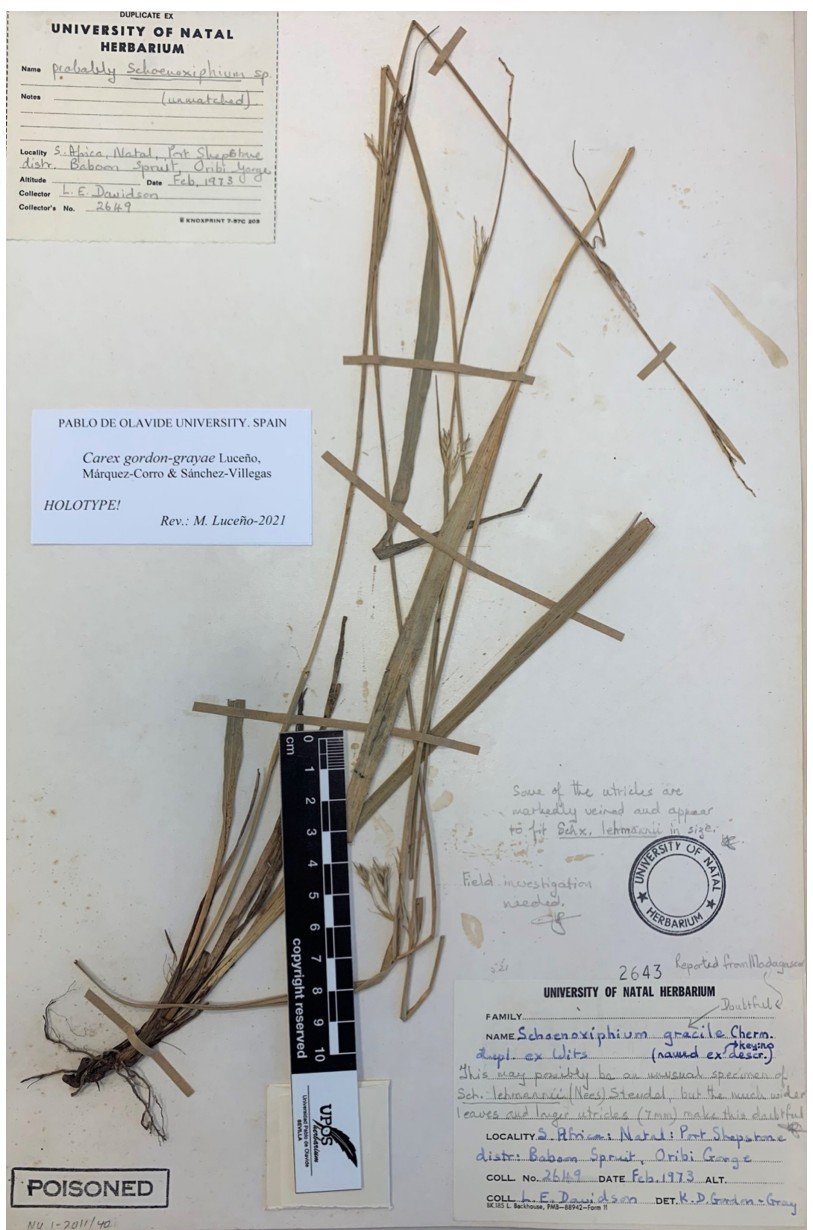

**Figure 20 Holotype of *Carex gordon-grayae*.** Photo of voucher NU-2643 deposited at University of Natal (South Africa). Photo by M. Míguez.

with lamina. Lowest bract of the inflorescence usually glumaceous with setaceous apex or, more rarely, leaf-like, shorter to longer than the inflorescence, not sheathing except when the lowest partial inflorescence is long-pedunculate and clearly separate from the remaining ones, in this case sheath up to 25(40) mm long. Inflorescence reduced to a terminal, androgynous spike or branching up to 3 times; partial inflorescences 1–5(7), overlapping and sessile or, more rarely, the lowest one clearly distant and long-pedunculate, patent, erect-patent or somewhat reflexed except the lowest one distant and pedunculate that is suberect when present. Glumiform perigynia and glumiform

cladoprophylls absent. Tubular cladoprophylls absent or hyaline when the lowest partial inflorescence is distant and long-pedunculate. Utriculiform cladoprophylls present except in unispicate inflorescences. Male glumes (3)3.5–4.2(4.5) × (0.9)1.2–1.5(1.7) mm, ovate-lanceolate, obovate or elliptical, brown with a green central band, acute to acuminate, frequently ending in an arista up to 0.8 mm. Female glumes 3.1–4.2 × 1.2–2 mm, lanceolate to ovate, pale to dark brown with a green central band, acute, acuminate to aristate, sometimes with a scabrid acumen up to 1.8 mm. Unisexual utricles 3.8–5.3(7.2) × 0.5–1.2 mm, narrowly linear-oblongoid to narrowly elipsoid, distinctly stipitate, usually straight, more rarely slightly arcuate, straw-coloured to yellowish-brown when mature, smooth to dispersely and shortly hispid in the upper half, with few to numerous faintly to strongly prominent veins across the entire surface, patent at the maturity, gradually attenuated into a more or less irregular beak up to 1 mm; rachilla reaching or protruding a little from the apex of the utricle or, more rarely, much protruding from the mouth and bearing some small, sterile glumes at the top. Bisexual utricles wide and asymmetrically truncate. Achenes (2.)2.5–3.2(3.6) × 0.4–1 mm, oblong-trigonous, yellowish to pale-brown, tipped by a short, obtusely trigonous to long-pyramidal, persistent style base.

### Distribution
Endemic to the Drakensberg Mountains with isolated localities in Eastern Cape and southeastern KwaZulu-Natal. Lesotho (Butha Buthe, Leribe, Maseru Qacha's Nek and Thaba Tseka districts) and South Africa (Eastern Cape, Free State, KwaZulu-Natal and Western Cape). [27 CPP LES NAT OFS]. Figure 15D.

### Habitat
Open, stony and mesophilous to damp meadows, bogs, and shrubland, usually at high altitudes (Grassland Biome: Drakensberg Grassland; Indian Ocean Coastal Belt Biome: KwaZulu Natal Coast Belt); (550)1,650–3,150 m.

### Etymology
Named after Donald Joseph Boomer Killick (1926–2008), South African botanist specialized in Drakensberg flora, who collected the type material of this species.

### Iconography
Figures 17C and 21.

### Conservation
Not considered of conservation concern at the national level in South Africa, and thus categorized as Least Concern (LC) (see SANBI, 2020, sub Schoenoxipium molle; sub S. filiforme).

### Notes
This species is the most morphologically variable in the section, in particular its inflorescence structure. This has caused taxonomic delimitation problems (e.g. the former Schoenoxiphum molle and S. strictum; see Gordon-Gray, 1995; SANBI, 2020). The common forms of C. killickii present fertile culms of 20–50(60) cm, canaliculate to

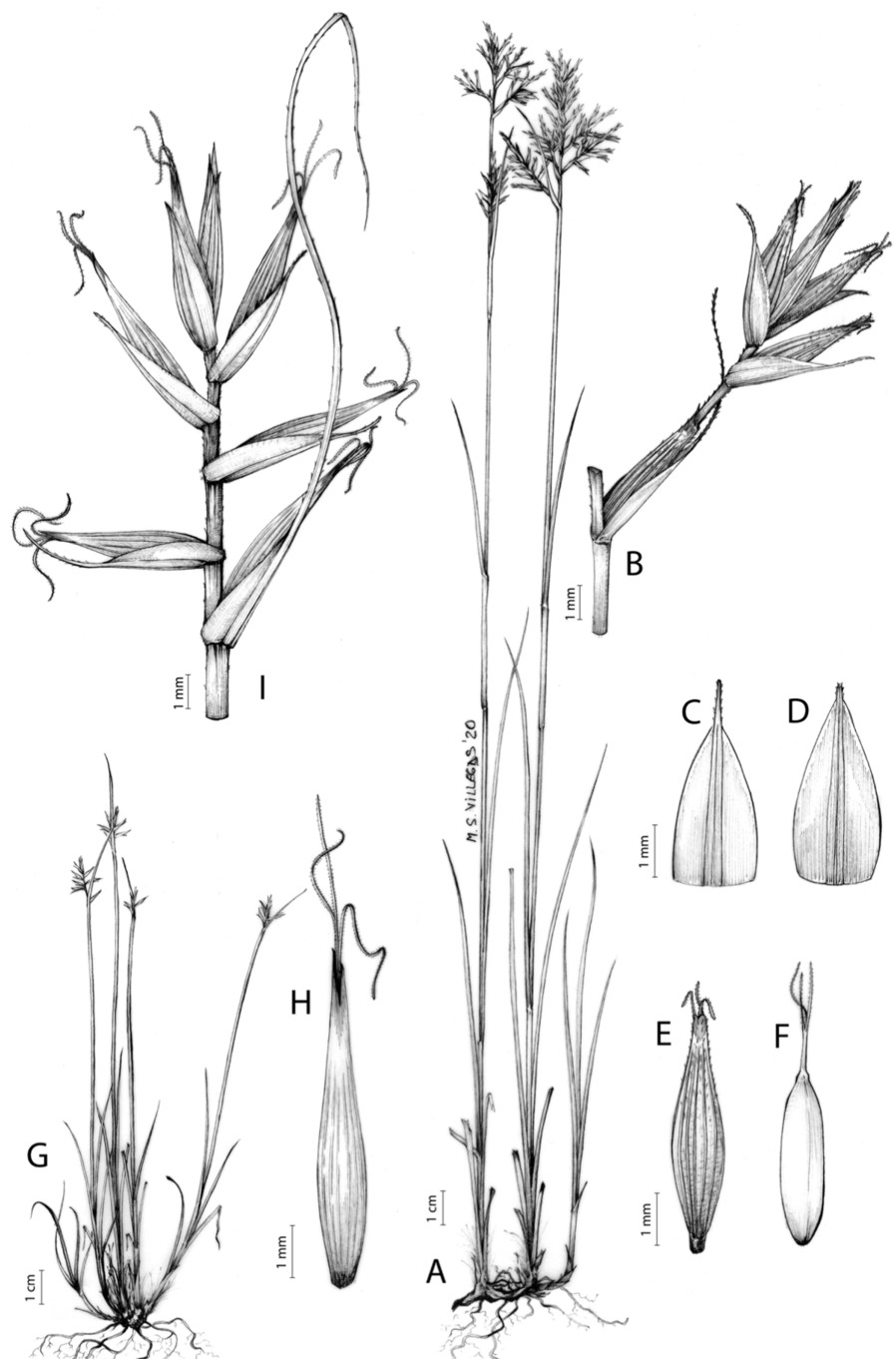

**Figure 21 Illustration of *Carex killickii*.** (A) General aspect of the common morphotype. (B) Utriculiform cladoprophyll. (C) Female glume. (D) Male glume. (E) Unisexual utricle of the common morphotype. (F) Achene. (G) General aspect of the summit morphotype. (H) Unisexual utricle of the summit morphotype. (I) Inflorescence of the summit morphotype. Illustration by M. Sánchez-Villegas.

involve, filiform leaves, unispicate or branched inflorescences and narrowly ellipsoid, smooth or slightly scabrid, utricles up to 5.3 × 0.9–1.2 mm long, with numerous prominent veins. Nevertheless, the specimens that inhabits summits of the Drakensberg mountains

are shorter (8.5–30 cm), with unispicate inflorescences and unisexual utricles narrowly linear, smooth or rarely slightly scabrid, 5–7.2 × 0.5–0.8 mm, mostly without numerous veins and, however noticeable, hardly prominent. This form corresponds to the *C. killickiii* Nelmes type material, described from the mountain summits of Lesotho. Moreover, transition morphotypes towards the more common forms appear in lower altitudes from those mountains. Regarding the type material of *Schoenoxiphium filiforme* Kük. (1910), described from the Great Winterberg Mountain range (Eastern Cape, north of Adelaide), it shows unispicate inflorescences and fertile culms of 20–25 cm, but the utricles are more similar to the common forms. Additionally, as it is shown in the phylogeny of Fig. S2, the sample from the Lesotho summits is nested within more typical forms of the species (unispicate or with branched inflorescences). However, given the molecular variability of this taxon and the fact that a single sample from the summit form was included in the RAD phylogeny, we cannot rule out that the summit morphotype corresponds to a different taxon. That can only be resolved by carrying out a phylogeographic study of the species.

Moreover, densely caespitose forms inhabit the lower slopes of the Drakensberg (1,800–2,300 m), forming tussocks with fertile culms up to 70(95) cm, flat or slightly canaliculate leaves up to 2.5(3.6) mm width that surpass the culms length, inflorescences always branched and utricles sparsely hispid towards the apex or hispid just in the lower half. This morphotype corresponds to the material described as *Schoenoxiphium molle Kukkonen (1983)*. Likewise, in areas with similar habitats appear forms with smooth culms and less wider leaves, flat or canaliculate, that have been described as *S. strictum* Kukkonen. Between these two and the typical forms of the species, a multitude of intermediates phenotypes can be found, so we consider that, until further phylogeographic studies are carried out, these forms should be included under the wide morphological variability of *C. killickii*.

Finally, since the population from Oribi gorge (Port Shepstone distr., KwaZulu-Natal, L. E. Davidson 2648 (NU)) grows in remarkably low altitude (ca. 550 m), with respect to the remaining populations of the species, it would be interesting to further study this population.

**Selected reference**
*Gordon-Gray (1995)*.

**Carex kukkoneniana** Luceño & Martín-Bravo, Bot. J. Linn. Soc. 179: 27, 2015.

Type: [South Africa] Natal [KwaZulu-Natal], IV-1884, G. *Buchanan 134* (holotype: K-000693792 digital image!).

≡ *Carex buchananii* C.B.Clarke in Harvey & auct. suc. (eds.), Fl. Cap. 7: 305. 1898, nom. illeg.; non *Carex buchananii* Berggr. (1880).
≡ *Schoenoxiphium buchananii* C.B.Clarke ex Kük. in Engler (ed.), Pflanzenr. 38(IV, 20): 31 (1909).
≡ *Kobresia buchananii* (C.B.Clarke) T.Koyama, J. Fac. Sci. Univ. Tokyo, Sect. 3, Bot. 8: 80. 1961.

Rhizome caespitose, stout, with short internodes, dark-brown. Flowering culms (77) 100–180(200) cm long, erect, obtusely trigonous, smooth, leafy up to half its length, (2.6) 3–3.7(4) mm wide at the middle. Leaves (4.1)5.5–9.6(12.8) mm wide, shorter than the inflorescence, moderately rigid, glaucous, flat in cross-section, scabrous on the margins, mainly in the uppermost parts, and sometimes also along the abaxial midrib; adaxial surface smooth except at the apical part; straight at the apex; ligule 4–17 mm long. Basal sheaths entire to scarcely fibrous, usually with lamina. Lowest bract leaf-like, shorter than the inflorescence, with a sheath (20)29–63(80) mm long. Inflorescence c. ¼ of the length of the culm, branching up to 4 times; partial inflorescences (4)6–9(11), the 2–4 lowest long-pedunculate and more or less distant, the upper sessile to shortly pedunculate and overlapping. Glumiform perigynia and glumiform cladoprophylls absent. Tubular cladoprophylls always present, hyaline to brownish. Utriculiform cladoprophylls always present. Male glumes (1.9)2.2–3.6(4.2) × 1.4–2.1 mm, ovate, elliptic or obovate, yellowish-brown to dark brown, with a green central band, usually mucronate, with an aculeate mucro up to 0.6 mm long. Female glumes (2.7)3.1–3.8(4.4) × 1.5–2.3(2.8) mm, ovate, yellowish-brown to brown, with a green central band, ending in an aculeate mucro up to 2(4) mm. Unisexual utricles (3.7)5–6.8(7.6) × (0.8)0.9–1.1(1.2) mm, linear to narrowly ellipsoid, long-stipitate, straight or, more rarely, arquate, straw-coloured to yellowish-brown when mature, more or less aculeate in the upper third, with numerous prominent veins across the entire surface, suberect to erecto-patent, gradually attenuate into a split in one side beak up to 1.5(1.9) mm; rachilla reaching to protruding from the apex of the utricle by up to 0.5 mm, sometimes bearing 1-few small, sterile glumes at the top. Bisexual utricles with the apex wide and obliquely truncate. Achenes (2.9)3–3.9(4.1) × (0.6)0.8–1.1 mm, oblong, trigonous, straw-coloured to yellowish-brown when mature, tipped by a neck-like, more rarely subterete, persistent style base.

### Distribution
Eastern Africa, from Tanzania to South Africa: Eswatini (Hhohho district), Malawi (Chitipa and Rumphi districts), South Africa (Eastern Cape, KwaZulu-Natal and Mpumalanga provinces), Tanzania (Iringa and Mbeya regions) and Zimbabwe (Manicaland province). [25 TAN 26 MLW ZIM 27 CPP NAT SWZ TVL]. Figure 15E.

### Habitat
Edges of streams, marshy grounds and other damp soils in mountains (Grassland Biome: Drakensberg Grassland, Sub-Escarpment Grassland and Mesic Highveld Grassland); 950–2,500 m.

### Etymology
Named after Dr. Toivo Ilkka Kalervo Kukkonen (1926–2020), Finnish botanist specialized in sedges, who worked in the former genus *Schoenoxiphium* and described several new species.

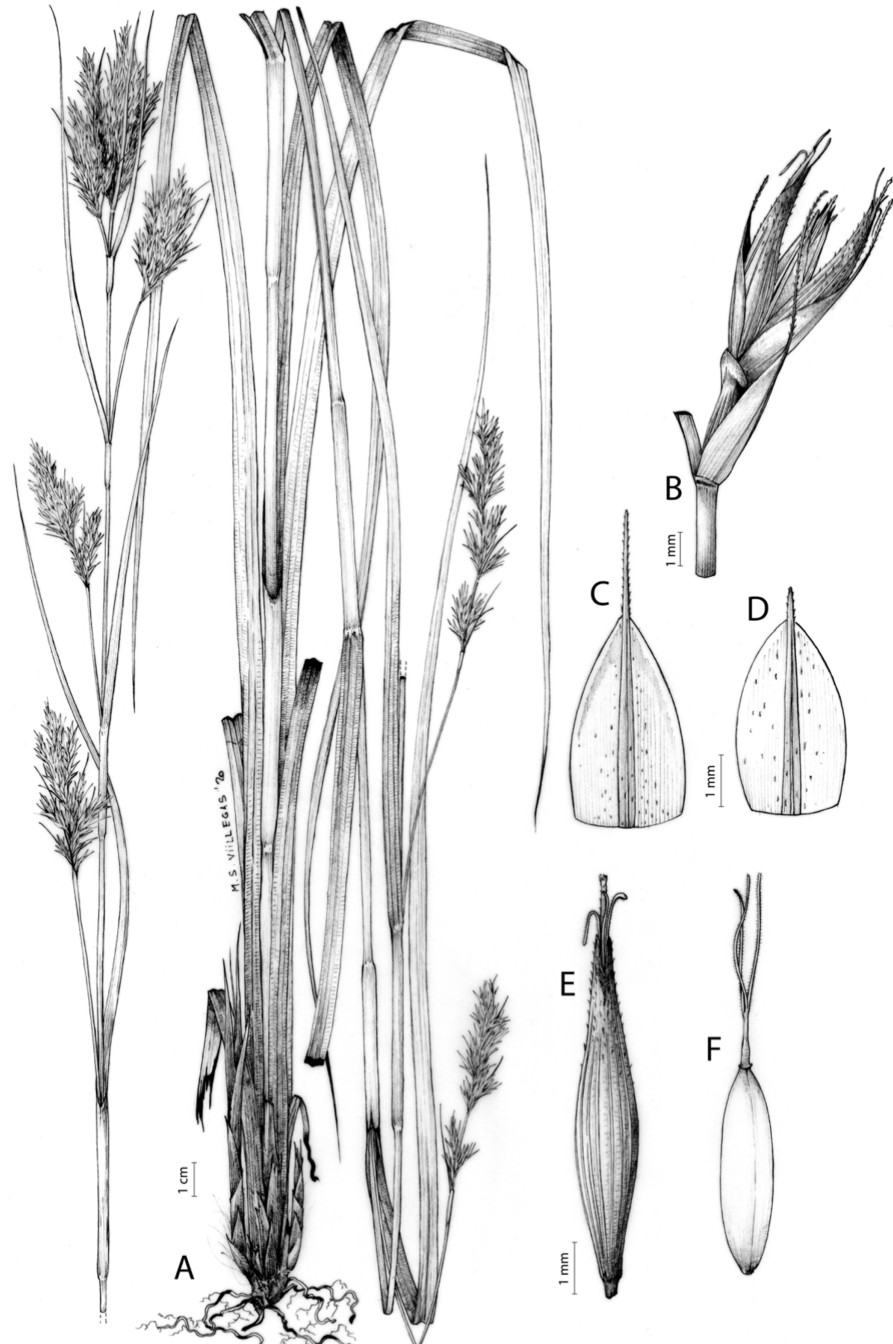

**Figure 22 Illustration of *Carex kukkoneniana*.** (A) General aspect of the plant and inflorescence. (B) Utriculiform cladoprophyll. (C) Female glume. (D) Male glume. (E) Unisexual utricle. (F) Achene. Illustration by M. Sánchez-Villegas.

**Iconography**

Figure 22. *Haines & Lye (1983*, sub *Schoenoxiphium rufum)*.

**Conservation**

Not evaluated (NE).

**Carex lancea** (Thunb.) Baill., Hist. Pl. 12: 341, 1894.

Type. [South Africa] Cap. b. Spei, *Thunberg s.n.* (lectotype: UPS-1350 digital image!, designated by *Kukkonen (1983)*).

≡ *Schoenus lanceus* Thunb., Prodr. Pl. Cap.: 17, 1794 [basionym.]
≡ *Schoenoxiphium capense* Nees, Linnaea 7: 533. 1832.
≡ *Schoenoxiphium lanceum* (Thunb.) Kük. in Engler (ed.), Pflanzenr. 38(IV, 20): 28, 1909.
≡ *Kobresia lancea* (Thunb.) Koyama, J. Fac. Sci. Univ. Tokyo, Sect. 3. Bot. 8: 80, 1961.

= *Schoenoxiphium meyerianum* Kunth, Enum. Pl. 2: 530. 1837.
Type: [South Africa] Cap. b. spei, *Drège s.n.* (lectotype: P-00461996 digital image! here designated; isolecto-: LE-00010148 digital image!).

= *Schoenoxiphium sickmannianum* Kunth, Enum. Pl. 2: 530. 1837.
Type: [South Africa] Cape b. spei, *Ecklon s.n.* "Sickmann in herb. Lucae." Not found.

Rhizome caespitose with more or less short internodes, stout, dark-brown. Flowering culms (50)60–110(170) cm long, erect, acutely trigonous, smooth, leafy up to half its length, (1)1.5–2 mm wide at the middle. Leaves (3)4–8(8.8) mm wide, shorter than the inflorescence, moderately coriaceous, green, flat in cross-section, but with an adaxial groove in the midrib area and trigonous to the apex, very scabrous along the edges and also along the abaxial midrib; adaxial surface smooth; usually curved at the apex; ligule 0.7–1.5(2) mm long. Basal sheaths entire to scarcely fibrous, with lamina. Lowest bract of the inflorescence leaf-like, shorter than the inflorescence length, with a sheath 40–60 mm long. Inflorescence occupying between ⅓ and ½ of the length of the culm, branching up to 4 times; partial inflorescences (8)11–14(16), the 3–7 lowest distant, pedunculate, nodding, with the part of the peduncle protruding from the sheath much shorter than the linear fertile part, the upper subsessile to shortly pedunculate and overlapping. Glumiform perigynia present, unisexual or bisexual. Glumiform cladoprophylls present. Tubular cladoprophylls present at the base of the distant partial inflorescences, hyaline. Utriculiform cladoprophylls absent. Male glumes 4.1–6.4(6.9) × (1.1)1.5–(2.2) mm, ovate-lanceolate to ovate, yellowish-brown to brown, with a green central band, acuminate or ending in an aculeate mucro or ariste up to 1 mm long. Female glumes 5–6.2 × 1.8–2.2 mm, lanceolate to ovate, straw-coloured to pale brown, with a green central band, acuminate, ending in an aculeate mucro up to 1(1.7) mm long. Unisexual utricles 5.5–6.1 × 0.7–1 mm, narrowly linear to narrowly oblong, stipitate, straight, straw-coloured to pale brown when mature, smooth, more rarely scabrous in the upper tiers, with numerous prominent veins across the entire surface, suberect, gradually attenuate in a more or less split, more or less scabrous beak up to 1.7 mm long; rachilla reaching to protruding from the apex up to 0.6 mm. Bisexual utricles absent. Achenes 3.5–4.5 × 0.6–1.2 mm, oblong-trigonous, pale to dark-brown when mature, tipped by a longly pyramidal, persistent style base.

**Distribution**
Endemic to Cape region, South Africa (Eastern and Western Cape provinces). [27 CPP].
Figure 15F.

**Habitat**
Edges of streams and other damp and shady places in forest (Afrotemperate, Subtropical and Azonal Forests Biome: Southern Afrotemperate Forest; Fynbos Biome: Sandstone Fynbos; Succulent Karoo Biome: Rainshadow Valley Karoo); 20–1,500 m.

**Etymology**
From the latin *lancea-ae*, spear, probably because of the shape of its broadly linear leaves.

**Iconography**
Figures 23 and 24A. *Kükenthal (1909*; sub *Schoenoxiphium lanceum*).

**Conservation**
Not considered of conservation concern at the national level in South Africa, and thus categorized as Least Concern (LC) (*SANBI, 2020*; sub *Schoenoxiphium lanceum*).

**Notes**
There are a number of potential syntypes of the name *Schoenoxiphium meyerianum* Kunth which do not fully match the lectotype label but could have been original material seen by Kunth (e.g. LE00010147 digital image!, P00461995 digital image!). Likewise, we have not been able to trace an adequate lectotype for the name *Schoenoxiphium sickmannianum* Kunth. A potential syntype ([South Africa, Cape Town, Mount Table] In umbrosis inter saxa altitud. 4 mont. tabul, *C.F. Ecklon 851*; HAL-0109862 digital image!, TUB007505 digital image!) which belongs to *C. lancea* is nonetheless cited by Kunth as doubtful.

**Carex ludwigii** (Hochst.) Luceño & Martín-Bravo, Bot. J. Linn. Soc. 179: 27, 2015.

≡ *Schoenoxiphium ludwigii* Hochst., Flora, 28: 764 (1845) [basionym].
Type: [South Africa] In Cap b. Spei, 1837, B. de Ludwig (lectotype: TUB-007504 digital image!, here designated; iso-: TUB-007503 digital image!, M-0110633 digital image!)

= *Schoenoxiphium rufum* Nees in Linnaea 10: 201 (1836).
Type: [South Africa] Ceded Territory, bei Phillipstown am Katrivier, 2,000–3,000′, October, leg. *Ecklon & Zeyher site 33* (see *Drège, 1847b*: 586) (lectotype: S-10-1122, designated by *Kukkonen (1983)* digital image!, isolecto-: G-00195305 digital image!).

≡ *Carex rufa* (Nees) Baill., Hist. Pl. 12: 340. 1894, nom. illeg.; non Lam. (1779).
≡ *Archaeocarex rufus* (Nees) Fedde & J.Schust., Just's Bot. Jahresber. 41(2): 7 (1913) (publ. 1918).
≡ *Kobresia rufa* (Nees) T.Koyama, J. Fac. Sci. Univ. Tokyo, Sect. 3, Bot. 8: 80 (1961).

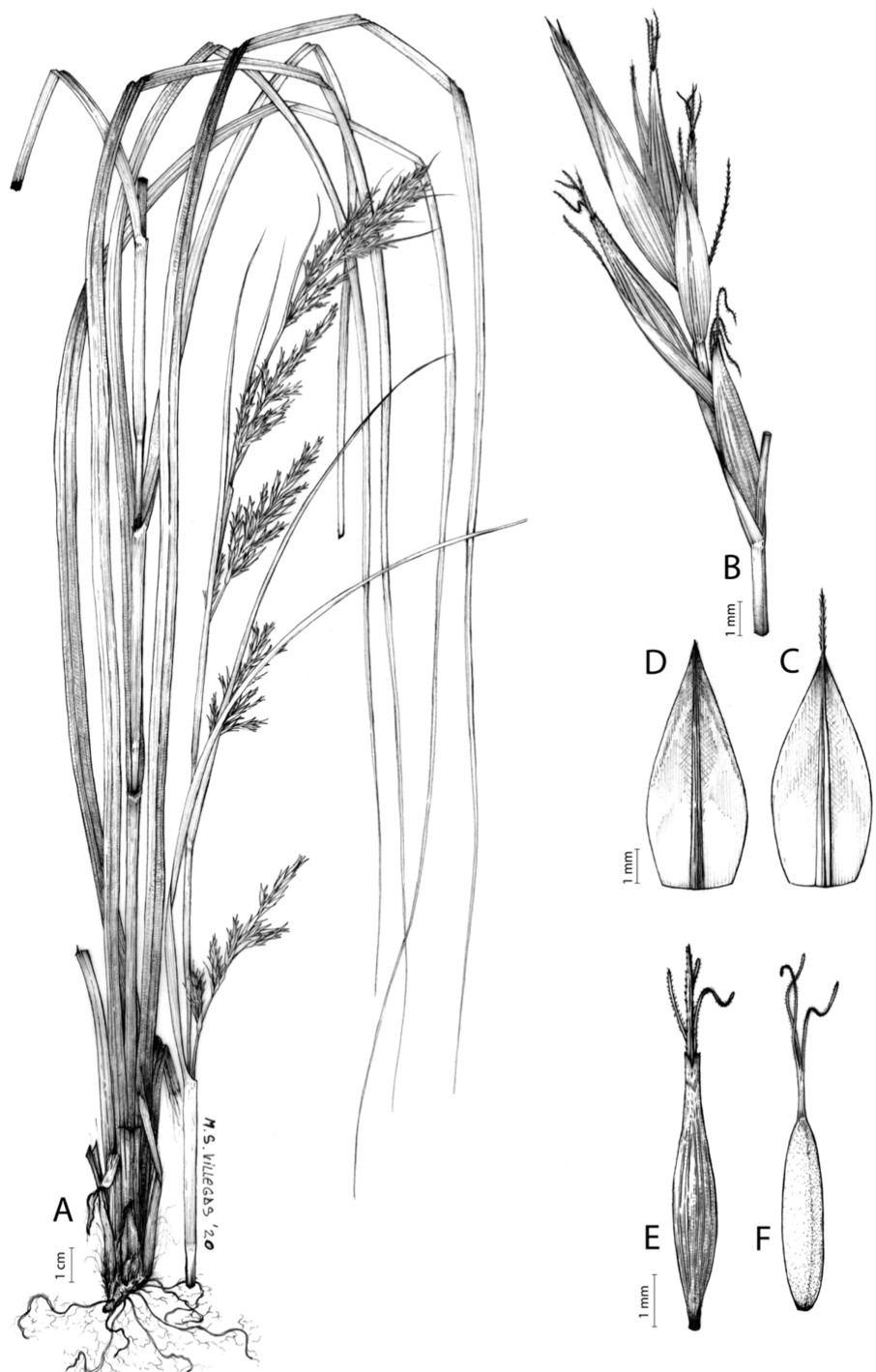

**Figure 23** **Illustration of *Carex lancea*.** (A) General aspect of the base of the plant and inflorescence. (B) Glumiform cladoprophyll axilating a male spikelet and three unisexual utricles. (C) Female glume. (D) Male glume. (E) Unisexual utricle. (F) Achene. Illustration by M. Sánchez-Villegas.

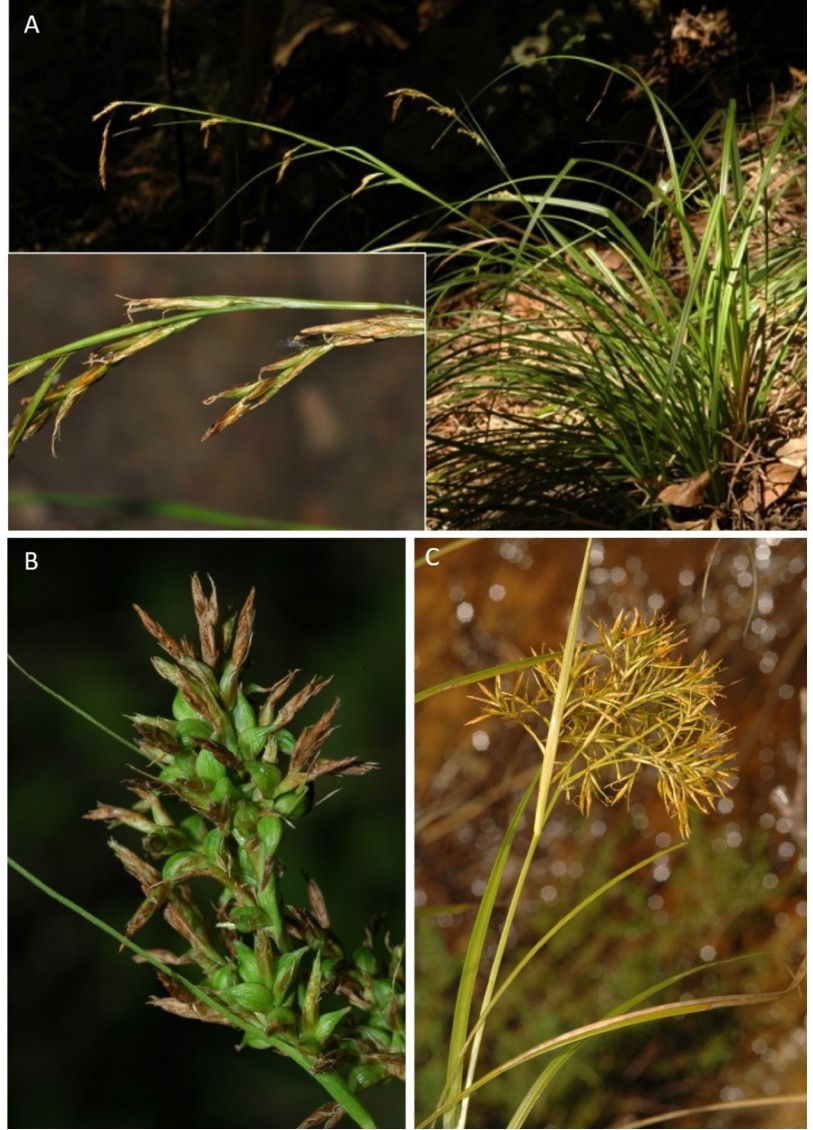

**Figure 24 Morphology of different species of *Carex* section *Schoenoxiphium*.** (A) *C. lancea*.
(B) *C. ludwigii*. (C) *C. multispiculata*. Photos by M. Luceño.

=*Schoenoxiphium dregeanum* Kunth, Enum. Pl. 2: 529 (1837); non *Carex dregeana* Kunth
(1837).
Type: [South Africa] Cap-de-Bonne-Espérance. Stormberg, 17-XII-1832, 5,000-6,000 ped.,
*Drège 7399a* (lectotype: P-00461997 digital image!, here designated; isolecto-: P00540803
digital image!). Syntypes: Afrique Australe, Wynberg, 5-V-1856, *Drège 7399c*, P-00461998
digital image!; Cap b. sp., *Drège s.n.*, LE-00010146 digital image!.
≡ *Schoenoxiphium rufum* var. *dregeanum* (Kunth) Kük., in Engler (ed.), Pflanzenr.
38(IV, 20): 30 (1909).

= *Schoenoxiphium rufum* var. *pondoense* Kük. in Engler (ed.), Pflanzenr. 38(IV, 20): 31 (1909).
Type. [South Africa] Pondoland, *Bachmann n. 116*. Not seen, originally probably at B, likely destroyed.

Rhizome more or less caespitose, with short or somewhat elongated internodes, stout, medium to dark-brown. Flowering culms (34)65–110(115) cm long, erect, obtusely trigonous, smooth, leafy up to half its length, (1.1)1.8–2.3(2.5) mm wide at the middle. Leaves (2.4)4.5–7.5(9.4) mm wide, shorter than the inflorescence, moderately rigid, more or less glaucous, flat to slightly V-shaped in cross-section, scabrous on the margins, except at the base, and along the abaxial midrib; adaxial surface scabrid at the apical parts; straight to curved at the apex; ligule (1)3–6(9.5) mm long. Basal sheaths with lamina or, more rarely, 1-few lowermost bladeless, entire or, more rarely, somewhat fibrous. Lowest bract of the inflorescence leaf-like, shorter than the inflorescence, rarely equaling it, with a sheath (20)29–59(66) mm long. Inflorescence up to the upper ⅓(⅔) of the length of the culm, branching up to 4 times; partial inflorescences (5)8–14, (2)4–5(6) lowermost distant, usually the 1(5) lowermost nodding. Glumiform perigynia and glumiform cladoprophylls absent, except for the occasional presence of some bisexual, glumiform perigynium. Tubular cladoprophylls always present. Utriculiform cladoprophylls usually present. Male glumes (2)2.2–4(4.5) × (0.6)1.3–2.5(2.9) mm, ovate, yellowish-brown to brown, with a green central band, ending in an aculeate mucro up to 1(1.6) mm long. Female glumes (1.7)2.3–3(4) × 1.7–2.1(3) mm, widely ovate to suborbicular, yellowish-brown to brown, with a green central band, ending in an aculeate mucro up to 2.5(2.9) mm long. Unisexual utricles (2.6)2.8–3.7(4.9) × 1.2–1.6(1.7) mm, broadly ovoid, stipitate, slightly to, more frequently, strongly curved, rarely straight, straw-coloured to brownish when mature, usually hispid in the upper third, with conspicuous prominent veins over the entire surface, suberect to erecto-patent, abruptly contracted into a bidentate, truncate to one-side shortly split beak 0.5-1.2 mm long; rachilla reaching to, rarely, protruding from the apex of the utricle by up to 0.5 mm. Bisexual utricles similar in shape to the unisexual ones, but with the apex widely and obliquely truncate or, less frequently, open almost to the base. Achenes 2–2.4(3.1) × (1)1.2–1.5(1.9) mm, ovoid-trigonous, sometimes with a deep, transverse groove in one side, straw-coloured to yellowish-brown when mature, tipped by a short, neck-like, more rarely subterete-trigonous, frequently asymmetric, persistent style base.

**Distribution**
Endemic to Lesotho (Berea, Leribe and Maseru districts) and South Africa (Eastern Cape, Free State, KwaZulu-Natal, Mpumalanga and Western Cape provinces). [27 CPP LES NAT OFS TVL]. Figure 15G.

**Habitat**
Edges of streams and other damp places in mountains (Grassland Biome: Drakensberg Grassland, Mesic Highveld Grassland and Subscarpment Grassland); 1,350–2,850 m.

**Etymology**

Named after Carl Ferdinand Heinrich von Ludwig (Baron von Ludwig) (1784–1847), German pharmacist, naturalist and businessman who started Cape Town's first botanic garden.

**Iconography**

Figures 24B and 25.

**Conservation**

The species has been evaluated as Least Concern (LC) at global level (IUCN, 2018). However, this was based on a distribution range that may likely include populations belonging to other species (*C. kukkoneniana*). It has also been evaluated as LC at the national level in South Africa (*SANBI, 2020*; sub *Schoenoxiphium rufum*).

**Notes**

When this species and *C. bolusii* C.B. Clarke are sympatric, it is possible to observe individuals showing intermediate morphological features between both taxa, probably due to hybridization. References to this species for the countries placed north of South Africa (see *Kukkonen, 1983*) belong to *C. kukkoneniana*.

There is a number of additional potential syntypes of *Schoenoxiphium dregeanum* Kunth that bear a handwritten label by this author, but whose indications (sometimes lacking any) are not a perfect match to the protologue (e.g. HAL-0109860 digital image!, LE-0010145 digital image!, S-08-6967 digital image!, TUB-007502 digital image!).

**Carex multispiculata** Luceño & Martín-Bravo, Bot. J. Linn. Soc. 179: 28, 2015.

≡ *Schoenoxiphium madagascariense* Cherm., Bull. Soc. Bot. France 70: 299. 1923; non *Carex madagascariensis* Boeckeler (1884).
Type: Madagascar. Mont Tsaratanana, dans les bruyères, 2,700 m, XII-1912, *Perrier de la Bâthie 2501* (lectotype: P-s.n. digital image!, here designated, isolecto- P-00540579!, see notes below).

Rhizome not caespitose, with long internodes, stout, dark-brown. Flowering culms (25)40–80(90) cm long, erect, acutely trigonous, smooth, leafy ⅔–⅘ of its length, (1.5)2–2.8(3.5) mm wide at the middle. Leaves (2.1)6.5–10(11) mm wide, longer than the inflorescence, rarely shorter, coriaceous, more or less glaucous, flat, slightly carinate or somewhat canaliculate in cross-section, scabrous along the edges and usually along the abaxial midrib; adaxial surface smooth or sparsely aculeate on the midrib in the upper third of its length; straight at the apex; ligule 0.1–0.4 mm long. Basal sheaths entire, with lamina or, rarely, the lowermost bladeless. Lowest bract of the inflorescence leaf-like, longer than the inflorescence length, rarely shorter, not sheathing or the lowermost with a sheath up to 40 mm long. Inflorescence branching up to 3(4) times, very dense, rarely somewhat lax, multispiculate, broadly ovoid to suborbicular, occupying up to the upper ⅕ of the of the culm, rarely up to ⅓, but then composed by 2–3 three suberect to erect, ovoid to suborbicular parts, one terminal and 1–2 lateral, long-pedunculate, distant parts; partial inflorescences (9)

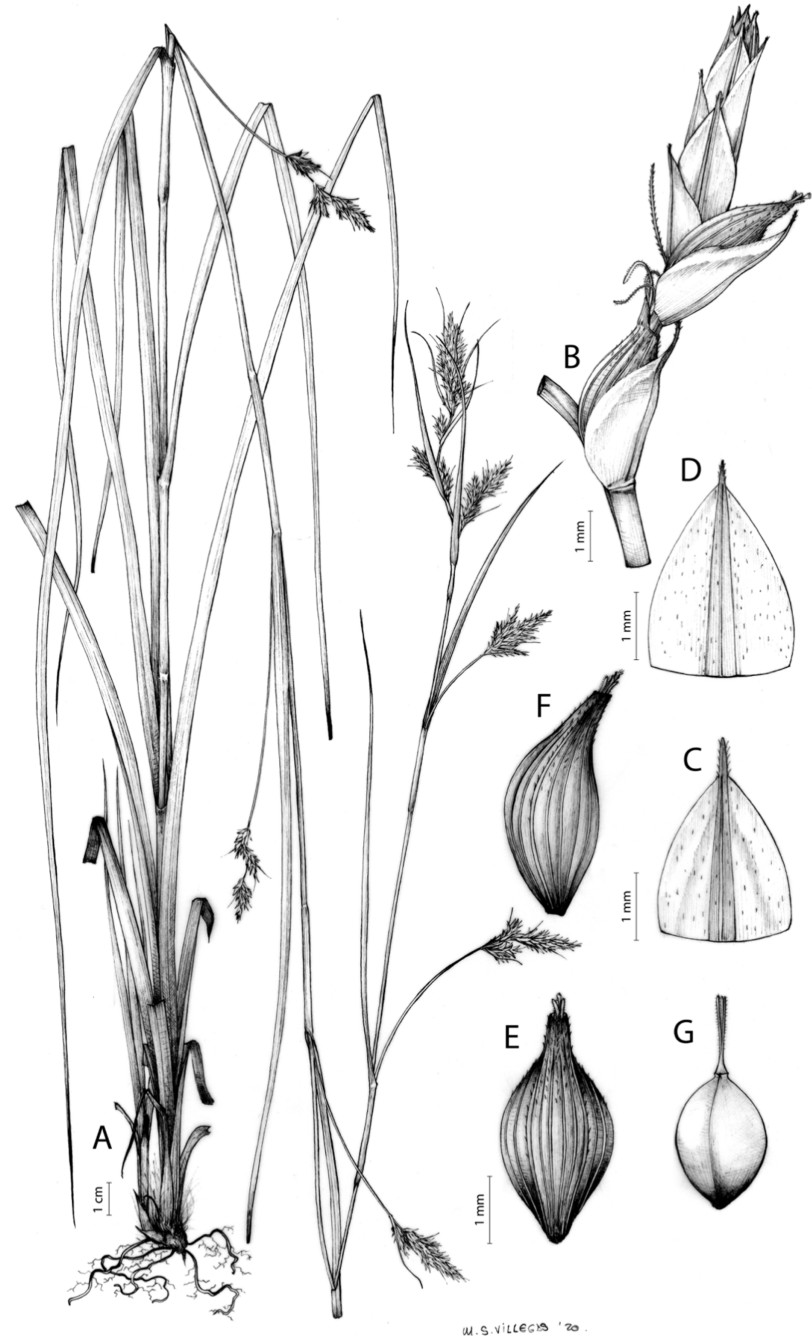

**Figure 25 Illustration of *Carex ludwigii*.** (A) General aspect of a fertile culm showing five nodding partial inflorescences. (B) Utriculiform cladoprophyll. (C) Female glume. (D) Male glume. (E) Abaxial view of an unisexual utricle. (F) Lateral view of an unisexual utricle. (G) Achene. Illustration by M. Sánchez-Villegas.

11–21(25), erect, subsessile, overlapping, or the 1–2 lowermost long-pedunculate and distant. Glumiform perigynia and glumiform cladoprophylls occasionally present. Tubular cladoprophylls always present at the base of the peduncles of the lowermost partial inflorescences, hyaline to brownish. Utriculiform cladoprophylls always present. Male

glumes 3–6 × (1)1.5–2 mm, ovate-lanceolate to ovate, yellowish-brown to brown, with a green central band, usually with an aculeate mucro or arista up to 1(1.5) mm long. Female glumes 3–6.5 × 1.8–2.3 mm, ovate, predominantly brownish, with a green central band, ending in an aculeate mucro up to 2(6) mm long. Unisexual utricles, when present, 4.4–6.5 × 0.6–0.8 mm, narrowly linear, straw-coloured, yellowish-brown to subhyaline when mature, stipitate, straight to arcuate, smooth, with numerous prominent veins across the entire surface, suberect to erecto-patent, gradually attenuated into an obliquely truncate, irregular to slightly bifid beak up to 2(2.5) mm long; rachilla protruding from the apex up to 5(8) mm long. Bisexual utricles widely and obliquely truncate and more or less split at the apex. Achenes 3–5.5 × 0.5–0.7 mm, oblong-trigonous, yellowish-brown when mature, tipped by a shortly trigonous to pyramidal, persistent style base.

## Distribution
Eswatini (Hhohho district), Madagascar (Tsratanana mountains) and South Africa (KwaZulu-Natal, Free State and Mpumalanga provinces). [27 NAT OFS TVL SWZ 29 MDG]. Figure 26A.

## Habitat
Edges of stream and waterfalls, damp meadows, and other damp places in mountains (Grassland Biome, Drakensberg Grassland, Mesic Highveld Grassland and Subscarpment Grassland); 1,100–2,800 m.

## Etymology
From the Latin *multus-a-um*, many, and *spicula-ae*, spikelet, because of its very branched inflorescence which is composed of numerous spikelets.

## Iconography
Figures 24C and 27.

## Conservation
Not considered of conservation concern at the national level in South Africa, and thus categorized as Least Concern (LC) (*SANBI, 2020*; sub *Schoenoxiphium madagascariense*).

## Notes
Two vouchers are kept in P with the number 2501 of Perrier de la Bâthie, one of them (P-s. n.) contains a complete and typical specimen of the species including two fertile culms, while the other (P-00540579) includes a less typical individual with just a fertile culm, so we have designated as a lectotype the first of them.

**Carex perdensa** (Kukkonen) Luceño & Martín-Bravo, Bot. J. Linn. Soc. 179: 28, 2015.

≡ *Schoenoxiphium perdensum* Kukkonen, Bot. Not. 131: 265. 1978 [basionym]. Type: South Africa. [Eastern] Cape Prov., Distr. King William's Town, Keiskama Hoek, near Ghulu Kop, 4,000 ft, XII-1925, *R.A. Dyer 245a* (holotype: K-000693793 digital image!; iso-: K-000693794 digital image!, PRE-0106552 digital image!).

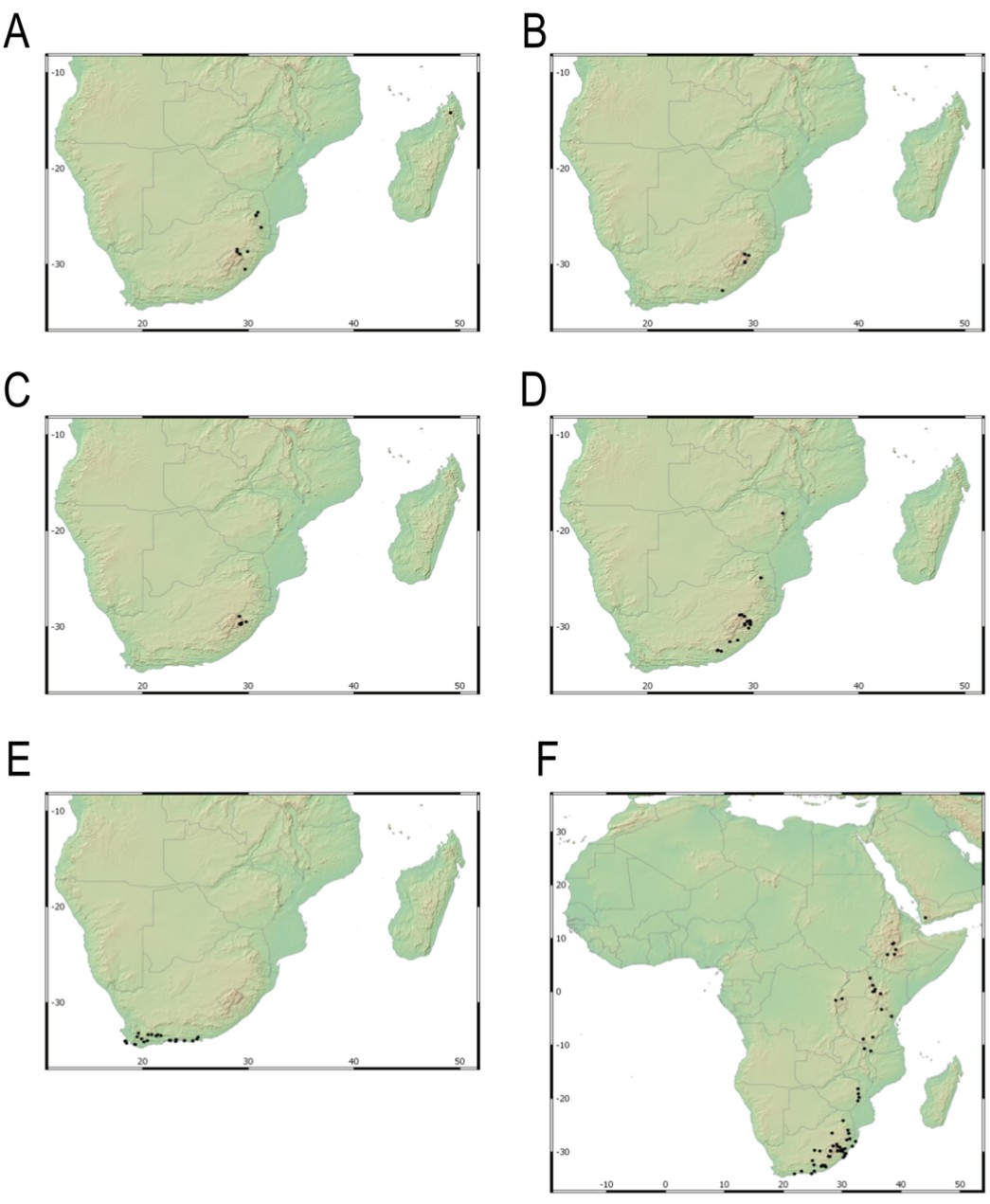

**Figure 26 Distribution map of different species of *Carex* section *Schoenoxiphium*.** (A) *C. multi-spiculata* (B) *C. perdensa*. (C) *C. pseudorufa*. (D) *C. schweickerdtii*. (E) *C. sciocapensis*. (F) *C. spartea*. Maps created with QGIS.

Rhizome densely caespitose, with very short internodes, slender, dark-brown. Flowering culms 7.5–15(25) cm long, obtusely trigonous, smooth, leafy only at base, 0.4–0.5(0.7) mm wide at the middle. Leaves (0.2)0.3–0.5(1.5) mm wide, shorter or longer than the inflorescence, scarcely rigid, green to slightly glaucous, involute or canaliculate, scabrous along the edges and the upper part of the abaxial midrib; adaxial surface smooth to somewhat scabrous to the apical part; ligule 0.3–1 mm long. Basal sheaths entire to scarcely fibrous, lowermost bladeless, uppermost with lamina. Lowest bract leaf-like, usually longer

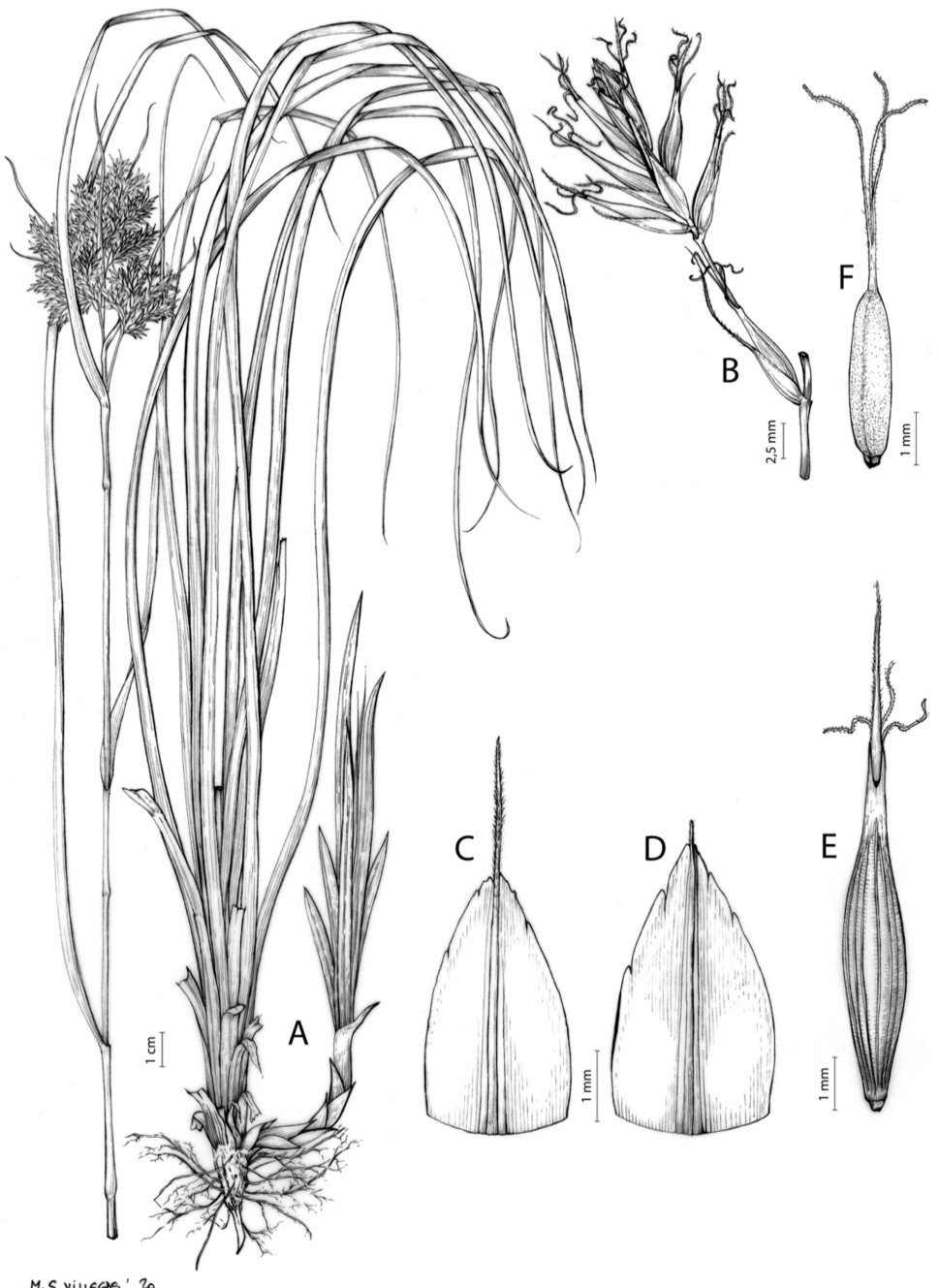

**Figure 27 Illustration of *Carex multispiculata*.** (A) General aspect of the base of the plant and inflorescence. (B) Utriculiform cladoprophyll. (C) Female glume. (D) Male glume. (E) Unisexual utricle. (F) Achene. Illustration by M. Sánchez-Villegas.

than the inflorescence, more rarely shorter, with a sheath 7–12 mm long. Inflorescence up to ¾ of the length of the culm, branching up to 2 times; partial inflorescences (2)3–4(5), very lax, bearing up to 2(3) female flowers, erect, distant, pedunculate. Glumiform perigynia and glumiform cladoprophylls absent. Tubular cladoprophylls present at the base of partial inflorescences, hyaline. Utriculiform cladoprophylls absent. Male glumes

2–3.2(3.6) × 1.2–1.5 mm, lanceolate to ovate, yellowish-brown, with a green central band, ending in an aculeate mucro up to 1.2 mm long. Female glumes (1.9)2.2–2.8(3.5) × 1.5–2 mm, widely ovate to suborbicular, predominantly yellowish-brown, with a green central band, ending in an aculeate mucro up to 1.2(2.8) mm. Unisexual utricles 2.6–3.6 × 1.4–1.9 mm, widely ovoid to widely ellipsoid, stipitate, straight to weakly curved, straw-coloured to yellowish-brown when mature, with numerous, more or less prominent veins across the entire surface, erect to erecto-patent, abruptly contracted into a smooth, bidentate to irregular beak 0.5–1 mm long; rachilla reaching the apex to protruding from it up to 0.7 mm. Bisexual utricles absent. Achenes 1.8–2.8 × 1–1.4 mm, widely ellipsoid-trigonous, straw-coloured to dark-brown when mature, tipped by a short obtusely trigonous, asymmetric, persistent style base.

**Distribution**
Endemic to South Africa (Eastern Cape and KwaZulu-Natal provinces). [27 CPP NAT]. Figure 26B.

**Habitat**
Grassland (Grassland Biome: Drakensberg Grassland); 1,200–1,950 m.

**Etymology**
From the Latin prefix *per*, very, and adjective *densus-a-um*, dense, because of the densely caespitose habit of this plant.

**Iconography**
Figures 28 and 29A. *Kukkonen (1978)*.

**Conservation**
Not considered of conservation concern at the national level in South Africa, and thus categorized as Least Concern (LC) (*SANBI, 2020*; sub *Schoenoxiphium perdensum*).

**Carex pseudorufa** Luceño & Martín-Bravo, Bot. J. Linn. Soc. 179: 28, 2015.

≡ *Schoenoxiphium burttii* Kukkonen, Notes Roy. Bot. Gard. Edinburgh 43: 365. 1986; non *Carex burttii* Noltie (1993).

Type: South Africa. KwaZulu-Natal, Underberg distr., 2929 CB, Chameleon area, c. 5 miles N of Castle View farm, 6,700 ft, forming loose clumps at streamside, 2-XII-1984, *Hilliard & Burtt 17812* (lectotype: E-00200232 digital image! here designated (see notes); isolecto-: GENT-0000090032820 digital image!, GENT-0000090032844 digital image!, GENT-0000090032943 digital image!, NU-0015687 digital image!).

Rhizome caespitose, with short internodes, very stout, pale-brown to dark-brown. Flowering culms (55)72–111(118) cm long, obtusely trigonous, smooth, leafy up to half its length, (2.3)2.5–4.3(5) mm wide at the middle. Leaves (5.5)8.3–13.9(17) mm wide, shorter than the inflorescence, moderately rigid, straight at the apex, light green, flat in cross-section, slightly scabrous along the margins and usually along the upper half of the abaxial midrib; adaxial surface usually finely papillose near de margins; ligule (1)3–6(8) mm long.

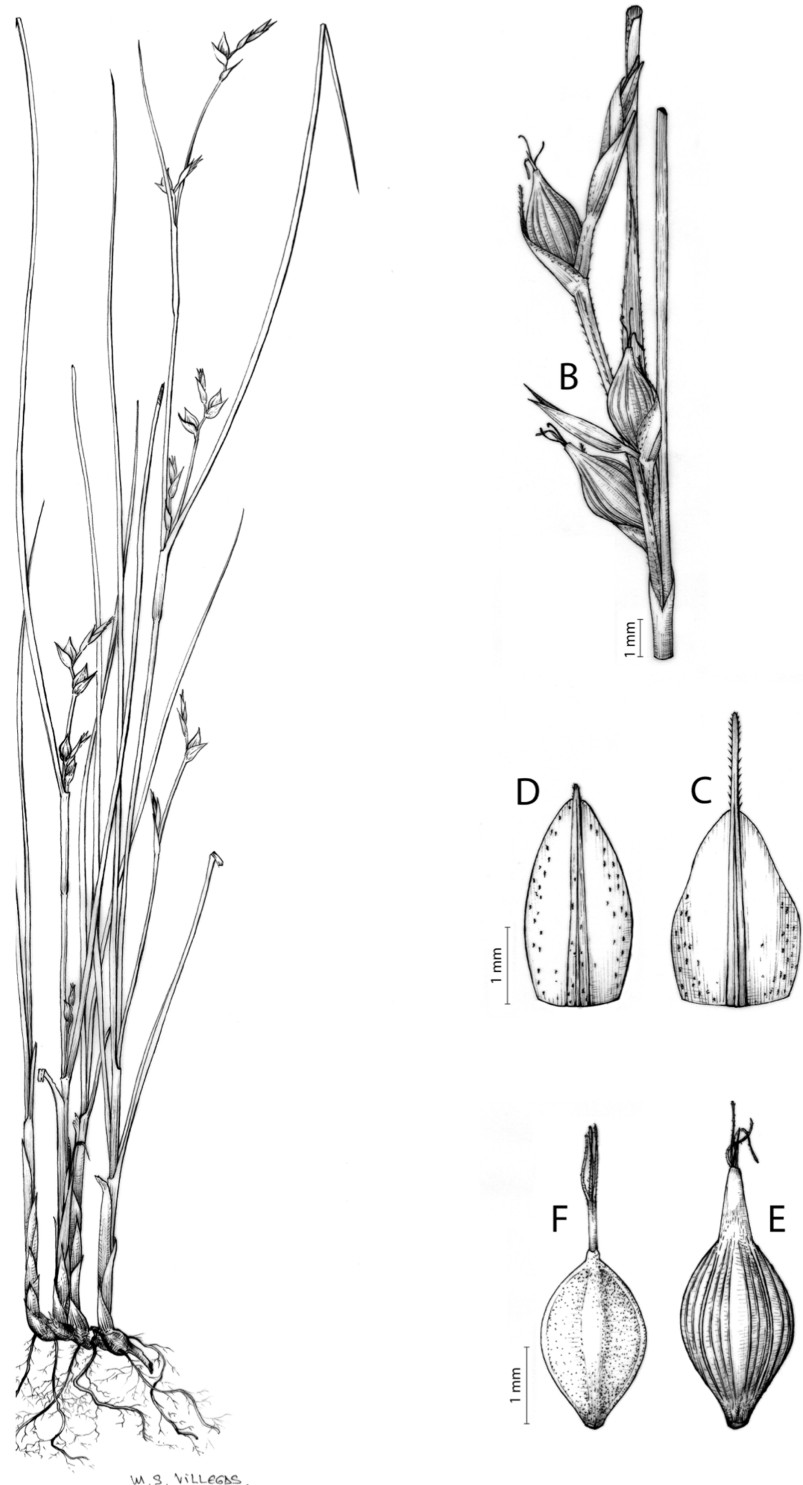

**Figure 28 Illustration of *Carex perdensa*.** (A) General aspect of the plant. (B) Partial inflorescence. (C) Female glume. (D) Male glume. (E) Unisexual utricle. (F) Achene. Illustration by M. Sánchez-Villegas.

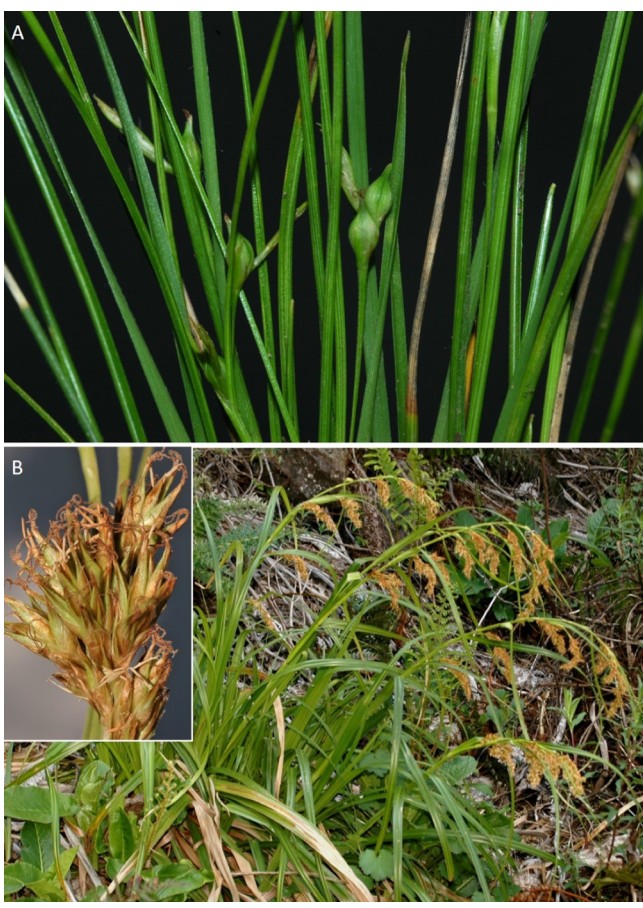

**Figure 29 Morphology of different species of *Carex* section *Schoenoxiphium*.** (A) *C. perdensa*. (B) *C. pseudorufa*. Photos by M. Luceño.

Basal sheaths with lamina, entire or, rarely, the lowermost somewhat fibrous. Lowest bract of the inflorescence leaf-like, from half of the inflorescence length to, more rarely, equaling it, with a sheath (37)39–69 mm long. Inflorescence up to the upper ¾ of the length of the culm, branching up to 4 times; partial inflorescences 7–13, very dense, ovoid to ellipsoid, the (3)4–6 lowermost distant and nodding, the uppermost overlapping, nodding to suberect. Glumiform perigynia and glumiform cladoprophylls absent. Tubular cladoprophylls always present. Utriculiform cladoprophylls absent. Male glumes (2.8)3.3–4.7(5.3) × (1)1.4–1.9(2.2) mm, ovate-lanceolate to ovate, brown to reddish brown, with a green central band, ending in an aculeate mucro up to 0.8 mm long. Female glumes (2.7)3–4.5(5.5) × 1.7–2.7(3,8) mm, widely ovate to suborbicular, brown to reddish brown, with a green central band, ending in a prominent aculeate mucro up to 1.1 mm long. Unisexual utricles (5.3)6.5–8(8.4) × (1.1)1.3–1.7(1.8) mm, linear, lancolate or narrowly ellipsoid, stipitate, straight to slightly arcuate, straw-coloured to reddish brown, sometimes red-spotted when mature, dispersely aculeate to hispid in the upper third, with conspicuous prominent veins over the entire surface, suberect to erecto-patent, gradually attenuate, rarely somewhat contracted, into an asymmetrically bidentate to bifid or irregular beak 1.3–3.2 mm long; rachilla usually reaching the apex of the utricle, rarely

protruding from it up to 0.5 mm. Bisexual utricles usually present, widely and obliquely truncate at the apex. Achenes (3.6)3.7–4.6(4.9) × (0.9)1.1–1.4(1.5) mm, more or less oblong-trigonous, straw-coloured to yellowish-brown when mature, tipped by a very short, obtusely trigonous, frequently somewhat asymmetric, persistent style base.

### Distribution
Endemic to the eastern slopes of the Drakensberg range in KwaZulu-Natal (South Africa). [27 NAT]. Figure 26C.

### Habitat
Edges of streams and wet meadows (Grassland Biome: Drakensberg Grassland); 1,850-3,200 m.

### Etymology
From the Greek ψευδής (pseudo, resembling but not equalling) and the Latin *rufus-a-um* (red), alluding to the resemblance of this species to *Carex ludwigii*, which was formerly known as *Schoenoxiphium rufum*.

### Iconography
Figures 29B and 30.

### Conservation
Considered as Least Concern (LC) at the national level in South Africa (*SANBI, 2020*; sub *Schoenoxiphium burttii*).

### Notes
*Kukkonen (1986)* indicated that the holotype of *S. burttii* was at H, but a thorough search in this herbarium (H. Väre, 2021, personal communication) has revealed that the specimen is lost, so the isotype at E is selected as the lectotype.

**Carex schweickerdtii** (Merxm. & Podlech) Luceño & Martín Bravo, Bot. J. Linn. Soc. 179: 28, 2015.

≡ *Schoenoxiphium schweickerdtii* Merxm. & Podlech, Mitt. Bot. Staatssamml. München 3: 529, 1960 [basionym].
Type: South Africa. Drakensberge-Mariepskop, Gipfelfluren, 2,000 m, 5-XII-1957, *Merxmüller 590* (holotype: M-0110634 digital image!; iso-: PRE-0107827 digital image!).

Rhizome caespitose, with very short internodes, stout, pale-brown to dark-brown. Flowering culms (25)30–90(130) cm long, erect, acutely trigonous, smooth or, rarely, very dispersely aculeate to the apex, leafy up to ⅓(⅔) of its length, 2–3(3.8) mm wide at the middle. Leaves (4.7)6.5–11.5(18) mm wide, shorter than the inflorescence, more rarely longer, quite rigid, light green, plicate in cross-section, scabrous along the edges and usually also along the abaxial midrib; adaxial surface smooth to scabrid on the keels at the apex; straight at the apex; ligule up to 2 mm long. Basal sheaths with lamina or the lowermost bladeless, entire to somewhat fibrous. Lowest bract of the inflorescence leaf-like, longer than the inflorescence, with a sheath (18)20–40(84) mm long, distichous and hiding

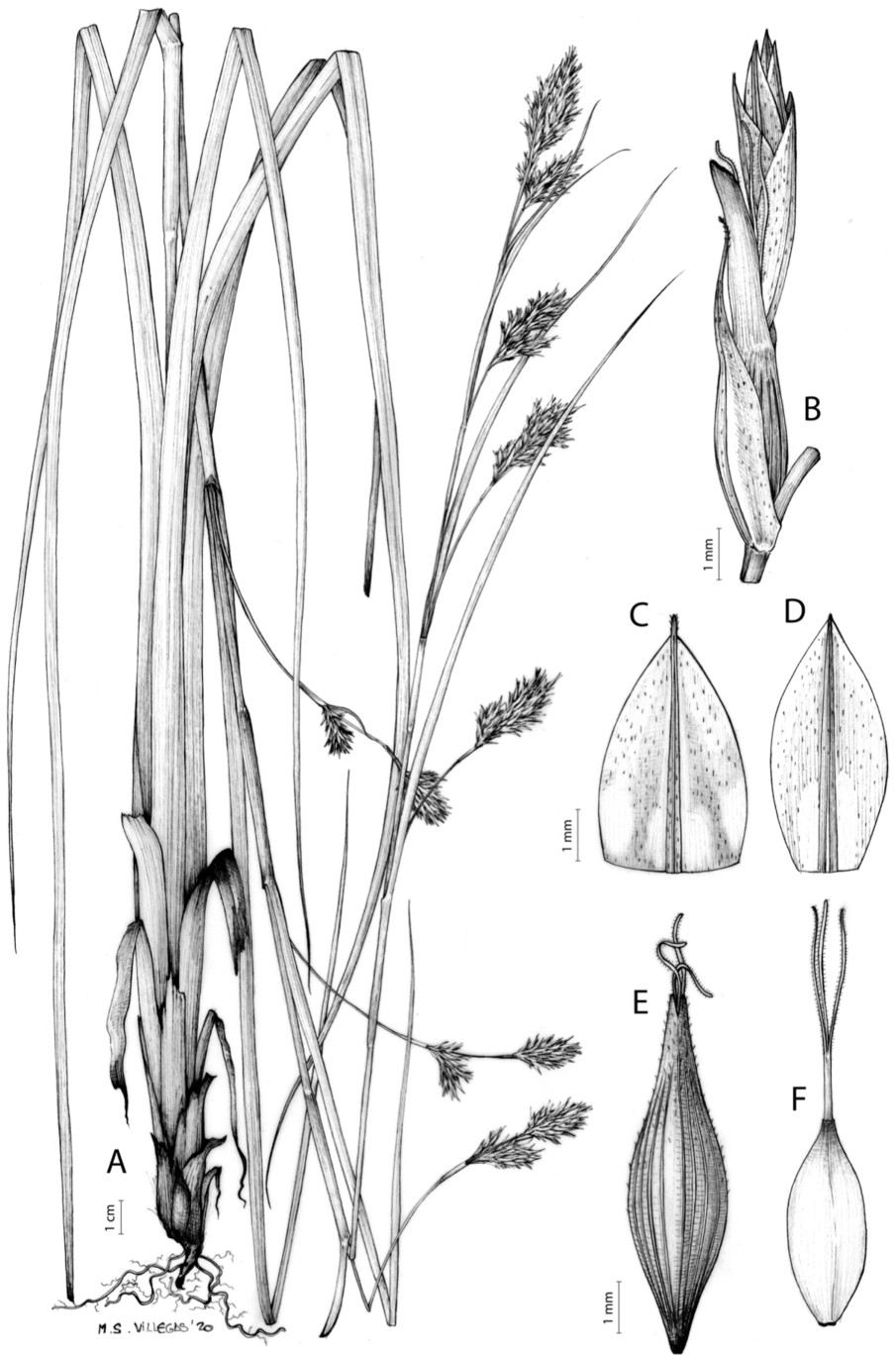

**Figure 30  Illustration of *Carex pseudorufa*.** (A) General aspect of the plant showing several nodding partial inflorescences. (B) Bisexual utricle. (C) Female glume. (D) Male glume. (E) Unisexual utricle. (F) Achene. Illustration by M. Sánchez-Villegas.

the inflorescence when young. Partial inflorescences 6–10(14), erect, branching up to 3(4) times, lowermost pedunculate and more or less distant, and uppermost subsessile. Glumiform perigynia usually absent. Glumiform cladoprophylls present at the base of the second order branches. Tubular cladoprophylls always present at the base of the lower

partial inflorescences, sometimes split almost to the base. Utriculiform cladoprophylls rare, more or less split. Male glumes (4)5–7 × (1)1.7–3 mm, ovate-lanceolate to elliptic, yellowish-brown to brown, with a green central band, acute or, more frequently, ending in an aculeate mucro or arista up to 0.7 mm long. Female glumes (5)6.5–8.6(9) × (1.5)2–2.5(3) mm, lanceolate, ovate or elliptic, brown to yellowish, with a green central band, ending in a smooth to slightly aculeate mucro up to 2(3.6) mm long. Unisexual utricles (6)7–9.5(10) × 0.9–1.1 mm, narrowly linear, straw-coloured to brownish when mature, stipitate, straight, smooth, with numerous prominent veins across the entire surface, erect, gradually attenuated into an irregular beak up to 3 mm long, rarely split almost to the base of the utricle (glumiform perigynium); rachilla usually protruding from the apex up to 1(1.5) mm. Bisexual utricles widely and obliquely truncate at the apex. Achenes (4.2)4.8–6.1 × 0.8–1 mm, oblong-trigonous, straw-coloured to dark brown when mature, tipped by an obtusely trigonous to pyramidal, persistent style base.

### Distribution
Lesotho (Butha Buthe district), South Africa (Eastern Cape, KwaZulu-Natal and Mpumalanga provinces), and Zimbabwe (Manicaland province). [26 ZIM 27 CPP LES NAT TVL]. Figure 26D.

### Habitat
Stony meadows and, more rarely, edges of streams and pools in mountains (Grassland Biome: Drakensberg Grassland and Mesic Highveld Grassland); 1,400–3,100 m.

### Etymology
Named after Herold Georg Wilhelm Johannes Schweickerdt (1903–1977), German botanist, who collected an important number of plants in South Africa, mainly in Mpumalanga and KwaZulu-Natal, but also in Mozambique and Zimbabwe.
The herbarium of the Pretoria University (PRU) was named in his honour.

### Iconography
Figs. 31 and 32A.

### Conservation
Not considered of conservation concern at the national level in South Africa, and thus categorized as Least Concern (LC) (*SANBI, 2020*; sub *Schoenoxiphium schweickerdtii*).

**Carex sciocapensis** Luceño, Márquez-Corro & Sánchez-Villegas, *nom. nov.*

≡ *Schoenoxiphium altum* Kukkonen, Notes Roy. Bot. Gard. Edinburgh 43: 365. 1986; non *C. alta* Boott. (1845).
Type: South Africa. Cape, George div., Saalsveld, Groeneweidebos, 300 m, 02-III-1982, *Geldenhuys 622* (lectotype: PRE-0812903-0 digital image!, here designated, see notes).

Rhizome not or laxly caespitose, with long internodes, moderately stout, medium to dark-brown. Flowering culms (25)37–68(96) cm long, obtusely trigonous, smooth, leafy usually up to the upper third of its length, (0.7)0.8–1.7(2.1) mm wide at the middle. Leaves

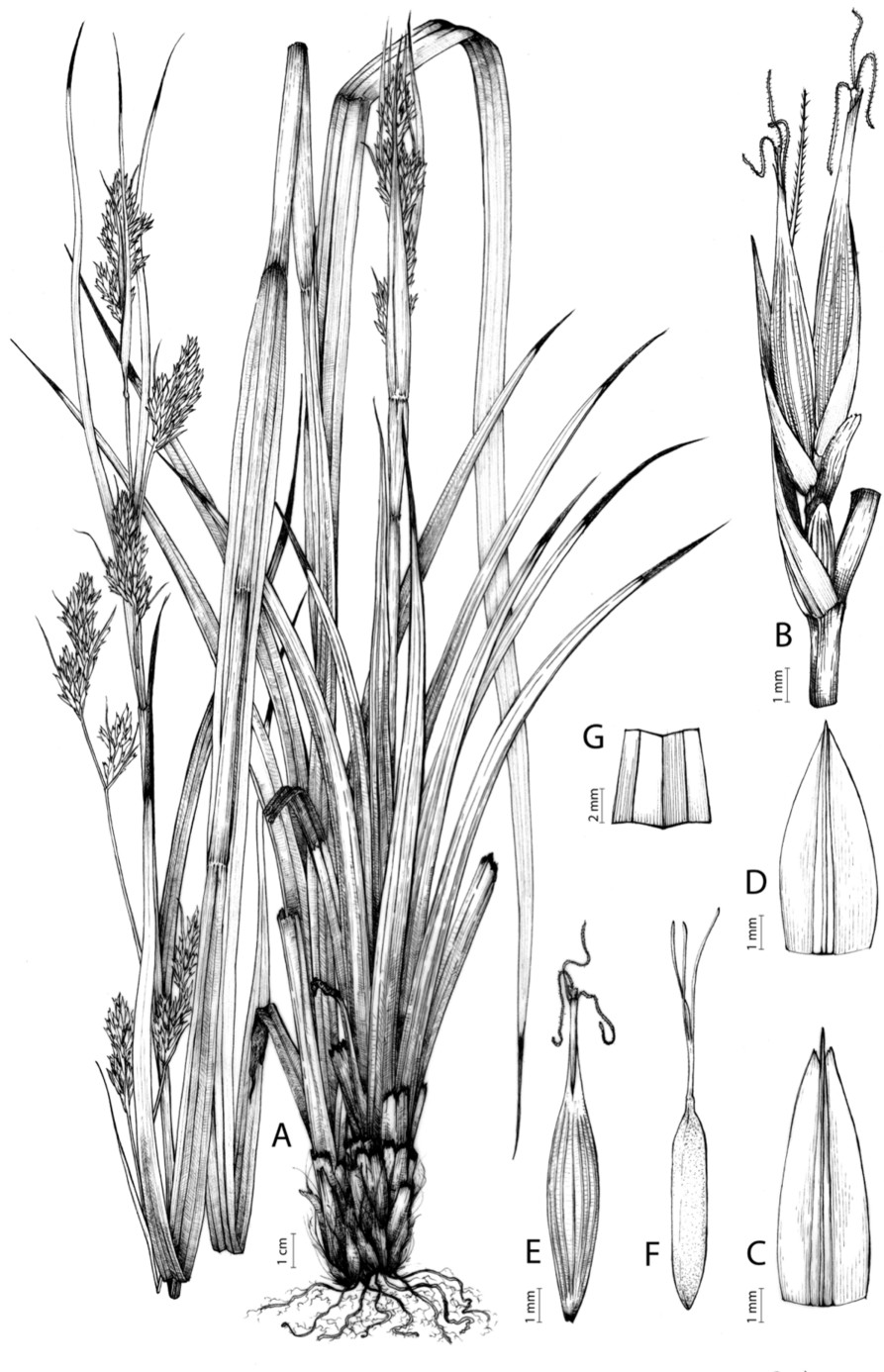

**Figure 31 Illustration of *Carex schweickerdtii*.** (A) General aspect of the plant showing one mature fertile culm (left) and one young fertile culm (right). (B) Utriculiform cladoprophyll. (C) Female glume. (D) Male glume. (E) Unisexual utricle. (F) Achene. (G) Fragment of a leaf. Illustration by M. Sánchez-Villegas.

(1)2.1–4.6(6) mm wide, shorter than the inflorescence, soft to scarcely rigid, light green to somewhat glaucous, flat in cross-section, finely papillose to slightly scabrous along the edges, and sometimes with a few prickles along the abaxial midrib; adaxial surface smooth;

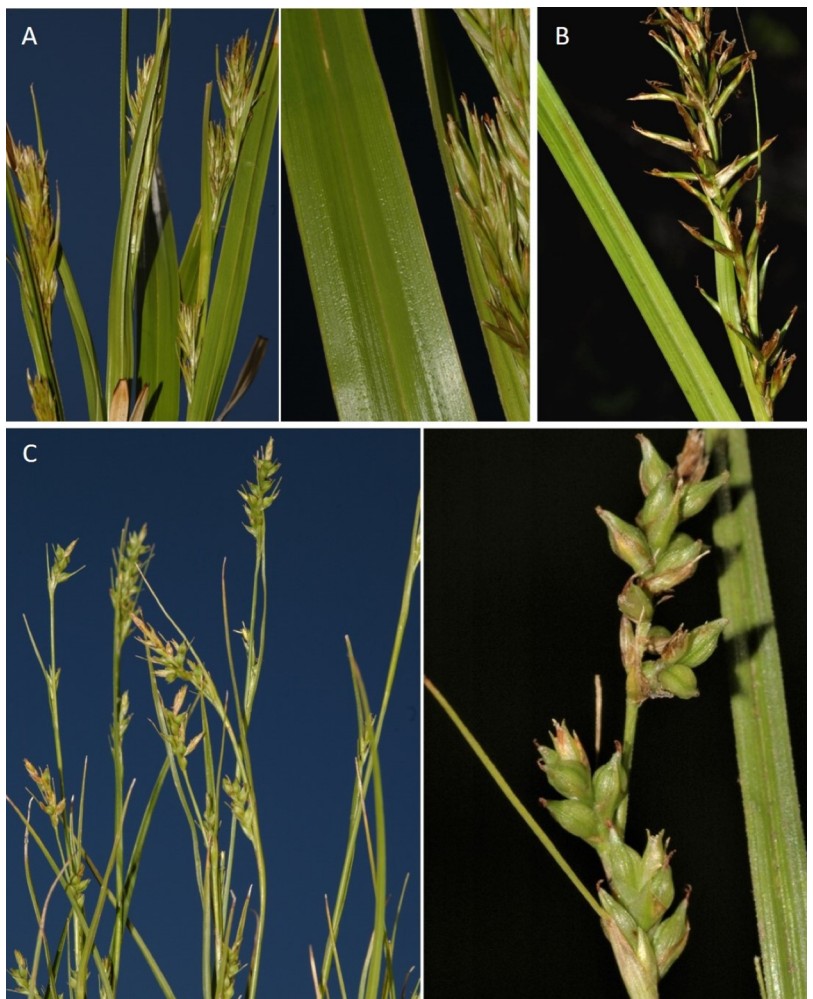

**Figure 32 Morphology of different species of *Carex* section *Schoenoxiphium*.** (A) *C. schweickerdtii.* (B) *C. sciocapensis.* (C) *C. spartea.* Photos by M. Luceño.

straight to somewhat curved at the apex; ligule (1.1)1.6–3.8(6) mm long. Basal sheaths usually bladeless, not or slightly fibrous. Lowest bract of the inflorescence leaf-like, equaling or longer than the inflorescence length, not sheathing. Inflorescence oblong, rarely ovoid, branching up to three times; partial inflorescences (3)4–7, somewhat distant to overlapping, sessile. Glumiform perigynia and glumiform cladoprophylls absent. Tubular cladoprophylls always present, hyaline. Utriculiform cladoprophylls usually present in the lower and central parts of developed inflorescence. Male glumes (3.2)4.5–6.5(7.1) × (1.4)1.6–2(2.2) mm, oblong-lanceolate to narrowly elliptical, light to yellowish-brown with a green central band, acute to shortly acuminate. Female glumes (3.9)4.2–6.8(9.2) × (1.7)1.8–2.4(2.8) mm, ovate to elliptical, yellowish to dark brown with a green central band, obtuse to acute. Unisexual utricles (6.1)6.4–8.9(10.1) × (0.7)0.9–1.2(1.4) mm, linear-oblong to very narrowly lanceolate in outline, long-stipitate,

straight, straw-coloured to yellowish-brown when mature, smooth or very dispersely and shortly aculeate at the apex, with numerous prominent veins across the entire surface, suberect to patent, gradually attenuated into an obliquely trucate to slightly bifid beak up to 1.5 mm long; rachilla reaching the utricle or protruding from it by up to 6 mm or, more rarely, protruding from the mouth and bearing some small, sterile glumes at the top, but then, beak wide and obliquely truncate. Bisexual utricles widely and obliquely truncate at the apex. Achenes (3.9)4–5(5.4) × 0.7–1(1.1) mm, oblong-trigonous, light to dark brown when mature, tipped by a very short, obtusely trigonous, persistent style base.

## Distribution
Endemic to Eastern and Western Cape provinces (South Africa). [27 CPP]. Figure 26E.

## Habitat
Shady places, mainly in margins and clearing of southern afrotemperate forest, but also in fynbos areas, especially on shale, dolerite and sandstone soils (Afrotemperate Forest Biome: Southern Afrotemperate Forest; Fynbos Biome: Sandstone Fynbos; and Succulent Karoo Biome: Rainshadow Valley Karoo); 10–1,900 m.

## Etymology
From the Greek σκῐᾰ́ (skiā́), shade, and the Latin *capens-is*, from Cape region, in reference to the fact that it grows in shady areas of humid forests and its similarity to *C. capensis*.

## Iconography
Figures 32B and 33.

## Conservation
It has been considered Near Threatened (NT) at the national level in South Africa due to its restricted range (with a substantial portion of its habitat lost in the recent past because of exploitation of native forests and timber plantations) and competition with alien invasive plants (*SANBI, 2020*; sub *Schoenoxiphium altum*).

## Notes
This species is very closely related to *C. capensis*; in fact, both taxa are scarcely differentiated from a molecular point of view (see Phylogenetic section and Fig. 4); however, most individuals of both taxa are easily separated by its morphological features (see descriptions of both species). Nevertheless, specimens exhibiting intermediate morphology and ecology can be found, mainly in mountain areas, but also in Cape Peninsula, where some intermediate specimens and similar forms to *C. sciocapensis* inhabit in sunny places. Deeper systematic studies are needed to determine the existence of one or more cryptic species and to check if hybridization processes are taking place between these two taxa.

*Kukkonen (1986)* indicated that the holotype of *S. altum* was at H, but a thorough search in this herbarium (H. Väre, 2021, personal communication) has revealed that the specimen is lost, so the isotype at PRE is selected as the lectotype.

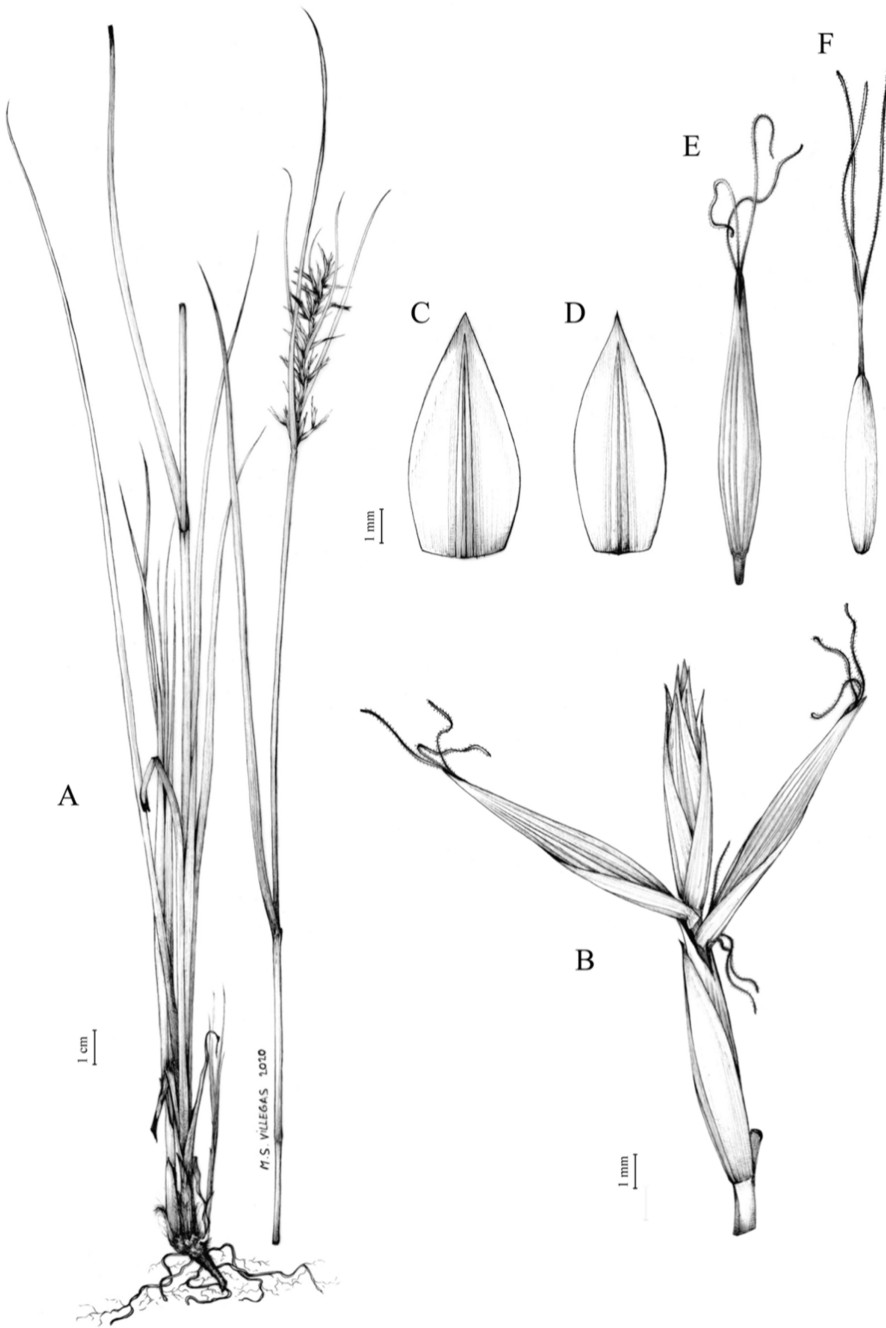

**Figure 33 Illustration of *Carex sciocapensis*.** (A) General aspect of the base of the plant and fertile culm showing the inflorescence. (B) Utriculiform cladoprophyll. (C) Female glume. (D) Male glume. (E) Unisexual utricle. (F) Achene. Illustration by M. Sánchez-Villegas.

**Carex spartea** Wahlenb., Kongl. Vetensk. Acad. Nya Handl. 149, 1803.

Type: [South Africa] Cap. bonae Spei, *Thunberg* (lectotype: UPS-21933 specimen on the right, digital image! here designated; isolecto- UPS-21933 specimen on the left, digital image!, SBT-10980 digital image!).

≡ *Schoenoxiphium sparteum* (Wahlenb.) C.B. Clarke, Bull. Misc. Inform. Kew, Addit. Ser. 8: 67 (1908).

≡ *Archaeocarex spartea* (Wahlenb.) Pissjauk. Bot. Mater. Gerb. Bot. Inst. Komarova Akad. Nauk S.S.S.R. 12: 83 (1950).

≡ *Kobresia spartea* (Wahlenb.) T. Koyama, J. Fac. Sci. Univ. Tokyo, Sect. 3, Bot. 8: 80 (1961).

≡ *Uncinia spartea* (Wahlenb.) Spreng., Syst. Veg. 3: 830 (1826).

≡ *Uncinia spartea* Nees, Linnaea 10: 205 (1835-1836), nom. illeg.

≡ *Uncinia sprengelii* Nees, Linnaea 10: 205 (1835-1836), nom. superfl.

= *Carex indica* Schkuhr, Beschr. Riedgräs. 1: 37 (1801), nom. illeg., non L. (1771).
Type: [South Africa] Cap. bonae Spei, *Thunberg* (lectotype: HAL-0103917 digital image!, here designated)

≡ *Carex spartea* Thunb., Fl. Cap. 1: 343 (1811), nom. illeg.; non *C. spartea* Wahlenb. (1803)

= *Carex schimperiana* Boeckeler, Linnaea 40: 373, 1876.
Type: Ethiopia. Abyssinia, Dewra Tabor, 8,500 ft., 27-VIII-1863, *Schimper 1318* (lectotype: BM-000922720 digital image!, here designated; isolecto-: E-00286063 digital image!, K-000693814 digital image!).

≡ *Schoenoxiphium schimperianum* (Boeckeler) C.B.Clarke in Bull. Misc. Inform. Kew, Addit. Ser. 8: 67 (1908).

≡ *Schoenoxiphium sparteum* var. *schimperianum* (Boeckeler) Kük. in Engler (ed.), Pflanzenr. 38(IV, 20): 32 (1909).

= *Carex densenervosa* Chiov. in Ann. Bot. (Rome) 9: 149. 1911.
Type: [Ethiopia]. Semien pascoli ghiaiosi macchiosi aprici presso Cerà 13-VII-1909, *Chiovenda 979* (lectotype: FT-000664 digital image!, here designated; isolecto: K-000363532 digital image!). Syntype: [Ethiopia] Amhara-Dembià, Bamboló pascoli sassosi tra le macchie, 16-VII-1909, *Chiovenda 1041*, FT-000665 digital image!.

Rhizome caespitose or not, with short to very long internodes, moderately stout, dark-brown. Flowering culms (4.5)15–50(75) cm long, acutely, more rarely, obtusely trigonous, slightly scabrous at least to the apex, more rarely smooth, leafy up to ⅔ of its length, (0.4)0.6–1(1.4) mm wide at the middle. Leaves (0.4)1.5–3(5) mm wide, shorter, more rarely longer, than the inflorescence, soft to slightly rigid, straight to slightly curved at the apex, light green, flat, more rarely, scarcely carinate, in cross-section, scabrous along the margins and along the abaxial midrib to the apex; adaxial surface smooth to scarcely aculeate in some veins; ligule (0.5)1–2.5(3.5) mm long. Basal sheaths with lamina, more rarely, the lowermost bladeless, usually densely fibrous. Lowest bract of the inflorescence leaf-like, shorter, rarely longer, than the inflorescence, with a sheath (9)13–45 mm long. Inflorescence up to the upper ¾(⅘) of the length of the culm, branching up to 3 times; partial inflorescences (3)5–7(8), the (3)4–5(6) lowermost distant,

pedunculate and erect, the (1)2–3 uppermost overlapping, subsessile, rarely all sessile and overlapping (in dwarf individuals). Glumiform perigynia and glumiform cladoprophylls absent. Tubular cladoprophylls always present. Utriculiform cladoprophylls rarely present. Male glumes 2–3.2 × (1.2)1.5–2.2 mm, ovate-lanceolate to widely ovate, straw-coloured to yellowish-brown, with a green central band, acute to ending in an aculeate mucro up to 0.3(0.5) mm long. Female glumes (1.9)2–3.2(3.7) × (1)1.3–1.9(2) mm, ovate, yellowish-brown, pale reddish brown or straw coloured, with a green central band, acute to ending in an aculeate mucro up to 2(6) mm long. Unisexual utricles (2)2.5–3.3(4.1) × (0.9)1.1–1.5(1.8) mm, widely ovoid to elipsoid, stipitate, straight, more rarely slightly curved, straw-coloured to brownish when mature, smooth, with conspicuous prominent veins over the entire surface, suberect to erecto-patent, abruptly contracted, very rarely attenuate, into a smooth, occasionally (some individuals from Eastern Cape) with a few, very disperse, minute prickless, asymmetrically bidentate or irregularly truncate beak (0.2)0.5–0.8(1) mm long; rachilla usually reaching the apex of the utricle, rarely protruding from it up to 0.2(0.4) mm. Bisexual utricles frequently present, widely and obliquely truncate at the apex. Achenes (1.8)1.9–2.6(2.9) × (0.8)1–1.2(1.4) mm, ellipsoid-trigonous, more or less straw-coloured to brown when mature, tipped by a very short, obtusely trigonous, sometimes slightly and shortly neck-like, rarely asymmetric, persistent style base.

### Distribution
Eastern and southern parts of Africa and southeast of the Arabian Peninsula. Democratic Republic of the Congo (North Kivu province), Eswatini (Manzini district), Ethiopia (Amhara, Oromia and Southern Nations regions), Kenya (Baringo, Kericho, Nyandarua and Trans Nzoia counties), Lesotho (Leribe and Maseru districts), Malawi (Rumphi district), South Africa (Eastern Cape, Free State, Gauteng, KwaZulu-Natal, Limpopo, Mpumalanga and Western Cape provinces), Tanzania (Iringa region), Uganda (Northern and Western regions), Republic of Yemen (Ta'izz governorate) and Zimbabwe (Manicaland province). [23 CON 24 ETH 25 KEN TAN UGA 26 MLW ZIM 27 CPP LES NAT OFS SWZ TVL 35 YEM]. Figure 26F.

### Habitat
Dry to damp grassland, edges of streams, grassy clearing of shrubs (Albany Thicket, Fynbos, Grassland, Indian Ocean Coastal Belt and Savanna biomes); 45–2,800 m (up to 2,310 m in South Africa).

### Etymology
From the Latin *sparteus-a-um*, similar to esparto grass (*Macrochloa tenacissima* (L.) Kunth), a steppary species used for its strong fibers.

### Iconography
Figures 32C and 34. *Gordon-Gray (1995*; sub *Schoenoxiphium sparteum*).

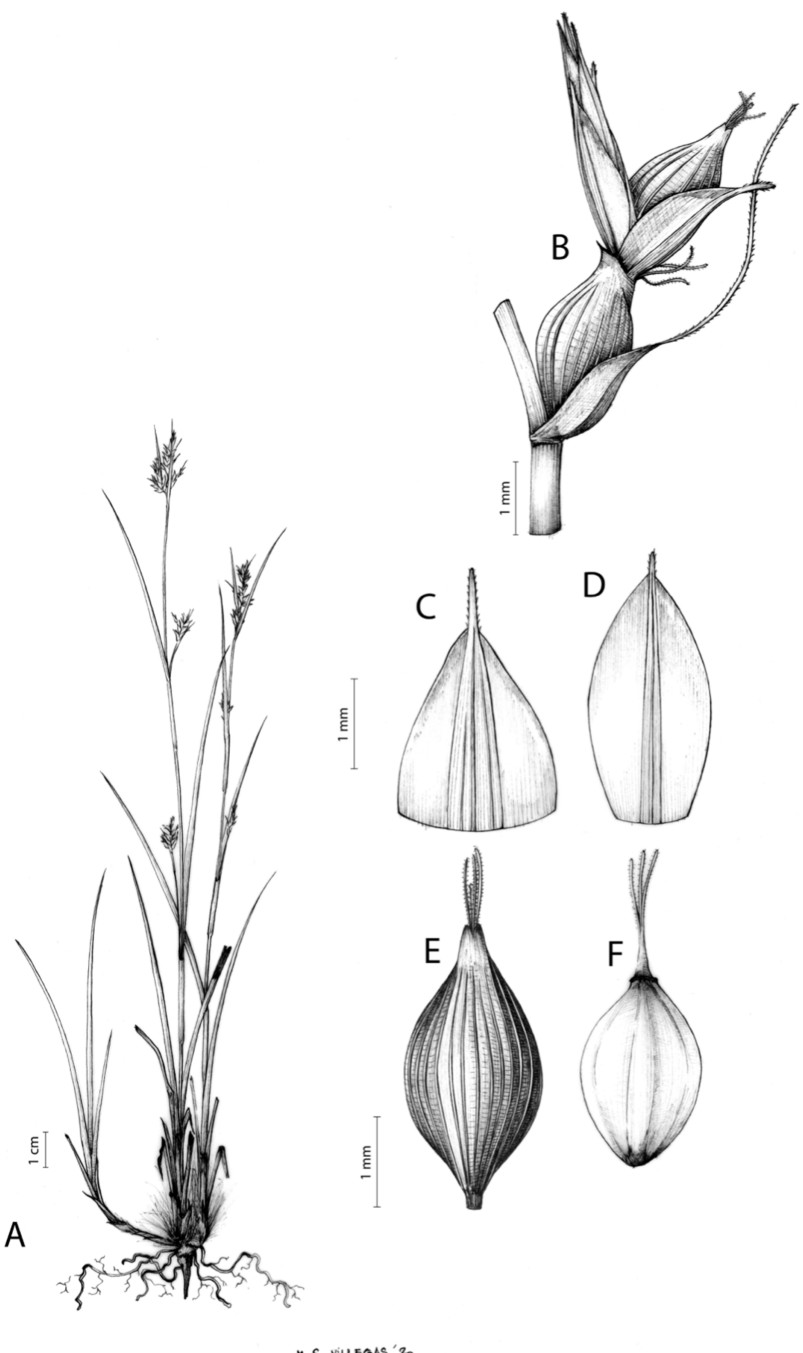

**Figure 34 Illustration of *Carex spartea*.** (A) General aspect of the plant showing a long rhizome internode, an sterile culm and two fertile culms. (B) Utriculiform cladoprophyll. (C) Female glume. (D) Male glume. (E) Unisexual utricle. (F) Achene. Illustration by M. Sánchez-Villegas.

## Conservation

Due to the long-standing problems concerning delimitation of this species, it has not previously been evaluated under the taxonomic concept here proposed, and should therefore be considered as Not Evaluated (NE).

**Notes**

This species has been often mistaken with the short, narrow leaves and small utricles (3.5–4.2 mm long) forms of *C. esenbeckiana*, from which it is differentiated because of: (i) the light green color of *C. spartea* against the ashen green or bluish-green of *C. esenbeckiana*, (ii) lower ratio of beak:body of the utricle length in *C. spartea*, (iii) the very short, obtusely trigonous (rarely slightly and shortly neck-like), persistent style base of *C. spartea* against the distinctly neck-like style base of *C. esenbeckiana*, and (iv) *C. spartea* inhabits sunny meadows, whereas the habitat of *C. esenbeckiana* includes areas of permanent shade, generally in forest understories. *Carex spartea* has also been mistaken with *C. bolusii*, but the latter present noticeable prickles in the utricle beak (very rarely some specimens show the beak almost smooth), basal sheaths entire to slightly fibrous, and smooth, obtusely trigonous culms, whereas *C. spartea* presents a smooth utricle beak–very rarely, a few individuals from Eastern Cape show some utricles with very disperse, minute prickles towards the apex, basal sheaths generally strongly fibrous and culms more or less acutely trigonous, sometimes with some prickles sparse towards the apex. Finally, *C. spartea* is easily differentiated from *C. dregeana* because of the rachilla length, that in the latter reaches a maximum of ½(¾) of the utricle length, whereas in *C. spartea* equals or surpasses the utricle length. These species are also distinguishable by the superior partial inflorescences bracts: very wide and with a scarious margin in *C. dregeana*, and narrower without bearing the scarious margin in *C. spartea*.

The specimens of *C. spartea* from its northern distribution tend to present longer leaves that frequently surpass in length the culms. However, in those areas there are also populations with leaves shorter than the culms, as well as intermediate specimens.

## CONCLUSIONS

We have here presented a fully updated and integrative monograph of *Carex* section *Schoenoxiphium* based on morphological, molecular and cytogenetic data. We recognize 21 morphologically well delimited species, and one of them, *C. gordon-grayae*, has been described here as a new species. The 20 species included in the phylogenetic study are placed in five well supported clades, except *C. acocksii* (weakly supported here as sister of Clade D, but see *Márquez-Corro et al., 2020*) and *C. gordon-grayae* (an independent lineage). Monophyly has been confirmed for all sampled species except for *C. sciocapensis* that was revealed as paraphyletic, probably due to recent divergence. For the first time, we have reported cytogenetic counts for this section, made on 44 populations and 15 taxa; we have observed a conspicuous reduction of chromosome numbers, probably due to a massive series of fusion events that have occurred through the diversification of the lineage, rather than several polyploid event. Further work is needed to (i) investigate the presence of cryptic species in Clades B and C, (ii) elucidate the phylogenetic position of *C. gordon-grayae*, and (iii) evaluate the potential role of ecological specialization and karyotype-related adaptation in the diversification of the section.

## ACKNOWLEDGEMENTS

Authors are grateful to B. Gehrke (UIB) for her help during fieldwork, previous works on *Carex* section *Schoenoxiphium* and taxonomic comments. J. Muñoz (MA) for the elaboration of some distribution maps. C. Potgieter (NU) provided us with the technical support that made chromosomal counts possible at the University of Natal. M. González Muñoz, S. Guerra-Cárdenas, V. Pineda Labella, J.M. Sierra, M. Pirie and C.H. Stirton, for help during fieldwork. J.C. Zamora for assistance during the nomenclatural survey. The staff and curators of herbaria BOL, E, EA, GRA, H, HAL, NBG, NU, P, PRE, S, SALA, UPOS, UPS for essential help during the study of herbarium collections (in situ visits, digitalization of specimens, search of type material, loan management, etc.), especially L. Scott (E), P.M. Musili (EA), V.R. Clark (GRA), A. Taponen and H. Väre (H), M. Lehnert (HAL), A. Magee (NBG), B. Bytebeier (NU), F. Jabbour (P), C. Archer, L. Makwarela, W. Sepheka and E. van Wyck (PRE), J. Lundberg (S), E. Rico (SALA), C. Barciela, J.M. Cobos, J. Fernández, I. Jurado, R. Mora, M. Parra and J.M. Yáñez (UPOS), and M. Hjertson (UPS).

### Funding

This work was supported by the Spanish Ministries of Science grants: CGL2005-06017-C02-02 (2005-2008), CGL2009-09972 (2010-2012), CGL2012-38744 (2013-2015) and CGL2016-77401-P (2017-2019). The funders had no role in study design, data collection and analysis, decision to publish, or preparation of the manuscript.

### Grant Disclosures

The following grant information was disclosed by the authors:
Spanish Ministries of Science grants: CGL2005-06017-C02-02 (2005-2008), CGL2009-09972 (2010-2012), CGL2012-38744 (2013-2015) and CGL2016-77401-P (2017-2019).

### Competing Interests

Marcial Escudero and Santiago Martín-Bravo are Academic Editors for PeerJ. The authors declare that they have no competing interests.

### Author Contributions

- Modesto Luceño conceived and designed the experiments, performed the experiments, analyzed the data, prepared figures and/or tables, authored or reviewed drafts of the paper, participated in field trips, and approved the final draft.
- Tamara Villaverde conceived and designed the experiments, performed the experiments, analyzed the data, prepared figures and/or tables, authored or reviewed drafts of the paper, participated in field trips, and approved the final draft.
- José Ignacio Márquez-Corro conceived and designed the experiments, performed the experiments, analyzed the data, prepared figures and/or tables, authored or reviewed drafts of the paper, participated in field trips, and approved the final draft.

- Rogelio Sánchez-Villegas conceived and designed the experiments, performed the experiments, analyzed the data, prepared figures and/or tables, authored or reviewed drafts of the paper, participated in field trips, and approved the final draft.
- Enrique Maguilla conceived and designed the experiments, performed the experiments, analyzed the data, prepared figures and/or tables, authored or reviewed drafts of the paper, participated in field trips, and approved the final draft.
- Marcial Escudero conceived and designed the experiments, performed the experiments, analyzed the data, prepared figures and/or tables, authored or reviewed drafts of the paper, and approved the final draft.
- Pedro Jiménez-Mejías conceived and designed the experiments, prepared figures and/or tables, authored or reviewed drafts of the paper, and approved the final draft.
- Manuel Sánchez-Villegas performed the experiments, prepared figures and/or tables, authored or reviewed drafts of the paper, participated in field trips, and approved the final draft.
- Monica Miguez analyzed the data, prepared figures and/or tables, authored or reviewed drafts of the paper, organized the field trips, and approved the final draft.
- Carmen Benítez-Benítez performed the experiments, analyzed the data, prepared figures and/or tables, authored or reviewed drafts of the paper, and approved the final draft.
- A. Muthama Muasya performed the experiments, analyzed the data, prepared figures and/or tables, authored or reviewed drafts of the paper, participated in field trips, and approved the final draft.
- Santiago Martín-Bravo conceived and designed the experiments, performed the experiments, analyzed the data, prepared figures and/or tables, authored or reviewed drafts of the paper, participated in field trips, and approved the final draft.

### Field Study Permissions

The following information was supplied relating to field study approvals (i.e., approving body and any reference numbers):

Permits to collect were issued by CapeNature (collecting number permit AAA0005-00054-0028) and Ezemvello KZN Wildlife (34/2008) and by The Lesotho Ministry of Tourism, Environment and Culture (collecting number permit MTEC/NES/CONV/1 to Tamara Villaverde).

### DNA Deposition

The following information was supplied regarding the deposition of DNA sequences:

The newly generated sequence data are available at GenBank: MW574411 to MW574420 and MW584588 to MW584617 and are available as part of Data S1 (sequence alignment used for the analyses).

Sequencing reads are available at Bioproject: PRJNA520983 and PRJNA698457.

The RAD-seq matrix is available at figshare: Villaverde, Tamara (2021): RADseq_matrix_20201218.phy.zip. figshare. Dataset. DOI 10.6084/m9.figshare.14073821.v1.

## Data Availability

Sequencing reads are found under bioprojects PRJNA520983 and PRJNA698457.

RAD-seq matrix available here: https://figshare.com/s/ba1e3d5e0aa67250f9ea.

Concatenated matrix of the four individual markers is provided in the Supplemental File Data S1.

## New Species Registration

The following information was supplied regarding the registration of a newly described species:

*Carex gordon-grayae.*

## Supplemental Information

Supplemental information for this article can be found online at http://dx.doi.org/10.7717/peerj.11336#supplemental-information.

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
