# Peer review of "An integrative monograph of Carex section Schoenoxiphium (Cyperaceae)"

_PeerJ, doi:10.7717/peerj.11336_

## Round 0.1 · original submission · Minor Revisions

Thank you for considering previous issues raised by the reviewers. The three of them and I agree that with these few corrections the paper will be ready to publish, The issues include some typographic errors and some nomenclature issues.

·

Basic reporting

This is very strong and detailed.

Experimental design

This is appropriate for the nature of the study. The research objectives are effectively stated.

It is a sound piece of work that involves a rigorous investigation of this group of plants. The Methods are described in appropriate details.

Validity of the findings

The findings are valid and robust given the many types of data considered.

The data has been provided, but additional captions would improve their usefulness for others in the future.

Additional comments

General comments
This integrative monograph is an exemplar work in the discipline that uses several types of data to support the species circumscriptions. This work should be referred to for many years to come, as the species are well-supported and described. It is also evident that this is the culmination of several years of detailed work.

My suggestions are mostly minor. I noticed a few grammatical/spelling errors throughout. I also think that a few figures could be slightly improved and many captions expanded.

Specific comments

Abstract
- Lines 37-38: I recommend to not start a new sentence with a number when using formal english.

Introduction

This is an excellent, detailed Introduction that is well-written and sufficiently justifies the study.


Results
- Line 336: I recommend to write out number when starting a sentence in formal english.

Discussion
- Line 403: I recommend to write out number when starting a sentence in formal english.

Key

It might make it a little easier for the reader if you cite/refer to your figures in the Key couplets. For example, for C. ackocksii, you could refer to Fig. 7 directly in the key.

I say this because I found the drawings very useful for understanding exactly what you mean in your leads.


Descriptions

- Line 920: I am not quite sure what you mean by CW here.

- Line 1217: Do you mean "to the apex"?

- Line 1271: "Similar ones" might be more appropriate than "close ones" here.

- Line 1321: Should this be very "abruptly"?

- Line 1346: Charles Darwin?

- Line 1513: I think that "as it is shown in the phylogeny of Figure S2" would be better here.

- Line 1765: Do you mean 'longer than the inflorescence' here?

- Line 1889: Change from 2,2 to 2.2

- Line 2149: Do you mean C. spartea or C. esenbeckiana here?

- Lines 2164-2166: I think the wording of this paragraph could be reworked. Unlike most of the other parts of the manuscript, I had to read it a few times to understand what you are trying to state here.


Table 1:
- Remove italics for authrity of S. lanceum

Table 2:
- I think that you should spell out in a little more detail the information you have about the populations of origen. For example, I assume that the bold code is a collection number, but I am not sure.



Figures

Figure 3 caption
- I think that it should read "countries are colored to reflect their Schoenoxiphium species richness"

Figure 3
- I find the legends and axes very hard to read, since they are relatively small. Would it be possible to enlarge them?
- the use of colors in Figure 3B is very difficult to follow since there are so many colors here (i.e. it is too busy). A suggestion is to divide this into separate maps for each clade.


Figure 5 caption
- I am not quite sure what the first number in brackets before the count number represents. I am assuming this is the collection number (e.g. MM4544), but I think that this needs to be stated explicitly in the caption.

Figure 15
- the clear circles are very difficult to see in this figure. I would suggest replacing them with another symbol.


Figures S1 and S2:

- I don't see a caption here. Would it be possible to include one with these .pdf's so that they are stand-alone documents?

Table S1
- this Table is also missing a caption. I quickly browsed through it and noticed that there are many spelling mistakes. I suggest spending some time to clean it up a bit before submitting it for publication so that it looks more professional.
- are these all of the specimens that were measured, or were only a subset of these specimens measured?

Table S2
- I am assuming that these were the new Sanger sequences produced for this study, but I am not sure if this since there is no Table caption.

Tables S3 and S4
- a caption would be useful here.

Reviewer 2 ·

Basic reporting

The manuscript presents complete, nice piece of work, technically very sound and written in strong professional language.
Background of the taxonomically most complicated section has well discussed in the introduction part.
Authors described one new species Carex gordon-grayii on the basis of three older collections, which is well described and nicely illustrated. Beside these, all the illustrations are well drawn; photographs are also very good and publishable resolution. Key provided here in the monograph is well constructed.
The new species (Carex gordon-grayii) described by the authors diagnose well in comparative manner with nearest species.
Authors have first time reported chromosome numbers with cytogenetic counts on 44 populations from 15 species of the section Schoenoxiphium and one hybrid, which are really worth mentioning to understand the evolution of genome size of the taxa. Reporting of Carex gordon-grayii chromosome number would have been better, though has not been performed probably because unavailability of live specimens.
References are very accurate and updated. To me, the manuscript warrants publication in PeerJ after very few minor corrections.

Experimental design

The methodologies are well described in the materials and methods section. Authors also critically performed nomenclatural revision and typified 19 names of the section as per ICN rules and recommendations. Collections of the samples were made with proper protocol required for Rare, Endemic and Endangered species. Finally, authors have performed everything required for a monographic study.

Validity of the findings

no comment

Additional comments

Lines: 37-38: “985 herbarium specimens were examined and the majority of the species were studied in the field”. Specimen serial number according to Table S1 is 885. Other 100 specimens probably included within these. If possible, please insert a column in Table S1, after the column "Voucher" to indicate total number of sheets/specimens.
Line137 ... “i) degeneration of three nuclei during pollen formation (pseudomonads)...” though http://www.mobot.org/MOBOT/research/APweb/ mentioned “pollen as pseudomonads, (grains 2-celled)”. Is the second condition found in other taxa of Cyperaceae?
Both for cytological reporting and images, authors mentioned superscripts III+II+I etc. Can you please give a note of few words for readers about these superscripts?
Can you please include few more photographic images (if you have, as you have sampled in-situ or of the original specimens; though not mandatory) as one more Figure for newly described species corresponding to Figure 19 (i.e. C. gordon-grayii) as done for other species under “Morphology of different species of Carex section Schoenoxiphium”?
In all the illustrations of Carex spp., authors represented either (A) General aspect or (A) General aspect and inflorescence. Can these be specified further?

In Figure 4A Phylogenetic relationships within Carex section Schoenoxiphium, in the tree C. gordon-grayii is spelled as C. gordon-grayi. Please check and correct it.

Lines 878-879: “...(lectotype: K-000693810 digital image!, here designated; iso: BOL-70348 digital image!). Syntype: Natal, Buchanan 328 (K-000693809 879 digital image!)”. Lectotype of the name correctly designated here. Then BOL-70348 mentioned as iso to mean isotype or isolectotype? If the specimen is isotype then it is all right but if it is isolectotype, it is sometimes written as “isolecto”. Please check in other species, where names are lectotypified. Secondly, Buchanan 328 (K-000693809) considered as syntype of the name, but in typification note (line 948) considered as paratype.

Lines 1368-1370: “Paratypes: South Africa, Eastern Cape, Hagga Hagga district, marsh, 7-II-1995, B. Sonnenberg 446 (NU-s.n.!). Ibidem, Kentani, 32º30’7.628”S 28º20’9.499”E, edge of stream in forest, 4-IV-1904, A.M. Pegler 1097 (BOL-4655 digital image!)”. Please include notes on types for the readers if possible.
Lines 2395-2396: “Vegetti AC. 2003. Synflorescence typology in Cyperaceae. Annales Botanici Fennici 40(1):35- 46 ISSN 0003-3847”. ISSN 0003-3847 is not required.
In caption of Figures 18 and 19: (A) General aspect vs. (A) General aspect and Inflorescence. But in both the illustrations inflorescence are drawn in detached condition from the mother plant. It is also found in Figures 21, 22 and 32.

·

Basic reporting

This is a very well done treatment of a group for which there was a lot of confusing information. It is concise, considering the large amount of information presented, and well written.

The organization is appropriate I spot checked bibliographic citations and found all correct.

The drawing are very nice, as are the photos and other illustrations. It is a beautifully illustrated work.

It is comprehensive and detailed, and laid out in a logical, parallel manner. I had no substantive comments placed in the text.

Experimental design

no comments

Validity of the findings

I think the findings are solid and this will set the standard for all future work.

My notes do include one item that must be corrected The one new species published herein, Schoenoxiphium gordon-grayii, has a problem that needs to be fixed.. Article 60.8. ex (a) of the International Code of Nomenclature treats the “y” in Gray as a vowel, therefore substantival epithets should be “grayi” (male) and for Kath Gordon-Gray it should be Schoenoxiphium gordon-grayae. This is a small but important matter.

On a more minor note, there are 6 names in Schoenoxiphium in IPNI not accounted for in this treatment. There can be little doubt that these are not relevant for this monograph, but there is an opportunity here for completeness and also to prevent any confusion, so these name could be disposed of briefly in a few sentences, if the authors choose.

Schoenoxiphium clarkeanum Kuk. Bull. Herb. Boissier Ser. II. iv. 49.
Schoenoxiphium fragile C.B. Clarke Bull. Misc. Inform. Kew, Addit. Ser. 8: 67. 1908
Schoenoxiphium hissaricum Pissjauk. Bot. Mater. Gerb. Bot. Inst. Komarova Akad. Nauk S.S.S.R. 12: 72. 1950
Schoenoxiphium kobresioideum Kuk. Bull. Jard. Bot. Buitenzorg ser. III, xvi. 312 (1940).
Schoenoxiphium kuekenthalianum N.A. Ivanova Bot. Zhurn. S.S.S.R. 24: 501, in obs. 1939
Schoenoxiphium laxum N.A. Ivanova Bot. Zhurn. S.S.S.R. 24: 501, in obs. 1939

Additional comments

none

---

## Round 0.2 · accepted · Accept

I appreciate your changes and for taking into account the suggestions by the three reviewers. My only suggestion is that during the editorial process you have to correct a few typos I found.